# Towards Offline Opponent Modeling with In-context Learning

**Yuheng Jing**[1,2]  **Kai Li**[1,2,†]  **Bingyun Liu**[1,2]  **Yifan Zang**[1,2]
**Haobo Fu**[6]  **Qiang Fu**[6]  **Junliang Xing**[5]  **Jian Cheng**[1,3,4,†]
[†] denotes corresponding authors
[1] Institute of Automation, Chinese Academy of Sciences
[2] School of Artificial Intelligence, University of Chinese Academy of Sciences
[3] School of Future Technology, University of Chinese Academy of Sciences
[4] AiRiA    [5] Tsinghua University    [6] Tencent AI Lab
{jingyuheng2022,kai.li,liubingyun2021,zangyifan2019,jian.cheng}
@ia.ac.cn, {haobofu,leonfu}@tencent.com, jlxing@tsinghua.edu.cn

## Abstract

Opponent modeling aims at learning the opponent's behaviors, goals, or beliefs to reduce the uncertainty of the competitive environment and assist decision-making. Existing work has mostly focused on learning opponent models *online*, which is impractical and inefficient in practical scenarios. To this end, we formalize an Offline Opponent Modeling (OOM) problem with the objective of utilizing pre-collected *offline* datasets to learn opponent models that characterize the opponent from the viewpoint of the controlled agent, which aids in adapting to the unknown fixed policies of the opponent. Drawing on the promises of the Transformers for decision-making, we introduce a general approach, **T**ransformer **A**gainst **O**pponent (TAO), for OOM. Essentially, TAO tackles the problem by harnessing the full potential of the supervised pre-trained Transformers' *in-context learning* capabilities. The foundation of TAO lies in three stages: an innovative offline policy embedding learning stage, an offline opponent-aware response policy training stage, and a deployment stage for opponent adaptation with in-context learning. Theoretical analysis establishes TAO's equivalence to Bayesian posterior sampling in opponent modeling and guarantees TAO's convergence in opponent policy recognition. Extensive experiments and ablation studies on competitive environments with sparse and dense rewards demonstrate the impressive performance of TAO. Our approach manifests remarkable prowess for fast adaptation, especially in the face of unseen opponent policies, confirming its in-context learning potency.

## 1 Introduction

A general autonomous agent must possess the ability to model complex behaviors, goals, or beliefs of other agents in the environment to refine its decision-making, a long-standing problem often referred to as **opponent modeling** in the adversarial domains (Albrecht & Stone, 2018; Nashed & Zilberstein, 2022; Yu et al., 2022). Nevertheless, most existing work adopts the setting of learning an opponent model *online*, which is prohibitively *impractical* and *inefficient*. In practical situations, it is not always feasible to engage with the competitive environment or opponent policies to execute the online learning process. For instance, e-commerce platforms are constrained to modeling the behavior of specific users offline by passively accumulating their data (Li & Zhao, 2021). Even if online learning is available, the opponent may have switched the policy across episodes, possibly long before the current opponent's model is learned. Then a multitude of rollouts are necessary to acquire a new opponent model, which is exorbitantly inefficient (Albrecht & Stone, 2018).

Given the multifaceted issues in learning the opponent model online, we contemplate extending the opponent modeling problem into the *offline* setting, a concept we refer to as **Offline Opponent Modeling (OOM)**, to make learning more accessible and enhance the sample efficiency of opponent modeling. The underlying insight is that acquiring replay data or historical plays of a certain quality

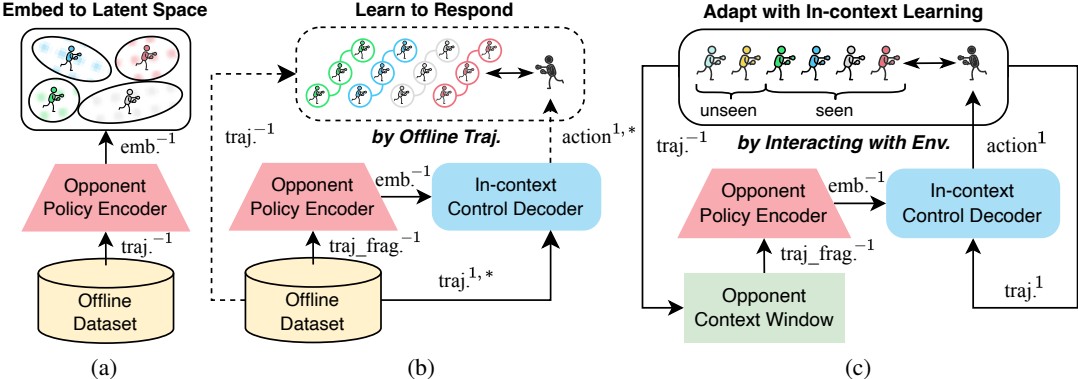

Figure 1: Procedures of TAO. (a) Offline stage 1: Embed the trajectories of different opponent policies into a latent space through representation learning. (b) Offline stage 2: Learn to respond to different opponent policies based on consecutive fragments sampled from their trajectories. (c) Deployment stage: Deploy in a new environment and adapt to unknown opponent policies through in-context learning. Superscript 1 denotes the terms of the controlled agent, $-1$ denotes the terms of the opponent, and $*$ denotes terms generated by the approximate best response policy.

is generally straightforward (Levine et al., 2020). The acquired offline data can be used to learn the response policy against the fixed opponent policy found in offline data rather than undertaking thousands of rollouts to learn through trial and error. Numerous real-life applications echo this principle. For instance, mastering strategies to handle certain scenarios in Go games through practicing high-quality Go exercises; understanding the basketball playing styles and patterns of specific opponents by analyzing replay videos of their NBA performances (Daly-Grafstein & Bornn, 2020).

Recent studies have shown that the Transformer (Vaswani et al., 2017), supervised pre-trained for decision-making, possesses a robust in-context learning ability in meta-Reinforcement Learning (RL) (Lee et al., 2023). This ability is exactly what is demanded in opponent modeling. Such insight stems from the fact that when we deploy the opponent model into a new environment, we expect to adapt to unknown opponent policies as swiftly as possible, ideally without gradient updates. We provide the **related work** on *opponent modeling*, *offline meta-RL*, etc. in Appendix A.

Embracing this idea, we introduce **T**ransformer **A**gainst **O**pponents **(TAO)**, a general Transformer-based approach dedicated to resolving the defined OOM problem. The procedures of TAO unfolds in three stages (see Fig. 1): 1) We *generatively* and *discriminatively* embed opponent trajectories using an **Opponent Policy Encoder (OPE)** to acquire an *opponent policy representation* that is beneficial for the downstream learning; 2) Upon the characterization of opponent policies offered by OPE, we adopt *in-context learning-based* supervised pre-training for an **In-context Control Decoder (ICD)** to learn to respond reasonably according to opponent trajectories; 3) We deploy in a new environment inhabited by unknown opponent policies and adapt with in-context learning with the help of an **Opponent Context Window (OCW)**, which continuously collects opponent trajectories.

Theoretically, we prove that TAO is equivalent to Bayesian posterior sampling in opponent modeling. Further analysis demonstrates the convergence of TAO for opponent policy recognition. We conduct experiments in both a sparse reward and a dense reward environment, comparing TAO with a series of baselines for opponent modeling and offline meta-RL. Our approach demonstrates outstanding performance under different kinds of test settings. Concurrently, ablation studies on TAO's components underscore their vital roles in the process. Surprisingly, when we deploy TAO into a new environment inhabited by unknown opponent policies, it exhibits a faster and superior adaptation than other approaches, particularly in confronting previously unseen opponent policies. This underlines the formidable in-context learning capability inherent in our approach.

## 2 PROBLEM FORMULATION

We use the Partially-Observable Stochastic Game (POSG) (Yang & Wang, 2020) for a basic formalization of the competitive environment. A POSG is defined by a tuple $\langle \mathbb{I}, \mathcal{S}, \{\mathcal{O}^i\}_{i \in \mathbb{I}}, \mathcal{A}, T, \{R^i\}_{i \in \mathbb{I}}, \{\Omega^i\}_{i \in \mathbb{I}} \rangle$, where $\mathbb{I} = \{1, 2, \ldots, N\}$ is the set of agents. $\mathcal{S}$ is the state space. $\mathcal{O}^i$ is the observation

space of agent $i$. $\mathcal{A} = \mathcal{A}^1 \times \mathcal{A}^2 \times \cdots \times \mathcal{A}^N$ is the joint action space. $T : \mathcal{S} \times \mathcal{A} \times \mathcal{S} \to [0,1]$ denotes the transition dynamics, which defines the probability distribution on the next state given the previous state and the joint action. $R^i : \mathcal{S} \times \mathcal{A} \times \mathcal{S} \to \mathbb{R}$ denotes the reward function of agent $i$. $\Omega^i : \mathcal{S} \times \mathcal{A} \times \mathcal{O}^i \to [0,1]$ denotes the agent $i$'s observation function, which defines the probability distribution over its possible next observation given the previous state and the joint action.

In this study, we utilize 1 to denote the controlled agent and $-1$ to denote the opponent and focus on modeling one opponent. We assume that the opponent's policy originates from a set of fixed policies $\Pi = \{\pi^{-1,k}(a^{-1}|o^{-1})\}_{k=1,2,\dots,K}$, which are obtained by the scripts or RL algorithms pre-training.

## 2.1 Offline Opponent Modeling

Assuming that the interaction between the controlled agent and the opponent policy $\pi^{-1,k}$ generates their respective trajectories, denoted as $\tau^{1,k} = (o_0^{1,k}, a_0^{1,k}, r_0^{1,k}, o_1^{1,k}, a_1^{1,k}, r_1^{1,k}, \dots) \in \mathcal{T}^{1,k}$ and $\tau^{-1,k} = (o_0^{-1,k}, a_0^{-1,k}, r_0^{-1,k}, o_1^{-1,k}, a_1^{-1,k}, r_1^{-1,k}, \dots) \in \mathcal{T}^{-1,k}$. The resultant dataset is thus denoted as $\mathcal{D}^k = \{\mathcal{T}^{1,k}, \mathcal{T}^{-1,k}\}$. Within the context of offline learning, we presume the availability of the dataset $\mathcal{D}^{\text{off}} = \{\mathcal{D}^k\}_{k=1,2,\dots,K}$. Specifically, $\mathcal{T}^{1,k}$ is acquired through interactions with $\pi^{-1,k}$ while employing its approximate best response policy $\pi^{1,k,*}$, usually with certain noise.

The objective of OOM is to use $\mathcal{D}^{\text{off}}$ to pre-train an *opponent-aware adaptive* controlled agent policy $M_\theta(a^1|o^1; D)$ and deploy $M_\theta$ into a new environment with an unknown *test opponent policy set* $\Pi^{\text{test}}$, such that the controlled agent achieves the maximum expected *return* (*i.e.*, cumulative reward):

$$\max_\theta \mathbb{E}_{\pi^{-1} \sim \Pi^{\text{test}}, \mathcal{D}^{\text{off}}, T, \Omega} \left[ \sum_{t=0}^{\infty} R_t^1 \,\middle|\, a_t^1 \sim M_\theta, \pi^{-1} \right]. \tag{1}$$

$D$ is the opponent's information data, which is sampled from $\mathcal{D}^{\text{off}}$ during offline pre-training and is required to be collected during deployment. See Appendix B for detailed assumptions of OOM.

## 2.2 Rationale behind OOM

**Why not learning online?** On the one hand, online learning struggles to satisfy real-world scenarios, given that interactions with opponent policies are not consistently accessible. On the other hand, online learning of opponent models bears substantial complexity and inefficiency. As with many existing algorithms, they gather copious data and undertake intricate online updating processes.

**Relationship with offline meta-RL.** The core objective of offline meta-RL lies in addressing *task* adaptation problems using offline datasets within a single-agent environment. In contrast, OOM mainly focuses on *opponent* adaptation using offline datasets within a multi-agent setting. In a broader sense, OOM can be perceived as a specialized extension of offline meta-RL. Specifically, the heightened challenge of OOM compared to offline meta-RL stems from the increased complexity of environment dynamics arising due to opponent policies and the involvement of multiple agents.

**Main Challenges.** As implied by Eq. (1), the main challenges of OOM arise from several dimensions: 1) OOM requires effective learning of high-level semantics from different opponent policies to aid controlled agent's decision-making; 2) OOM necessitates extraction of robust response policies against corresponding opponent policies from $\mathcal{D}^{\text{off}}$; 3) OOM emphasizes generalization, a crucial aspect for transferring the garnered high-level knowledge to adapt to unseen opponent policies.

## 3 Transformer Against Opponent

We propose a three-stage Transformer-based approach, TAO, to address the three challenges of OOM mentioned above respectively. Firstly, we argue that opponent policies, compared to a naive task, involve more complex semantics, necessitating the learning of a good representation. To achieve this, we propose **Policy Embedding Learning (PEL)** to induce opponent policy embeddings conducive to downstream learning in Section 3.1; Secondly, to recognize and respond to the current opponent policy using only a subset of opponent trajectories, we introduce **in-context learning-based** supervised pre-training in Section 3.2. This pre-training enables predicting near-optimal actions conditioned on $D$; Thirdly, we aim to generalize the knowledge acquired from the offline dataset to respond to unknown opponent policies during deployment. Therefore, we collect

opponent trajectories during deployment, leveraging the model's in-context learning ability to adjust responses to continually adapt to the current opponent policy, as described in Section 3.3; Lastly, we theoretically analyze TAO's convergence on opponent policy recognition in Section 3.4.

### 3.1 OFFLINE STAGE 1: POLICY EMBEDDING LEARNING

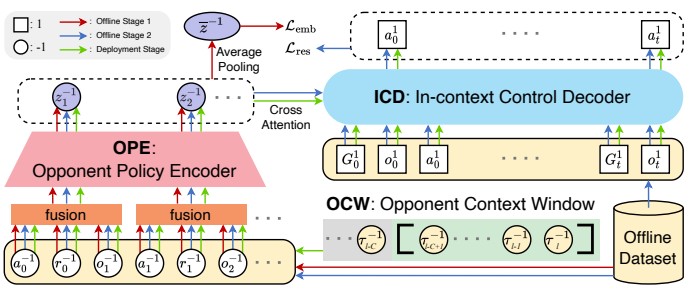

Figure 2: The overall architecture and flow of TAO.

In anticipation of attaining an opponent policy representation favorable to downstream learning, we propose PEL to characterize the distinct behavioral patterns and high-level semantics of different opponent policies. We aim for the resulting policy embedding to be both *generative* and *discriminative*: 1) *Generative*: the policy embedding should have the ability to emulate different opponent policies;

2) *Discriminative*: the policy embedding should be adept at distinguishing between different opponent policies. Intuitively, generative helps policy embedding implicitly recognize opponent policies, as different policies provide different soft labels. Discriminative helps policy embedding distinguish the opponent policies that are hard to distinguish.

Specifically, using an OPE $M_{\theta_e} : \mathcal{T}^{-1} \to \mathcal{Z}$ parameterized by $\theta_e$, we conduct PEL with opponent trajectories stemming from $\mathcal{D}^{\text{off}}$, where $\mathcal{T}^{-1} = \{\mathcal{T}^{-1,k}\}_{k=1,\dots,K}$. $z^{-1} = (z_0^{-1}, z_1^{-1}, \dots) \in \mathcal{Z}$ denotes a *trajectory embedding* consisting of *per-timestep embedding tokens* (*i.e.*, $z_t^{-1}$). As shown in Fig. 2, we fuse the embedding tokens of $a^{-1}, r^{-1}, o^{-1}$ and employ Average Pooling, denoted as function AP, to aggregate $z^{-1}$ to acquire an *average trajectory embedding* $\text{AP}(z^{-1}) = \overline{z}^{-1} \in \overline{\mathcal{Z}}$ for subsequent loss calculations. The implementation of OPE is based on GPT2 (Radford et al., 2019).

**Generative Loss.** As we aim at acquiring the ability to emulate different opponent policies, we propose conditional imitation learning (Hussein et al., 2017) using OPE $M_{\theta_e}$ and an ancillary decoder $\pi_{\phi_d} : \mathcal{O}^{-1} \times \overline{\mathcal{Z}} \to \mathcal{A}^{-1}$, parameterized by $\phi_d$. For a given opponent policy $\pi^{-1,k}$, we predict its actions $a^{-1,k}$ conditioned on its observations $o^{-1,k}$ along with the *average trajectory embeddings* $\overline{z}^{-1,k}$ obtained from its other trajectory. The corresponding generative loss is:

$$\mathcal{L}_{\text{gen}} = -\frac{1}{K} \sum_{k=1}^{K} \mathbb{E}_{\tau_i^{-1,k}, \tau_j^{-1,k} \sim \mathcal{T}^{-1,k}} \left[ \sum_{\langle o,a \rangle \sim \tau_i^{-1,k}} \log \pi_{\phi_d} \left( a \mid o, \text{AP} \left( M_{\theta_e} \left( \tau_j^{-1,k} \right) \right) \right) \right]. \quad (2)$$

Hence, by leveraging an auxiliary task that imitates the opponent's policies, we indirectly guide the *trajectory embeddings* of various opponent policies toward generative. Such generative imbues the policy embedding with certain recognition capabilities, as the actions of different opponent policies serve as soft labels for their respective policy types.

**Discriminative Loss.** As we expect the policy embedding to be adept at distinguishing between different opponent policies, we use the labels of *opponent policy type* (*i.e.*, index in *offline opponent policy set* $\Pi^{\text{off}}$) corresponding to the trajectories for an adequate distinction between them. We employ dot product to measure the similarity between two distinct *average trajectory embeddings*. Specifically, we maximize this similarity from the same opponent policy while minimizing this similarity from different opponent policies. The corresponding discriminative loss is:

$$\mathcal{L}_{\text{dis}} = -\mathbb{E}_{\mathcal{T}^{-1}} \left[ \sum_{i=1}^{B} \frac{1}{|\mathcal{E}_i| - 1} \sum_{j \in \mathcal{E}_i, j \neq i} \log \frac{\exp \left( \overline{z}_i^{-1} \cdot \overline{z}_j^{-1} / p \right)}{\sum_{l \in [B], l \neq i} \exp \left( \overline{z}_i^{-1} \cdot \overline{z}_l^{-1} / p \right)} \right]. \quad (3)$$

Here, $\overline{z}_i^{-1} = \text{AP}(M_{\theta_e}(\tau_i^{-1}))$, where $\tau_i^{-1} \sim \mathcal{T}^{-1}$. $B$ is the batch size of sampling. $p$ denotes the temperature factor. We define $\mathcal{K} : \mathcal{T}^{-1} \to [K]$ as an indicator function of the opponent policy type, then $\mathcal{E}_i = \{j \in [B] \mid \mathcal{K}(\tau_i^{-1}) = \mathcal{K}(\tau_j^{-1})\}$. Such a method drives trajectory embeddings of the same opponent policy towards proximity, while those from different opponent policies are pushed towards

dissimilarity. Intuitively, this could help recognize the different opponent policies, as even if the two opponent policies are similar, their policy embeddings will be relatively separate.

The comprehensive loss function for offline stage 1 is:

$$\mathcal{L}_{\text{emb}} = \alpha \cdot \mathcal{L}_{\text{gen}} + \lambda \cdot \mathcal{L}_{\text{dis}}. \tag{4}$$

Within, $\alpha$ and $\lambda$ are weighting coefficients for generative and discriminative losses.

## 3.2 OFFLINE STAGE 2: OPPONENT-AWARE RESPONSE POLICY TRAINING

Given that our goal is to handle multiple opponent policies using a single model, we necessitate a mechanism to recognize the current opponent's policy and subsequently produce the appropriate response actions. To this end, leveraging the characterization of the opponent policies offered by the trained OPE, we propose *in-context learning-based* supervised pre-training to learn an ICD $M_{\theta_d}$ : $\mathcal{Z} \times \mathcal{T}^1 \to \mathcal{A}^1$, parameterized by $\theta_d$, with the same $\mathcal{D}^{\text{off}}$. Formally, ICD autoregressively predicts near-optimal actions $a^{1,*}$ conditioned on trajectory $\tau^{1,*}$ and opponent's *trajectory embedding* $z^{-1}$.

Given the *opponent in-context data* (*i.e.*, $D$ in Section 2.1), we encode it using OPE to obtain a good representation $z^{-1}$. Conditioned on $z^{-1}$, ICD is designed to recognize the current opponent policy and generate an appropriate response. Intuitively, such supervised pre-training induces an opponent-aware adaptive response policy capable of recognizing and adapting to changes in opponent polices.

**Response Loss.** Postulating $\theta = \{\theta_e, \theta_d\}$, in-context learning-based supervised pre-training loss is:

$$\mathcal{L}_{\text{res}} = -\frac{1}{K} \sum_{k=1}^{K} \mathbb{E}_{\tau^{1,k} \sim \mathcal{T}^{1,k}} \left[ \sum_{\langle y_t, a_t \rangle \sim \tau^{1,k}} \log M_{\theta_d} \left( a_t \mid y_t; M_{\theta_e} \left( \texttt{GetOffD} \left( \mathcal{T}^{-1,k} \right) \right) \right) \right]. \tag{5}$$

Here, $y_t = (G_0, o_0, a_0, \ldots, G_t, o_t)$. $G_t = \sum_{t'=t}^{\infty} R_{t'}$ denotes Reward-To-Go (RTG). We use a funtion $\texttt{GetOffD}$ to sample opponent in-context data $D$. $\texttt{GetOffD}$ begins by sampling $C$ trajectories from $\mathcal{T}^{-1,k}$, following which it samples $H$ consecutive *fragments* $\{(a_{h'-1}^{-1,k}, r_{h'-1}^{-1,k}, o_{h'}^{-1,k})\}_{h'=h}^{h+H-1}$ from each trajectory and ultimately stitches them together. Such sampling method of opponent in-context data stems from the intuitions that given an opponent policy, its play style might be more pronounced over continuous timesteps (so we sample consecutive fragments), and it can showcase different playing behaviors across various episodes (so we sample from multiple trajectories).

Fig. 2 shows the flow of offline stage 2, and we use the same causal Transformer as in Chen et al. (2021) for the implementation of ICD. Intrinsically, we apply cross-attention to discover the association between the opponent's policy and the corresponding response policy. Thus, we have established an implicit mechanism for discerning opponent policies and generating respective responses.

## 3.3 DEPLOYMENT STAGE: IN-CONTEXT OPPONENT ADAPTING

We expect to generalize the knowledge acquired from $\mathcal{D}^{\text{off}}$ to respond to unknown opponent policies during deployment. Therefore, we introduce an OCW $\mathcal{W}$ to collect opponent trajectories to leverage $M_\theta$'s in-context learning ability to continually adapt to the current opponent policy.

$\mathcal{W}$ retains the nearest $C$ trajectories of the opponent $\{\tau_{l-C+1}^{-1}, \ldots, \tau_{l-1}^{-1}, \tau_l^{-1}\}$, where $l$ is the index of the last episode. The OCW and the flow of deployment stage are shown in Fig. 2. While interacting with the new environment, actions of the controlled agent are sampled using following formula:

$$a_t^1 \sim M_{\theta_d} \left( \cdot \mid y_t^1; M_{\theta_e} \left( \texttt{GetOnD} \left( \mathcal{W} \right) \right) \right). \tag{6}$$

The function $\texttt{GetOnD}$ samples $H$ *fragments* from each trajectory of $\mathcal{W}$ and then stitches them together. Thus, by freezing $\theta$, we can adapt to diverse opponent policies by accumulating their trajectory data.[1] The key impetus behind our approach is: If the current opponent policy is similar or identical to that of a prior one from $\Pi^{\text{off}}$, we aim to re-engage the suitable responses quickly. Conversely, if unfamiliar, we strive to exploit it by extrapolating from the existing knowledge.

---

[1]See the pseudocode of TAO in Appendix C.

## 3.4 THEORETICAL ANALYSIS

We conduct theoretical analysis from the following two aspects. 1) **Theorem 3.1**: We prove that under some reasonable assumptions, TAO is equivalent to *Posterior Sampling in Opponent Modeling (PSOM)*, an algorithm built upon *Posterior Sampling (PS)*, where PS guarantees convergence and is proven to be a *sample-efficient* RL algorithm (Osband et al., 2013). The detailed introduction of PSOM is in Appendix D.1; 2) **Theorem 3.2**: We prove the convergence of the PSOM algorithm for opponent policy recognition and demonstrate that PSOM converges to the optimal solution if PS converges to the optimal solution. Based on Theorem 3.1, TAO has the same guarantees as PSOM.

**Theorem 3.1** (Equivalence of TAO and PSOM). *Assume that the learned model $M_\theta$ is consistent and the sampling of $o^{-1}$ from $\mathcal{T}_{pre}^{-1}$ is independent of the opponent policies, then given $\bar{\pi}^{-1}$ and its $\mathcal{W}$, we have $P(\xi_H^1|\mathcal{W}, \bar{\pi}^{-1}; PSOM) = P(\xi_H^1|\mathcal{W}, \bar{\pi}^{-1}; M_\theta)$ for all possible $\xi_H^1$.*

Given any $\pi^{-1} \in \Pi^{\text{train}}$, $\mathcal{T}_{\text{pre}}^{-1}(\cdot; \pi^{-1})$ denotes the *probability distribution on the trajectories set* $\mathcal{T}^{-1}$ of $\pi^{-1}$. $\xi_H$ denotes *history* consisting of $(o, a^*)$ with length $H$, where $a^*$ is the near-optimal action. $\bar{\pi}^{-1}$ denotes the true opponent policy interacted with during deployment and $\mathcal{W}$ consists of $\bar{\pi}^{-1}$'s in-context data. $P(\xi_H^1|\mathcal{W}, \bar{\pi}^{-1}; \text{PSOM})$ and $P(\xi_H^1|\mathcal{W}, \bar{\pi}^{-1}; M_\theta)$ denotes the same form of probability of generating $\xi_H^1$ conditioned on given $\mathcal{W}$ and $\bar{\pi}^{-1}$ under PSOM and TAO, respectively. We provide a detailed explanation and a complete proof of Theorem 3.1 in Appendix D.2.

**Theorem 3.2.** *When the true opponent policy $\bar{\pi}^{-1}$ originates from $\Pi^{\text{off}}$, the PSOM algorithm using $(o^{-1}, a^{-1})$ tuples as in-context data guarantees to converge, and it converges to the optimal solution if the original PS using $(o^1, a^1, o'^1, r^1)$ tuples as in-context data converges to the optimal solution.*

Founded on Theorem 3.1, TAO shares the same properties. See the full proof of Theorem 3.2 in Appendix D.3. Note that the proof is built upon using $(o^{-1}, a^{-1})$ tuples as in-context data, while TAO uses $(o^{-1}, a^{-1}, r^{-1})$ in implementation, providing more information and assuring the guarantees.

Intuitively, when the current true opponent policy $\bar{\pi}^{-1}$ belongs to $\Pi^{\text{off}}$, if PS correctly recognizes the $\bar{\pi}^{-1}$, then PSOM is guaranteed to recognize it correctly. Even if PS fails to recognize the $\bar{\pi}^{-1}$ correctly, PSOM still has a possibility of making a correct recognition. Based on Theorem 3.1, TAO shares the same properties as the PSOM algorithm mentioned above.

## 4 EXPERIMENTS

### 4.1 EXPERIMENTAL SETUP

**Environments.** We consider two emblematic competitive environmental benchmarks: 1) `Markov Soccer (MS)` (Lanctot et al., 2019): A sparse-reward discrete soccer environment that exemplifies a two-player zero-sum game; 2) `Particleworld Adversary (PA)` (Lowe et al., 2017): A dense-reward continuous multi-particle environment that exemplifies a three-player non-zero-sum game. See Appendix E for the environmental specifics.

**Baselines.** Given that most opponent modeling approaches employ online learning settings, we draw comparisons with *Embedding-based Opponent Modeling* approaches - these are comparatively straightforward to adapt to the OOM setting. To ensure an equitable comparison, we mandate all approaches to use the same neural architecture as ours (see Appendix F). The baselines include: 1) DRON-concat (He et al., 2016a): Encode the opponent's hand-crafted features with an auxiliary network; 2) DRON-MoE (He et al., 2016a): Encode the opponent's hand-crafted features using a Mixture-of-Expert (MoE) structure; 3) LIAM (Papoudakis et al., 2021): Encode opponent information with AutoEncoders (AE); 4) MeLIBA (Zintgraf et al., 2021): Encode opponent information with variational AE; 5) Prompt-DT (Xu et al., 2022): A offline meta-RL approach based on DT and prompt learning with the potential to tackle OOM; 6) TAO without PEL (abbreviated as TAO w/o PEL): A variant of TAO without pre-training OPE via $\mathcal{L}_{\text{emb}}$. See Appendix G for details of baselines.

**Opponent policy and offline dataset.** We provide the details of the opponent policies in Appendix H and the offline datasets construction procedure in Appendix I.

**Test settings.** We focus on testing against a *non-stationary unknown* opponent. *Non-stationary* refers to the opponent switching its policy every $E$ episodes, which is concealed from all approaches. Meanwhile, *unknown* signifies that the *type* (*i.e.*, index in $\Pi^{\text{test}}$) of the opponent's policy remains

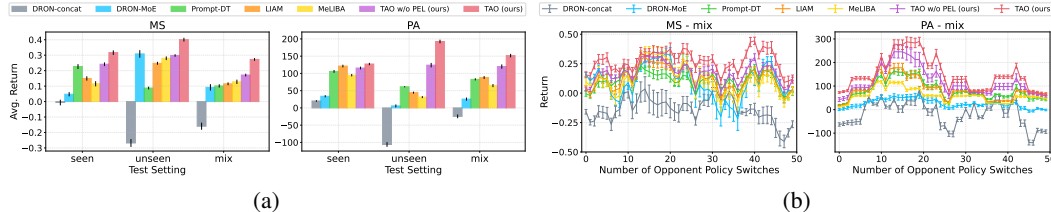

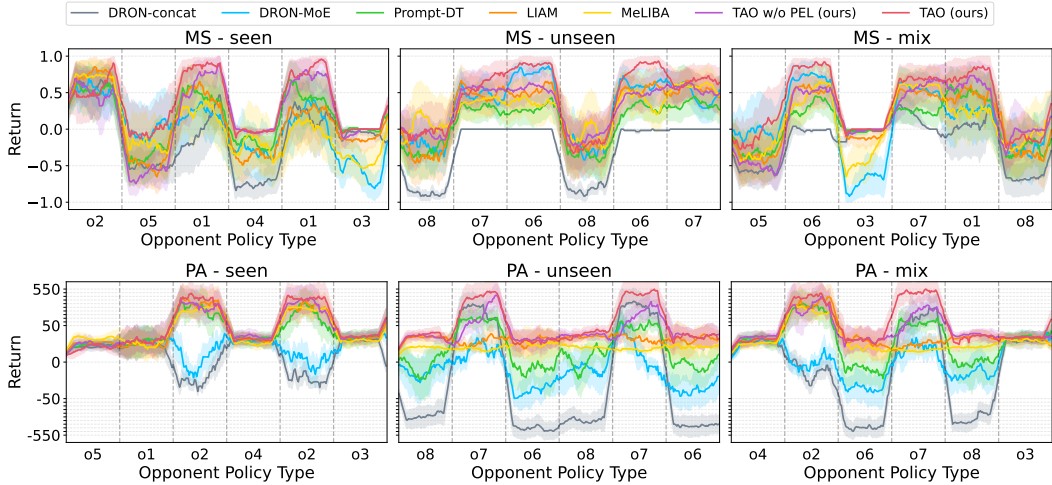

Figure 3: (a) Average performance under different test settings. The evaluation metrics for MS and PA are *return*. (b) Overall performance under the *mix* setting, where each point indicates the average result of $E$ episodes against an unknown opponent policy that the opponent switches to.

Figure 4: Detailed performance under three settings. We use $o1, \ldots, o5$ to denote all opponent policy types in *seen* and $o6, \ldots, o8$ to denote all those in *unseen*. The y-axis tick interval of PA is $50$.

perpetually concealed. We adopt three kinds of test settings for $\Pi^{\text{test}}$, where *seen* is equivalent to the *offline opponent policy set* $\Pi^{\text{off}}$, *unseen* contains strictly previously unseen opponent policies, and *mix* is a mixture of the former two. For all testing experiments, we adopt the same specifications as follows: 1) $E$ is set to 50; 2) We evaluate all approaches against the non-stationary unknown opponent over 2500 episodes; 3) We adopt the checkpoints at 2000 training steps of offline training to perform testing for all approaches; 4) We report the average and standard error of the mean results over 5 random seeds. See hyperparameter settings in Appendix J.

## 4.2 EMPIRICAL ANALYSIS

To comprehensively analyze TAO's effect and underlying mechanisms, we ask the following questions and design corresponding experiments to answer them:

**Question 1.** *Can TAO achieve competitive results against other approaches for the OOM problem?*

In Fig. 3(a), we show the average total performance under three different online test settings. We also provide Fig. 3(b) to show the smoothed per-opponent policy performance curve under the *mix* setting. Across the three settings in MS and PA, TAO demonstrates competitive results compared to other baselines, particularly in the *unseen* setting. This implies that TAO can respond reasonably to unknown opponent policies. It's worth noting that even without PEL, TAO still achieves results generally not worse than other baselines, emphasizing the critical role of TAO's design choice of in-context learning-based training.

Under the three settings, LIAM and MeLIBA generally show better results than DRON-MoE and DRON-concat. This could be due to the fact that reconstructing the opponent's information is more effective in uncovering the underlying relationship between the opponent's and controlled agent's policies than direct encoding. DRON-MoE surpasses DRON-concat in general, probably because the MoE network implicitly selects more appropriate responses than concatenating the hidden states. Under the *seen* setting, Prompt-DT's performance closely approaches that of TAO w/o PEL, suggesting that return-conditioned expert demonstrations may also indicate the current opponent's policy

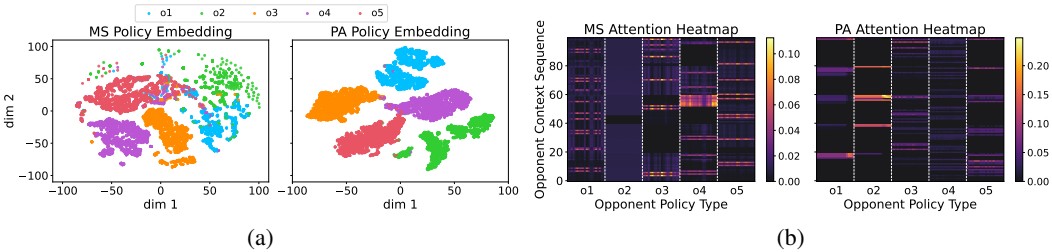

Figure 5: (a) Visualization for the dimensionality-reduced embeddings of different opponent policies with $t$-SNE. (b) The attention heatmap of TAO against different opponent policies. For each opponent policy type, we visualize the attention weights over the final 20 timesteps in an episode.

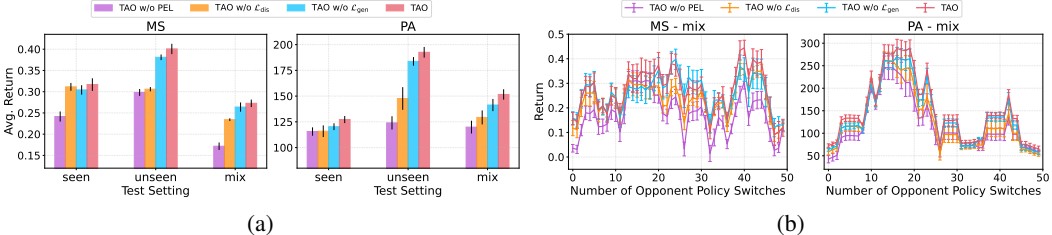

Figure 6: (a) and (b) are the ablation experiments of PEL under the same plotting settings as in Fig. 3.

to a certain extent. Whereas under the *unseen* and *mix* settings, Prompt-DT performs poorly, highlighting the difficulty of achieving effective generalization to unseen opponent policies when relying solely on the controlled agent's information without incorporating the opponent's. This suggests that the original offline meta-RL approaches might not be enough for tackling the OOM problem.

Fig. 4 illustrates the smoothed per-episode performance curve. Given that the $\Pi^{\text{test}}$ settings we employ encompass opponent policies with varying strengths, the game results exhibit substantial disparities accordingly. TAO delivers stable and favorable performance in MS and PA. We notice that TAO shows some adaptability, trying to stabilize its exploitation of the same opponent's policy throughout games and adjusting when the opponent's policy switches. In the *unseen* and *mix* settings, this adaptability of TAO becomes more pronounced in general. In contrast, other approaches always fail to respond to unseen opponent policies reasonably, and, in certain instances, they are susceptible to greater exploitation by the opponent.

**Question 2.** Does TAO learn a good policy embedding (representation) of opponent policies?

We use the OPE trained by offline stage 2 to embed all opponent trajectories in $\mathcal{D}^{\text{off}}$ and visualize the *dimensionality-reduced embeddings* (denoted as $\overline{z}_{\text{reduced}}^{-1}$) in Fig. 5(a). The colors indicate the respective opponent policy type to which $\overline{z}_{\text{reduced}}^{-1}$ belongs. It can be observed that $\overline{z}_{\text{reduced}}^{-1}$ corresponding to each opponent policy is roughly clustered into the same cluster. Intuitively, TAO learns a policy embedding that distinguishes different opponent policies well, showing, to a certain extent, that our loss design for PEL has achieved the effect we expected. This can help TAO's downstream learning, which is verified by the overall better results of TAO compared to TAO w/o PEL. The clustering effect in PA is more pronounced than in MS. This could be attributed to the greater divergence among opponent policies in PA, making them easier to differentiate.

In Fig. 5(b), we show the attention heatmap of TAO. This heatmap visualizes TAO's cross-attention weight across different opponent policies. The x-axis represents different opponent policy types. The y-axis represents the position in the *opponent context sequence* obtained by GetOnD($\mathcal{W}$), and each position has a weight indicated by the depth of color. In MS and PA, TAO exhibits a distinctive attention pattern for different opponent policies, with a propensity to concentrate on specific timesteps. Meanwhile, TAO maintains a relatively consistent attention distribution for a given opponent policy across various timesteps within the same episode. This suggests that TAO has learned a representation that is helpful with discerning the different opponent policies. Moreover, it has discovered some unique characteristics of each opponent policy's action sequence.

**Question 3.** *What is the contribution of each component in PEL to tackling the OOM problem?*

Table 1: Average performance on different opponent policy configurations.

| | | Opponent Policy Configuration 1 | | | Opponent Policy Configuration 2 | | |
|---|---|---|---|---|---|---|---|
| | | *seen* | *unseen* | *mix* | *seen* | *unseen* | *mix* |
| MS
(Avg. Return ↑) | TAO (ours) | **0.51 ± 0.01** | **0.55 ± 0.01** | **0.43 ± 0.01** | **0.43 ± 0.01** | **0.65 ± 0.02** | **0.40 ± 0.01** |
| | TAO w/o PEL (ours) | 0.46 ± 0.01 | 0.52 ± 0.01 | 0.36 ± 0.01 | 0.41 ± 0.00 | 0.42 ± 0.02 | 0.35 ± 0.01 |
| | DRON-concat | 0.15 ± 0.02 | -0.02 ± 0.05 | -0.05 ± 0.03 | -0.05 ± 0.01 | -0.21 ± 0.01 | -0.17 ± 0.01 |
| | DRON-MoE | 0.15 ± 0.01 | 0.38 ± 0.02 | 0.15 ± 0.02 | 0.09 ± 0.01 | 0.31 ± 0.04 | 0.11 ± 0.01 |
| | LIAM | 0.43 ± 0.01 | 0.47 ± 0.01 | 0.32 ± 0.01 | 0.23 ± 0.01 | 0.47 ± 0.01 | 0.24 ± 0.01 |
| | MeLIBA | 0.38 ± 0.01 | 0.41 ± 0.01 | 0.30 ± 0.01 | 0.29 ± 0.01 | 0.42 ± 0.01 | 0.25 ± 0.01 |
| | Prompt-DT | 0.48 ± 0.01 | 0.36 ± 0.01 | 0.33 ± 0.01 | 0.39 ± 0.02 | 0.20 ± 0.01 | 0.21 ± 0.02 |
| PA
(Avg. Return ↑) | TAO (ours) | **105.6 ± 3.0** | **109.5 ± 2.4** | **101.6 ± 3.4** | **33.5 ± 0.5** | **39.2 ± 0.9** | **35.3 ± 0.8** |
| | TAO w/o PEL (ours) | 69.3 ± 2.7 | 81.5 ± 7.9 | 67.4 ± 4.4 | 25.9 ± 0.7 | 31.2 ± 1.1 | 29.4 ± 0.8 |
| | DRON-concat | 21.4 ± 3.3 | -40.0 ± 10.4 | -12.7 ± 4.9 | 28.0 ± 0.8 | 21.5 ± 0.8 | 26.0 ± 0.4 |
| | DRON-MoE | 94.2 ± 1.7 | 20.5 ± 0.6 | 60.9 ± 1.0 | 33.1 ± 0.6 | 18.5 ± 0.4 | 27.4 ± 0.5 |
| | LIAM | 83.6 ± 5.9 | 60.5 ± 3.5 | 70.5 ± 4.3 | 33.5 ± 0.6 | 17.1 ± 0.4 | 27.6 ± 0.6 |
| | MeLIBA | 72.4 ± 5.6 | 48.5 ± 3.5 | 58.2 ± 4.5 | 30.8 ± 0.7 | 19.3 ± 0.4 | 27.1 ± 0.6 |
| | Prompt-DT | 66.9 ± 8.2 | -10.1 ± 4.0 | 23.8 ± 9.9 | 22.5 ± 0.8 | 25.0 ± 0.3 | 25.1 ± 0.2 |

In Fig. 6, we show the results for ablations of PEL. TAO without $\mathcal{L}_{gen}$ sets $\alpha$ to 0, while TAO without $\mathcal{L}_{dis}$ sets $\lambda$ to 0. TAO w/o PEL performs the worst under all three settings, while in **Question 1**, we found that TAO w/o PEL showed good results and verified the effectiveness of in-context learning-based supervised pre-training. The improvement in results suggests that PEL further benefits the TAO's subsequent in-context learning-based training process as we intended. TAO without $\mathcal{L}_{gen}$ is almost uniformly superior to TAO without $\mathcal{L}_{dis}$, while consistently worse compared to TAO, implying that discriminative learning makes a more critical contribution to the improvement of TAO's ability. One possible explanation is that discerning different opponent policies holds greater significance for the controlled agent's decision-making than simulating the opponent's policy.

**Question 4.** *Does TAO exhibit robustness to different **opponent policy configurations**?*

Since $\Pi^{off}$ is a subset of the $\Pi^{test}$ under the *mix* setting, we can choose different opponent policies to put into $\Pi^{off}$ and set up the three test settings accordingly to evaluate the robustness of all approaches to different **opponent policy configurations**. Specifically, we establish two configurations: For the first one, *seen* contains $o0, o2, o3, o4, o6$, *unseen* contains $o1, o5, o7$; for the second one, *seen* contains $o0, o2, o3, o5, o7$, *unseen* contains $o1, o4, o6$. In different configurations, disparities are evident in the numerical outcomes across the board because of the varying strength of the opponent's policy in $\Pi^{off}$, greatly impacting the subsequent testing stage. Therefore, we focus on the relative performance between different approaches.

Table 1 shows the primary outcomes, where the results in bold denote the best among all approaches. TAO maintains its competitive performance in both environments and across different opponent policy configurations. In contrast, other approaches even fail under the new configuration. For example, Prompt-DT performs poorly in PA under configuration 1, probably because the policies of seen and unseen opponent are pretty different. Thus, Prompt-DT struggles to generalize knowledge from seen opponent policies to unseen opponent policies using only controlled agent trajectory prompts. This indicates TAO is a relatively more robust approach compared to other baselines under OOM setting.

**Additional empirical analysis.** We provide experimental analysis on performance in the offline training, adaptation speed in the testing, and ablation studies of the OCW's size $C$ in Appendix K.

## 5 CONCLUDING REMARKS

In this paper, we formalize a problem, OOM, aiming to mitigate the impracticality and inefficiency commonly associated with existing opponent modeling research. To tackle the OOM problem, we propose a general Transformer-based approach TAO, whose procedures consist of 3 stages: 1) Offline stage 1: Employ PEL to generatively and discriminatively embed opponent trajectories into latent spaces; 2) Offline stage 2: Adopt supervised pre-training to generate approximate best responses to different opponent policies; 3) Deployment stage: Accumulate information in new environments and adapt to unknown opponent policies with in-context learning. We provide theoretical analyses on TAO's equivalence to PSOM and guarantee TAO's convergence in opponent policy recognition. Extensive empirical analysis validates TAO's efficacy in addressing the OOM problems. Our results indicate that TAO responds reasonably to opponent policies even unseen before, showcasing its in-context learning capabilities. We discuss **limitations and future work** in Appendix L.

## ACKNOWLEDGMENTS

This work is supported in part by the National Science and Technology Major Project (2022ZD0116401); the Natural Science Foundation of China under Grant 62076238, Grant 62222606, and Grant 61902402; the Jiangsu Key Research and Development Plan (No. BE2023016); and the China Computer Federation (CCF)-Tencent Open Fund.

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

## A  RELATED WORK

**Opponent modeling.** Recent studies on opponent modeling based on machine learning techniques can be broadly divided into the following categories:

**1)** *Embedding-based Opponent Modeling*: Embed the opponent's policy into a latent space through representation learning methods to assist the decision-making of the controlled agent. Grover et al. (2018) employed imitation learning (Hussein et al., 2017) and contrastive learning (Jaiswal et al., 2020) to generate policy embeddings of opponent trajectories. They then combined these embeddings with RL for policy optimization. Similar efforts, such as He et al. (2016a) and Hong et al. (2018), utilized auxiliary networks for encoding hand-crafted opponent features, predicting opponent actions, and tuning the policy network for better performance. Papoudakis et al. (2021) proposed using an autoencoder to utilize the observations and actions of the controlled agent to reconstruct the observations and actions of the opponent to learn embeddings to assist decision-making. In contrast, Papoudakis & Albrecht (2020) and Zintgraf et al. (2021) adopted Variational AutoEncoders (VAE) (Kingma & Welling, 2013) to learn the high-dimensional distribution of opponent policies.

**2)** *Opponent Modeling with Online Reasoning*: Detect or infer the opponent's policy online through methods such as Bayesian and then yield responses accordingly. Bard et al. (2013) first trained the mixture-of-expert counter-strategies against a known set of fixed opponent policies. Subsequently, they employed a bandit algorithm during testing to select the most appropriate counter-strategy. Based on BPR+ (Rosman et al., 2016; Hernandez-Leal et al., 2016), Zheng et al. (2018) proposed a rectified belief model to more accurately detect opponent policies. Additionally, they introduced a distillation policy network to achieve improved outcomes. DiGiovanni & Tewari (2021) employed Thompson sampling (Thompson, 1933) and change detection to tackle opponent switching between stationary policies. Fu et al. (2022) proposed using a bandit algorithm to choose greedy or conservative policies to play against non-stationary opponent. The greedy policy was trained by VAE along with conditional RL and updated online through variational inference. The conservative policy was a fixed, robust policy. Lv et al. (2023) proposed a similar approach to exploit non-stationary opponent.

**3)** *Opponent Modeling with Evolving Opponents*: Estimate the opponent's policy and corresponding impacts considering the opponent's learning (*i.e.*, gradient update). Foerster et al. (2018a) modeled the update of the opponent's policy and estimated the learning gradient of the opponent's policy. Foerster et al. (2018b) introduced an Differentiable Monte-Carlo Estimator operation to consider how to shape the learning dynamics of other agents based on Foerster et al. (2018a)'s approach. Letcher et al. (2019) further combined stability guarantees from LookAhead with opponent-shaping abilities from Foerster et al. (2018a) to improve theoretically and experimentally. Al-Shedivat et al. (2018) and Kim et al. (2021) utilized meta-learning to adapt to evolving opponent. Kim et al. (2021) also introduced a term that closely relates to shaping the learning dynamics of other agents' policies, thereby considering the impacts of agent's current policy on future opponent policies.

**4)** *Opponent Adapting with Meta-learning*: Based on meta-learning methods, train against a set of known opponent policies to quickly adapt to unknown opponent policies at test time. While most meta-RL methods assume the tasks in training and testing share a similar distribution, this category explores the potential application of meta-RL to competing with unknown non-stationary opponent. Al-Shedivat et al. (2018) presented a gradient-based meta-learning algorithm for continuous adaptation in non-stationary and competitive environments, demonstrating its effectiveness in improving adaptation efficiency. Kim et al. (2021) proposed a novel meta multi-agent policy gradient algorithm that effectively addresses the non-stationary policy dynamics inherent in multi-agent RL. Zintgraf et al. (2021) introduced a meta-learning approach for deep interactive Bayesian RL in multi-agent settings, which leverages approximate belief inference and policy adaptation for improved opponent adapting. Wu et al. (2022) proposed a meta-learning framework, named Learning to Exploit (L2E), for implicit opponent modeling, enabling agents to quickly adapt and exploit opponent with diverse styles through a few interactions during training.

**5)** *Theory of Mind (ToM)-based Opponent Modeling*: Reason about opponent's mental states and intentions, to predict and adapt to opponent behavior by modeling their beliefs, goals, and actions. Von Der Osten et al. (2017) proposed the multi-agent ToM model to predict opponents' actions and infer strategy sophistication in stochastic games. Upon Zheng et al. (2018)'s Bayesian online opponent modeling approach, Yang et al. (2019) introduced the ToM method, which uses higher-level decision-making to play against the opponent who also performs opponent modeling.

Rabinowitz et al. (2018) and Raileanu et al. (2018) also explored methodologies for modeling an opponent's mental state. Rabinowitz et al. (2018) employed three networks to reason about agent actions and goals, simulating a human-like ToM. Raileanu et al. (2018) proposed using its own policy along with the opponent's observations and actions to learn the opponent's goals.

**6)** *Recursive Reasoning*: Predict opponent's actions by simulating multi-level nested beliefs, enabling anticipation of opponent behavior based on their expectations of the controlled agent's reasoning process. Wen et al. (2019) and Wen et al. (2021) proposed performing recursive reasoning by modeling hypothetical nested beliefs via the agent's joint Q-function. They showed improved adaptation in stochastic games. Dai et al. (2020) employed recursive reasoning by assuming agents select actions based on GP-UCB acquisition functions (Srinivas et al., 2010). It achieves faster regret convergence in repeated games. Yuan et al. (2020) proposed an algorithm using ToM-based recursive reasoning and adaptive RL to enable agents to develop a pragmatic communication protocol, allowing them to infer hidden meanings from context and improve cooperative multi-agent communication. Yu et al. (2022) introduced a model-based opponent modeling approach that uses recursive imagination in an environment model and Bayesian mixing to adapt to diverse opponents.

Most existing work assumes that the environment and opponent policies are consistently available, whereas our work focuses on opponent modeling under challenging scenarios where they are not.

**Offline meta-RL.** Offline meta-RL aims at using offline datasets to develop the agent's understanding of tasks within the training distribution and extrapolate this knowledge to novel tasks. (Li et al., 2020; Mitchell et al., 2021; Dorfman et al., 2021; Pong et al., 2022). The key principles are meta-learning, where the model learns to quickly adapt based on limited samples of a new task (Finn et al., 2017), and task information inference (Rakelly et al., 2019). Algorithmically, *in-context learning* stands within the domain of meta-learning and can be considered as taking a more agnostic approach by learning the learning algorithm itself (Duan et al., 2016; Wang et al., 2016; Mishra et al., 2018; Laskin et al., 2023). Lee et al. (2023) introduced supervised pre-training to empirically and theoretically showcase robust in-context learning abilities in decision-making. Seen through another lens, OOM can be approximately regarded as an offline meta-RL problem. When we consider the *opponent (teammate) policy* in the field of opponent modeling as a *task*, we can roughly regard OOM as an offline meta-RL problem. In precise terms, OOM poses more complexity than offline meta-RL, and we will discuss the relationship between the two in Section 2.2.

**Transformers for decision-making.** There has been growing interest in applying the Transformers to decision-making by formulating the problem as sequence modeling (Yang et al., 2023a; Li et al., 2023). Chen et al. (2021) proposed Decision Transformer (DT), which models action sequences conditioned on returns with a causal Transformer trained on offline data. Follow-up works studied enhancements like improved conditioning (Furuta et al., 2022; Paster et al., 2022) and architectural innovations (Villaflor et al., 2022). Another exciting direction leverages the generality and scalability of Transformers for multi-task learning (Lee et al., 2022; Reed et al., 2022). Transformers for decision-making also exhibited meta-learning capabilities (Melo, 2022). Recently, Lee et al. (2023) proposed a Transformer-based in-context learning approach that empirically and theoretically improves over behaviors seen in the dataset in terms of regret where DT cannot (Brandfonbrener et al., 2022; Yang et al., 2023b). Building upon these insights, we design a Transformer-based approach to fully leverage its in-context learning abilities to tackle the OOM problem.

**Offline multi-agent RL (MARL).** Offline MARL has emerged as an important subfield of MARL in recent years (Cui & Du, 2022; Tian et al., 2023; Wu et al., 2023; Jiang & Lu, 2023; Formanek et al., 2023). This line of research aims to learn effective policies purely from pre-collected datasets without any further multi-agent environmental interactions. The offline setting avoids costly or dangerous online exploration, making MARL more practical for real-world applications.

A key challenge in offline MARL is handling the distribution shift between the offline dataset and the true environment dynamics at test time (Jiang & Lu, 2021). When opponents employ policies not seen in the offline data, the performance of policies trained purely offline can degrade significantly. Recent works have proposed various techniques to address this issue. For example, Pan et al. (2022) introduced Offline MARL with Actor Rectification (OMAR) that uses zeroth-order optimization to update the actor based on the offline critic conservatively. Tseng et al. (2022) proposed distilling knowledge from a teacher Transformer trained with global information into decentralized student

policies to enable effective decentralized execution at test time. The Transformer architecture has also been increasingly adopted for offline MARL. For instance, the Multi-Agent Decision Transformer (MADT) (Meng et al., 2023) treats MARL as a sequence modeling problem and predicts actions in an autoregressive manner.

Offline MARL and OOM are aligned in their pursuit of extracting effective policies from offline datasets in multi-agent settings and utilizing them in testing environments to boost efficiency and bridge the gap with real-world scenarios. Nevertheless, most offline MARL approaches focus on cooperative settings, while in this work, we concentrate on OOM in competitive environments. Furthermore, whether explicitly or implicitly, OOM must introduce modeling of other agents in the environment for adaptability, while offline MARL does not emphasize such a requirement.

## B  DETAILED ASSUMPTIONS OF OOM

**Environment types to which OOM is applicable.** While our paper focuses on OOM within competitive settings, it's worth mentioning that the extension of OOM to cooperative and mixed environments where competition and cooperation are intertwined is straightforward. In cooperative and mixed environments, the corresponding OOM problem can be formalized by instantiating the opponent as other agents that are desired to be modeled. Their optimization objectives have the same form as Eq. (1) to learn an modeled agent-aware adaptive controlled agent policy.

**Number of controlled agent and opponent.** According to our formalization of the OOM problem, there is no limit on the number of controlled agent and opponent in the environment. Yet, some assumptions of the OOM problem need to be emphasized again: 1) The trajectories of all controlled agents in the offline dataset are generated by an approximately best response policy against a fixed opponent policy; 2) All opponent agents adopt the same fixed opponent policy.

**Availability of the environment.** During offline training, we hypothesize no interaction with the environment and the opponent. In another word, opponent policies in the *offline opponent policy set* $\Pi^{\text{off}}$ cannot be called explicitly for learning. It is only possible to sample the trajectories $\mathcal{T}^{-1,k}$ produced by each opponent policy from $\mathcal{D}^{\text{off}}$. Yet, for comparing the learning efficacy across all approaches, we permit a small number of interactions per set interval of checkpoints for evaluation. During testing (*i.e.*, deployment), interaction with the new environment is allowed, where the opponent adopts policies from the *test opponent policy set* $\Pi^{\text{test}}$.

**Specification of opponent policy.** All opponent policies in $\Pi$ are prohibited from learning, and there is no policy switching at the granularity of a timestep. In our experiments, we employ three distinct test settings for the opponent policy set $\Pi^{\text{test}}$: *seen*, equivalent to $\Pi^{\text{off}}$; *unseen*, encompassing previously unseen opponent policies; and *mix*, which combines elements of both. Our study includes diverse opponent policies that are independent of one another and possess varying degrees of strength. See the details of the opponent policies in Appendix H.

**Accessibility of opponent information.** During offline training stages, only limited opponent information can be gleaned from the provided $\mathcal{D}^{\text{off}}$, and no extra information can be sourced. However, during testing, collection of the opponent's trajectory $\tau^{-1}$ after each episode is permissible, enabling all approaches to adapt to unknown opponent policies based on their own mechanism.

## C    PSEUDOCODE OF TAO

---

**Algorithm 1** Transformer Against Opponent (TAO)

---

1: /* Offline Stage 1: Policy Embedding Learning */
2: Initialize model $M_{\theta_e}$ with parameters $\theta_e$, and set the batch size $B$.
3: **while** not converged **do**
4:     Initialize empty batches $\mathcal{B}_{\text{gen}} = \{\}, \mathcal{B}_{\text{dis}} = \{\}$.
5:     **for** $k$ in $[K]$ **do**
6:         Sample $B$ trajectory tuples $(\tau_i^{-1,k}, \tau_j^{-1,k}) \sim \mathcal{T}^{-1,k}$, and add these tuples into $\mathcal{B}_{\text{gen}}$.
7:     **end for**
8:     Sample $B$ trajectories $\tau^{-1} \sim \mathcal{T}^{-1}$ to form $\mathcal{B}_{\text{dis}}$.
9:     Calculate $\mathcal{L}_{\text{gen}}$ in Eq. (2) using $\mathcal{B}_{\text{gen}}$, and calculate $\mathcal{L}_{\text{dis}}$ in Eq. (3) using $\mathcal{B}_{\text{dis}}$.
10:     Calculate $\mathcal{L}_{\text{emb}}$ in Eq. (4), and backpropagate to update $\theta_e$.
11: **end while**
12: /* Offline Stage 2: Opponent-aware Response Policy Training */
13: Initialize model $M_{\theta_d}$ with parameters $\theta_d$, and set the OCW size $C$.
14: **while** not converged **do**
15:     Initialize empty batches $\mathcal{B}_{\text{res}} = \{\}$.
16:     **for** $k$ in $[K]$ **do**
17:         Sample $B$ trajectory tuples $(\tau^{1,k}, \{\tau_c^{-1,k}\}_{c=1,\dots,C})$, where $\tau^{1,k} \sim \mathcal{T}^{1,k}, \{\tau_c^{-1,k}\}_{c=1,\dots,C}$
             $\sim \mathcal{T}^{-1,k}$, and add these tuples into $\mathcal{B}_{\text{res}}$.
18:     **end for**
19:     Calculate $\mathcal{L}_{\text{res}}$ in Eq. (5) using $\mathcal{B}_{\text{res}}$, and backpropagate to update $\theta_d$.
20: **end while**
21: /* Deployment Stage: In-context Opponent Adapting */
22: Initialize empty OCW $\mathcal{W} = \{\}$, and set the opponent policy switching interval $E$.
23: **for** op in max_opponent_number **do**
24:     Sample unknown opponent policy $\pi^{-1} \sim \Pi^{\text{test}}$.
25:     **for** ep in $[E]$ **do**
26:         Deploy $M_\theta$ by sampling $a_t^1$ via Eq. (6) at each timestep $t$ in this ep, where $\theta = \{\theta_e, \theta_d\}$.
27:         Add $\tau^{-1} = (o_0^{-1}, a_0^{-1}, r_0^{-1}, \dots)$ into $\mathcal{W}$, and clip $\mathcal{W}$ to contain the nearest $C$ trajectories.
28:     **end for**
29: **end for**

---

# D  DETAILED THEORETICAL ANALYSIS

## D.1  ALGORITHM OF POSTERIOR SAMPLING IN OPPONENT MODELING

We instantiate a Bayesian posterior sampling algorithm in the context of opponent modeling, referred to as Posterior Sampling in Opponent Modeling (PSOM). In the PSOM algorithm, we use opponent trajectory consisting of consecutive $(o^{-1}, a^{-1})$ tuples as in-context data and add them to the OCW $\mathcal{W}$. Following up, we describe the PSOM algorithm in a most general way (Osband et al., 2013; Lee et al., 2023).

Given the initial distribution of opponent policies $\Pi_0 \leftarrow \Pi_{\text{pre}}$, where $\Pi_{\text{pre}}$ is the *probability distribution on* $\Pi^{\text{off}}$, and an empty OCW $\mathcal{W} = \{\}$, for $c \in [C]$:

1.  Sample the opponent policy $\pi_c^{-1}$ by $\Pi_c$ and compute the approximate best response policy $\pi_c^{1,*}$ for the controlled agent;

2.  Interact with the *true opponent policy* $\bar{\pi}^{-1}$ using $\pi_c^{1,*}$ and add the opponent trajectory to the $\mathcal{W}$.

3.  Update the posterior distribution $\Pi_c(\pi^{-1}) = P(\pi^{-1}|\mathcal{W})$.

## D.2  PROOF OF THEOREM 3.1

**Theorem 4.1** (Equivalence of TAO and PSOM). *Assume that the learned model $M_\theta$ is consistent and the sampling of $o^{-1}$ from $\mathcal{T}_{pre}^{-1}$ is independent of the opponent policies, then given $\bar{\pi}^{-1}$ and its $\mathcal{W}$, we have $P(\xi_H^1|\mathcal{W}, \bar{\pi}^{-1}; PSOM) = P(\xi_H^1|\mathcal{W}, \bar{\pi}^{-1}; M_\theta)$ for all possible $\xi_H^1$.*

*Proof.* Differing from the main text, in this section, we use $\pi^{-1}$ to denote opponent policies posteriorly sampled from $\Pi_{\text{pre}}$, where $\Pi_{\text{pre}}$ is the *probability distribution on* $\Pi^{\text{off}}$, and use $\bar{\pi}^{-1}$ to denote the true opponent policy interacted with during the deployment stage of TAO. For clarity and ease of understanding, all trajectory sequences are indexed starting from 1 in this section (originally starting from 0 in the main text). Define $\xi_H^1 = (o_1^1, a_1^{1,*}, \ldots, o_H^1, a_H^{1,*})$ as *controlled agent's history*, where $H$ denotes the maximum length of this history and $a^{1,*}$ is sampled from the approximate best response policy $\pi^{1,*}$ against the posteriorly sampled $\pi^{-1}$. $\mathcal{T}_{\text{pre}}^{-1}(\cdot; \pi^{-1})$ denotes the *probability distribution on the trajectories set* $\mathcal{T}^{-1}$ of $\pi^{-1}$ during TAO's offline stage 2.

Let $\pi^{-1} \sim \Pi_{\text{pre}}$ and OCW $\mathcal{W}$ contain sampled trajectory fragments of $\pi^{-1}$ and let $o_{\text{query}}^1 \in \mathcal{O}^1, a^{1,*} \in \mathcal{A}^1, \xi_{H-1}^1 \in (\mathcal{O}^1 \times \mathcal{A}^1)^{H-1}$ and $h \in [0, H-1]$ be arbitrary, the full joint probability distribution during TAO's offline stage 2 can be denoted as:

$$P_{\text{pre}}(\pi^{-1}, \mathcal{W}, \xi_{H-1}^1, h, o_{\text{query}}^1, a^{1,*}) = \Pi_{\text{pre}}(\pi^{-1})\mathcal{T}_{\text{pre}}^{-1}(\mathcal{W}; \pi^{-1})\mathfrak{D}_H(o_{1:H}^1)$$
$$(\prod_{i \in [H]} \pi^{1,*}(a_i^1|o_i^1; \pi^{-1})) \times \text{Unif}[0, H-1]\mathfrak{D}_{\text{query}}(o_{\text{query}}^1)\pi^{1,*}(a^{1,*}|o_{\text{query}}^1; \pi^{-1}). \quad (7)$$

Herein, $\mathfrak{D}_H \in \Delta((\mathcal{O}^1)^H)$, which is independent of the opponent policies. $\mathfrak{D}_{\text{query}}$ is set to the uniform over $\mathcal{O}^1$. Moreover, we do small modifications to the original TAO by sampling $h \sim \text{Unif}[0, H-1]$ and truncating $\xi_h^1$ from $\xi_{H-1}^1$.

We define the random sequences and subsequences of 1) the controlled agent's *observed trajectory* and 2) the controlled agent's *true trajectory* **under PSOM algorithm**, respectively, as follows:

1)  $\Xi_{\text{PSOM}}^1(h; \mathcal{W}) = (O_1^{1,\text{PSOM}}, A_1^{1,\text{PSOM}}, \ldots, O_h^{1,\text{PSOM}}, A_h^{1,\text{PSOM}})$;

2)  $\Xi_{\text{PSOM}}(h; \mathcal{W}) = (S_1^{\text{PSOM}}, A_1^{1,\text{PSOM}}, \ldots, S_h^{\text{PSOM}}, A_h^{1,\text{PSOM}})$.

These two types of trajectories are generated in the following manner:

$$\pi_{\text{PSOM}}^{-1} \sim P(\pi^{-1}|\mathcal{W}), O_1^{1,\text{PSOM}} \sim P(O_1^{1,\text{PSOM}}|S_1^{\text{PSOM}}), S_1^{\text{PSOM}} \sim \rho,$$
$$A_i^{1,\text{PSOM}} \sim \pi^{1,*}(\cdot|O_i^{1,\text{PSOM}}; \pi_{\text{PSOM}}^{-1}), A_i^{-1,\text{PSOM}} \sim \bar{\pi}^{-1}(\cdot|O_i^{-1,\text{PSOM}}), i \geq 1,$$
$$O_{i+1}^{1,\text{PSOM}} \sim P(\cdot|S_{i+1}^{\text{PSOM}}), S_{i+1}^{\text{PSOM}} \sim T(\cdot|S_i^{\text{PSOM}}, A_i^{1,\text{PSOM}}, A_i^{-1,\text{PSOM}}), i \geq 2.$$

Within, $\rho$ denotes the initial distribution on $\mathcal{S}$. Analogously, we define the random sequences and subsequences of 1) the controlled agent's *observed trajectory* and 2) the controlled agent's *true trajectory* **under TAO algorithm**, respectively, as follows:

1) $\Xi_{\text{pre}}^1(h; \mathcal{W}) = (O_1^{1,\text{pre}}, A_1^{1,\text{pre}}, \ldots, O_h^{1,\text{pre}}, A_h^{1,\text{pre}})$;

2) $\Xi_{\text{pre}}(h; \mathcal{W}) = (S_1^{\text{pre}}, A_1^{1,\text{pre}}, \ldots, S_h^{\text{pre}}, A_h^{1,\text{pre}})$.

These two types of trajectories are generated in the following manner:

$$O_1^{1,\text{pre}} \sim P(O_1^{1,\text{pre}}|S_1^{\text{pre}}), S_1^{\text{pre}} \sim \rho,$$
$$A_i^{1,\text{pre}} \sim P_{\text{pre}}(\cdot|O_i^{1,\text{pre}}, \mathcal{W}, \Xi_{\text{pre}}^1(i-1; \mathcal{W})), A_i^{-1,\text{pre}} \sim \bar{\pi}^{-1}(\cdot|O_i^{-1,\text{pre}}), i \geq 1,$$
$$O_{i+1}^{1,\text{pre}} \sim P(\cdot|S_{i+1}^{\text{pre}}), S_{i+1}^{\text{pre}} \sim T(\cdot|S_i^{\text{pre}}, A_i^{1,\text{pre}}, A_i^{-1,\text{pre}}), i \geq 2.$$

To simplify matters, we will refrain from explicitly referencing $\mathcal{W}$ for $\Xi^1$ and $\Xi$ in our notations, except when required to avoid confusion. Next, we introduce a common assumption to ensure the neural network fits the pre-training data distribution.

**Assumption D.1** (Learned model is consistent). *Let $M_\theta$ denote the pre-trained model derived by TAO. For any given $(O_i^{1,\text{pre}}, \mathcal{W}, \Xi_{pre}^1(h-1; \mathcal{W}))$, $P_{pre}(A_i^{1,\text{pre}}|O_i^{1,\text{pre}}, \mathcal{W}, \Xi_{pre}^1(h-1; \mathcal{W})) = M_\theta(A_i^{1,\text{pre}}|O_i^{1,\text{pre}}, \mathcal{W}, \Xi_{pre}^1(h-1; \mathcal{W}))$ for all possible $A_i^{1,\text{pre}}$.*

Upon Assumption D.1, we will limit our attention to $P_{\text{pre}}$ for the rest of the proof.

To prove that $\forall \xi_H^1, P(\xi_H^1|\mathcal{W}, \bar{\pi}^{-1}; \text{PSOM}) = P(\xi_H^1|\mathcal{W}, \bar{\pi}^{-1}; M_\theta)$ (*i.e.*, Theorem 3.1), it is equivalent to prove that

$$P(\Xi_{\text{PSOM}}^1(H) = \xi_H^1) = P(\Xi_{\text{pre}}^1(H) = \xi_H^1). \tag{8}$$

We will prove that $\forall h \in [H]$,

$$P(\Xi_{\text{PSOM}}(h) = \xi_h'^1) = P(\Xi_{\text{pre}}(h) = \xi_h'^1) \tag{9}$$

using **Mathematical Induction** and then introduce two lemmas for the evidence of Eq. (8). Here, $\xi_h'^1$ is *controlled agent's true history*, and it can be obtained by replacing all the $o$ in $\xi_h^1$ to $s$.

First, we propose a lemma to assist in proving Eq. (9) for the base case when $h = 1$.

**Lemma D.2.** *If the sampling of $o^{-1}$ from $\mathcal{T}_{pre}^{-1}$ is independent of the opponent policy, then $P_{pre}(\pi^{-1}|\mathcal{W}) = P(\pi_{PSOM}^{-1} = \pi^{-1}|\mathcal{W})$.*

*Proof.* Assuming the sampling of $o^{-1}$ from $\mathcal{T}_{\text{pre}}^{-1}$ is independent of the opponent policy, we have:

$$P(\pi_{\text{PSOM}}^{-1} = \pi^{-1}|\mathcal{W}) \propto \Pi_{\text{pre}}(\pi^{-1})P(\mathcal{W}|\pi^{-1}) \tag{10a}$$

$$\propto \Pi_{\text{pre}}(\pi^{-1}) \prod_{j \in [|\mathcal{W}|]} \pi^{-1}(a_j^{-1}|o_j^{-1}) \tag{10b}$$

$$\propto \Pi_{\text{pre}}(\pi^{-1}) \prod_{j \in [|\mathcal{W}|]} \pi^{-1}(a_j^{-1}|o_j^{-1})\mathcal{T}_{\text{pre}}^{-1}(o_j^{-1}) \tag{10c}$$

$$= \Pi_{\text{pre}}(\pi^{-1})\mathcal{T}_{\text{pre}}^{-1}(\mathcal{W}; \pi^{-1}) \tag{10d}$$

$$\propto P_{\text{pre}}(\pi^{-1}|\mathcal{W}). \tag{10e}$$

$\propto$ denotes that the two sides are equal up to multiplicative factors independent of $\pi^{-1}$. Eq. (10a) is derived through the Bayesian posterior probability formula. Eq. (10b) uses the fact that $o^{-1}$ in posterior sampling is independent of the opponent policies. Eq. (10c) holds because of the assumption that the sampling of $o^{-1}$ from $\mathcal{T}_{\text{pre}}^{-1}$ is independent of the opponent policies. Eq. (10d) uses the definition of $\mathcal{T}_{\text{pre}}^{-1}$. Eq. (10e) is derived through the Bayesian posterior probability formula. $\square$

Now, we prove that Eq. (9) holds when $h = 1$:

$$P(\Xi_{\text{PSOM}}(1) = \xi_1'^1) = P(S_1^{\text{PSOM}} = s_1, A_1^{1,\text{PSOM}} = a_1^1) \tag{11a}$$

$$= \rho(s_1) \int_{o_1^1} P(A_1^{1,\text{PSOM}} = a_1^1 | O_1^{1,\text{PSOM}} = o_1^1) P(o_1^1 | s_1) \mathrm{d}o_1^1 \tag{11b}$$

$$= \rho(s_1) \int_{\pi^{-1}, o_1^1} P(A_1^{1,\text{PSOM}} = a_1^1, \pi_{\text{PSOM}}^{-1} = \pi^{-1} | O_1^{1,\text{PSOM}} = o_1^1) P(o_1^1 | s_1) \mathrm{d}\pi^{-1} \mathrm{d}o_1^1 \tag{11c}$$

$$= \rho(s_1) \int_{\pi^{-1}, o_1^1} \pi^{1,*}(a_1^1 | o_1^1; \pi^{-1}) P_{\text{PSOM}}(\pi_{\text{PSOM}}^{-1} = \pi^{-1} | \mathcal{W}, O_1^{1,\text{PSOM}} = o_1^1) P(o_1^1 | s_1) \mathrm{d}\pi^{-1} \mathrm{d}o_1^1 \tag{11d}$$

$$= \rho(s_1) \int_{\pi^{-1}, o_1^1} \pi^{1,*}(a_1^1 | o_1^1; \pi^{-1}) P_{\text{PSOM}}(\pi_{\text{PSOM}}^{-1} = \pi^{-1} | \mathcal{W}) P(o_1^1 | s_1) \mathrm{d}\pi^{-1} \mathrm{d}o_1^1 \tag{11e}$$

$$= \rho(s_1) \int_{\pi^{-1}, o_1^1} \pi^{1,*}(a_1^1 | o_1^1; \pi^{-1}) P_{\text{pre}}(\pi^{-1} | \mathcal{W}) P(o_1^1 | s_1) \mathrm{d}\pi^{-1} \mathrm{d}o_1^1 \tag{11f}$$

$$= \rho(s_1) \int_{o_1^1} P_{\text{pre}}(a_1^1 | o_1^1, \mathcal{W}) P(o_1^1 | s_1) \mathrm{d}o_1^1 \tag{11g}$$

$$= P(\Xi_{\text{pre}}(1) = \xi_1'^1). \tag{11h}$$

Eqs. (11a) to (11d), (11g) and (11h) are derived using Bayesian Law of Total Probability and conditional probability formula based on Eq. (7). Eq. (11e) holds because the sampling of $o_1^1$ is independent of $\pi^{-1}$. Eq. (11f) is derived through Lemma D.2.

Now, we start proving Eq. (9) for the other cases when $h \neq 1$. Before starting, we introduce another lemma to help finish this proving process as well as the proof of Eq. (8).

**Lemma D.3.** *Given any $\xi_i'^1$, if $P(\Xi_{PSOM}(i) = \xi_i'^1) = P(\Xi_{pre}(i) = \xi_i'^1)$, then $P(\Xi_{PSOM}^1(i) = \xi_i^1) = P(\Xi_{pre}^1(i) = \xi_i^1)$.*

*Proof.*

$$P(\Xi_{\text{PSOM}}^1(i) = \xi_i^1) = \int_{\xi_i'^1} P(\Xi_{\text{PSOM}}(i) = \xi_i'^1) P(\xi_i^1 | \xi_i'^1) \mathrm{d}\xi_i'^1 \tag{12a}$$

$$= \int_{\xi_i'^1} P(\Xi_{\text{pre}}(i) = \xi_i'^1) P(\xi_i^1 | \xi_i'^1) \mathrm{d}\xi_i'^1 \tag{12b}$$

$$= P(\Xi_{\text{pre}}^1(i) = \xi_i^1). \tag{12c}$$

$\square$

Based on Eq. (11), we have $P(\Xi_{\text{PSOM}}^1(1) = \xi_1^1) = P(\Xi_{\text{pre}}^1(1) = \xi_1^1)$.

Now, we utilize the inductive hypothesis to demonstrate the validity of the entire statement. Suppose that $P(\Xi_{\text{PSOM}}(h-1) = \xi_{h-1}'^1) = P(\Xi_{\text{pre}}(h-1) = \xi_{h-1}'^1)$, according to Lemma D.3, we can conclude that $P(\Xi_{\text{PSOM}}^1(h-1) = \xi_{h-1}^1) = P(\Xi_{\text{pre}}^1(h-1) = \xi_{h-1}^1)$.

Since

$$P(\Xi_{\text{PSOM}}(h) = \xi_h'^1) =$$
$$P(\Xi_{\text{PSOM}}(h-1) = \xi_{h-1}'^1) P(S_h^{\text{PSOM}} = s_h, A_h^{1,\text{PSOM}} = a_h^1 | \Xi_{\text{PSOM}}(h-1) = \xi_{h-1}'^1)$$

and

$$P(\Xi_{\text{pre}}(h) = \xi_h'^1) =$$
$$P(\Xi_{\text{pre}}(h-1) = \xi_{h-1}'^1) P(S_h^{\text{pre}} = s_h, A_h^{1,\text{pre}} = a_h^1 | \Xi_{\text{pre}}(h-1) = \xi_{h-1}'^1),$$

to prove that $P(\Xi_{\text{PSOM}}(h) = \xi_h'^1) = P(\Xi_{\text{pre}}(h) = \xi_h'^1)$, it is equivalent to prove:

$$\begin{aligned} P(S_h^{\text{PSOM}} = s_h, A_h^{1,\text{PSOM}} = a_h^1 | \Xi_{\text{PSOM}}(h-1) = \xi_{h-1}'^1) \\ = P(S_h^{\text{pre}} = s_h, A_h^{1,\text{pre}} = a_h^1 | \Xi_{\text{pre}}(h-1) = \xi_{h-1}'^1). \end{aligned} \tag{13}$$

By expanding the formula, we can get:

$$P(S_h^{\text{PSOM}} = s_h, A_h^{1,\text{PSOM}} = a_h^1 | \Xi_{\text{PSOM}}(h-1) = \xi_{h-1}'^1)$$

$$= T(s_h | s_{h-1}, a_{h-1}^1; \bar{\pi}^{-1}) P(A_h^{1,\text{PSOM}} = a_h^1 | S_h^{\text{PSOM}} = s_h, \Xi_{\text{PSOM}}(h-1) = \xi_{h-1}'^1) \quad (14a)$$

$$= \int_{a_{h-1}^{-1}, o_{h-1}^{-1}} T(s_h | s_{h-1}, a_{h-1}^1, a_{h-1}^{-1}) \bar{\pi}^{-1}(a_{h-1}^{-1} | o_{h-1}^{-1}) P(o_{h-1}^{-1} | s_{h-1}) \mathrm{d}a_{h-1}^{-1} \mathrm{d}o_{h-1}^{-1} \cdot$$

$$\int_{\pi^{-1}, o_h^1, \xi_{h-1}^1} P(A_h^{1,\text{PSOM}} = a_h^1, \pi_{\text{PSOM}}^{-1} = \pi^{-1} | O_h^{1,\text{PSOM}} = o_h^1, \Xi_{\text{PSOM}}^1(h-1) = \xi_{h-1}^1)$$

$$P(\xi_{h-1}^1 | \xi_{h-1}'^1) P(o_h^1 | s_h) \mathrm{d}\pi^{-1} \mathrm{d}o_h^1 \mathrm{d}\xi_{h-1}^1. \quad (14b)$$

In Eq. (14b), the first integral term is independent of the algorithm choice, while the terms in the second integral term satisfies:

$$P(A_h^{1,\text{PSOM}} = a_h^1, \pi_{\text{PSOM}}^{-1} = \pi^{-1} | O_h^{1,\text{PSOM}} = o_h^1, \Xi_{\text{PSOM}}^1(h-1) = \xi_{h-1}^1)$$

$$= \pi^{1,*}(a_h^1 | o_h^1; \pi^{-1}) P(\pi_{\text{PSOM}}^{-1} = \pi^{-1} | O_h^{1,\text{PSOM}} = o_h^1, \Xi_{\text{PSOM}}^1(h-1) = \xi_{h-1}^1). \quad (15)$$

Based on Eq. (14) and Eq. (15), to prove that Eq. (13) holds, it is equivalent to demonstrate:

$$P(\pi_{\text{PSOM}}^{-1} = \pi^{-1} | O_h^{1,\text{PSOM}} = o_h^1, \Xi_{\text{PSOM}}^1(h-1) = \xi_{h-1}^1) = P_{\text{pre}}(\pi^{-1} | o_h^1, \mathcal{W}, \xi_{h-1}^1). \quad (16)$$

We prove that Eq. (16) holds through the following analysis:

$$P(\pi_{\text{PSOM}}^{-1} = \pi^{-1} | O_h^{1,\text{PSOM}} = o_h^1, \Xi_{\text{PSOM}}^1(h-1) = \xi_{h-1}^1)$$

$$= \frac{P(O_h^{1,\text{PSOM}} = o_h^1, \Xi_{\text{PSOM}}^1(h-1) = \xi_{h-1}^1 | \pi_{\text{PSOM}}^{-1} = \pi^{-1}) P(\pi_{\text{PSOM}}^{-1} = \pi^{-1} | \mathcal{W})}{P(O_h^{1,\text{PSOM}} = o_h^1, \Xi_{\text{PSOM}}^1(h-1) = \xi_{h-1}^1)} \quad (17a)$$

$$\propto P_{\text{pre}}(\pi^{-1} | \mathcal{W}) \prod_{i \in [h-1]} T(o_{i+1}^1 | \xi_i^1, \bar{\pi}^{-1}) \pi^{1,*}(a_i^1 | o_i^1; \pi^{-1}) \quad (17b)$$

$$\propto P_{\text{pre}}(\pi^{-1} | \mathcal{W}) \prod_{i \in [h-1]} \pi^{1,*}(a_i^1 | o_i^1; \pi^{-1}) \quad (17c)$$

$$\propto P_{\text{pre}}(\pi^{-1} | \mathcal{W}) \mathfrak{D}_{\text{query}}(o_h^1) \mathfrak{D}_{h-1}(o_{1:h-1}^1) \prod_{i \in [h-1]} \pi^{1,*}(a_i^1 | o_i^1; \pi^{-1}) \quad (17d)$$

$$\propto P_{\text{pre}}(\pi^{-1} | o_h^1, \mathcal{W}, \xi_{h-1}^1). \quad (17e)$$

Eq. (17a) is derived through the Bayesian posterior probability formula. In Eq. (17b), we decompose the probability of observing that sequence of observations $o^1$ and actions $a^1$ based on $P(\Xi_{\text{PSOM}}^1(h-1) = \xi_{h-1}^1) = P(\Xi_{\text{pre}}^1(h-1) = \xi_{h-1}^1)$. Eqs. (17c) and (17d) use the fact that the sampling of $o^1$ is only related to the true opponent policy $\bar{\pi}^{-1}$ and is independent of $\pi^{-1}$. Eq. (17e) is derived by the definition of $P_{\text{pre}}(\pi^{-1} | o_h^1, \mathcal{W}, \xi_{h-1}^1)$.

Therefore, we finish the proof of $P(\Xi_{\text{PSOM}}(h) = \xi_h'^1) = P(\Xi_{\text{pre}}(h) = \xi_h'^1)$, where

$$P(\Xi_{\text{PSOM}}(h) = \xi_h'^1)$$

$$= P(\Xi_{\text{pre}}(h-1) = \xi_{h-1}'^1) T(s_h | s_{h-1}, a_{h-1}^1; \bar{\pi}^{-1})$$

$$\int_{\pi^{-1}, o_h^1, \xi_{h-1}^1} \pi^{1,*}(a_h^1 | o_h^1; \pi^{-1}) P_{\text{pre}}(\pi^{-1} | o_h^1, \mathcal{W}, \xi_{h-1}^1) P(\xi_{h-1}^1 | \xi_{h-1}'^1) P(o_h^1 | s_h) \mathrm{d}\pi^{-1} \mathrm{d}o_h^1 \mathrm{d}\xi_{h-1}^1$$

$$\quad (18a)$$

$$= P(\Xi_{\text{pre}}(h-1) = \xi_{h-1}'^1) T(s_h | s_{h-1}, a_{h-1}^1; \bar{\pi}^{-1})$$

$$\int_{o_h^1, \xi_{h-1}^1} P_{\text{pre}}(a_h^1 | o_h^1, \mathcal{W}, \xi_{h-1}^1) P(\xi_{h-1}^1 | \xi_{h-1}'^1) P(o_h^1 | s_h) \mathrm{d}o_h^1 \mathrm{d}\xi_{h-1}^1 \quad (18b)$$

$$= P(\Xi_{\text{pre}}(h) = \xi_h'^1). \quad (18c)$$

Thus, based on Mathematical Induction, Eq. (9) holds for any $h \in [H]$. According to Lemma D.3, we can conclude $\forall h \in [H], P(\Xi_{\text{PSOM}}^1(h) = \xi_h^1) = P(\Xi_{\text{pre}}^1(h) = \xi_h^1)$. Hence, Eq. (8) is satisfied. This concludes the proof. $\qquad\square$

### D.3 Proof of Theorem 3.2

**Theorem 4.2.** *When the true opponent policy $\bar{\pi}^{-1}$ originates from $\Pi^{off}$, the PSOM algorithm using $(o^{-1}, a^{-1})$ tuples as in-context data guarantees to converge, and it converges to the optimal solution if the original PS using $(o^1, a^1, o'^1, r^1)$ tuples as in-context data converges to the optimal solution.*

*Proof.* In this section, we denote the OCW consisting of $(o^{-1}, a^{-1})$ tuples to contain in-context data as $\mathcal{W}$, and the OCW consisting of $(o^1, a^1, o'^1, r^1)$ tuples to contain in-context data as $\mathcal{W}'$, where $o'$ is the next observation of $o$ transitioned to. Note that we use $(o^{-1}, a^{-1}, r^{-1})$ tuples as in-context data in the original TAO, which provides more information than using $(o^{-1}, a^{-1})$ tuples. Thus, the guarantees analyzed here apply to the original TAO.

To begin, we propose a lemma and its corollary to demonstrate the convergence guarantee of the PSOM algorithm and analyze the properties of the opponent policy it converges to.

**Lemma D.4.** *Let* $\pi_0^{-1} = \arg\min_{\pi^{-1}\in\Pi^{train}} f(\pi^{-1}; \mathcal{W}) = \arg\min_{\pi^{-1}\in\Pi^{train}} -\int_{o^{-1},a^{-1}} P_{\mathcal{W}}(o^{-1}, a^{-1}) \log(P(a^{-1}|o^{-1}, \pi^{-1})) do^{-1} da^{-1}$, *then* $\forall f(\pi^{-1}; \mathcal{W}) \neq f(\pi_0^{-1}; \mathcal{W})$, $\frac{P(\pi^{-1}|(o^{-1},a^{-1})_{1:n})}{P(\pi_0^{-1}|(o^{-1},a^{-1})_{1:n})} \xrightarrow{P} 0$.

*Proof.* Here, $\pi_0^{-1}$ denotes the equivalent class of opponent policies to which the posterior probability converges with non-zero probability. $P_{\mathcal{W}}$ is the distribution of $(o^{-1}, a^{-1})$ samples in $\mathcal{W}$. $n$ is the length of the consecutive $(o^{-1}, a^{-1})$ sequences. To prove that $\frac{P(\pi^{-1}|(o^{-1},a^{-1})_{1:n})}{P(\pi_0^{-1}|(o^{-1},a^{-1})_{1:n})} \xrightarrow{P} 0$ under the given conditions, it is equivalent to prove:

$$L_{\pi^{-1},n} = -\log\frac{P(\pi^{-1}|(o^{-1},a^{-1})_{1:n})}{P(\pi_0^{-1}|(o^{-1},a^{-1})_{1:n})} \xrightarrow{P} +\infty. \tag{19}$$

By expanding the formula, we can get:

$$L_{\pi^{-1},n} = -\log\frac{P(\pi^{-1}|(o^{-1},a^{-1})_{1:n})}{P(\pi_0^{-1}|(o^{-1},a^{-1})_{1:n})} = -\log\frac{P(\pi^{-1})}{P(\pi_0^{-1})} - \sum_{i=1}^{n}\log(\frac{P(a^{-1}|\pi^{-1},o^{-1})}{P(a^{-1}|\pi_0^{-1},o^{-1})}).$$

According to the definition of $\pi_0^{-1}$ and the condition $f(\pi^{-1}; \mathcal{W}) \neq f(\pi_0^{-1}; \mathcal{W})$, we have

$$\mathbb{E}_{(o^{-1},a^{-1})\sim P_{\mathcal{W}}}[-\sum_{i=1}^{n}\log(\frac{P(a^{-1}|\pi^{-1},o^{-1})}{P(a^{-1}|\pi_0^{-1},o^{-1})})] = f(\pi^{-1}; \mathcal{W}) - f(\pi_0^{-1}; \mathcal{W}) = \mathcal{C} > 0.$$

Here, $\mathcal{C}$ is a positive constant. Therefore, based on Law of Large Numbers, we have $\lim_{n\to\infty} P(|\frac{L_{\pi^{-1},n}}{n} - \mathcal{C}| > \epsilon) = 0$, where $\epsilon$ is any positive number. Hence, Eq. (19) is satisfied and the proof ends. □

**Corollary D.5** (Corollary of Lemma D.4). *When $P_{\mathcal{W}}(o^{-1}, a^{-1}) = P_{\mathcal{W}}(o^{-1})P(a^{-1}|o^{-1}, \bar{\pi}^{-1})$, which means $\bar{\pi}^{-1}$ is originated from $\Pi^{off}$, then $\bar{\pi}^{-1} \in \pi_0^{-1}$.*

*Proof.* Since $P_{\mathcal{W}}(o^{-1}, a^{-1}) = P_{\mathcal{W}}(o^{-1})P(a^{-1}|o^{-1}, \bar{\pi}^{-1})$ holds, we have

$$\pi_0^{-1} = \arg\min_{\pi^{-1}\in\Pi^{train}} -\int_{o^{-1},a^{-1}} P_{\mathcal{W}}(o^{-1}, a^{-1})\log(P(a^{-1}|o^{-1}, \pi^{-1}))do^{-1}da^{-1} \tag{20a}$$

$$= \arg\min_{\pi^{-1}\in\Pi^{train}} -\int_{o^{-1},a^{-1}} P_{\mathcal{W}}(o^{-1})P(a^{-1}|o^{-1}, \bar{\pi}^{-1})\log(P(a^{-1}|o^{-1}, \pi^{-1}))do^{-1}da^{-1} \tag{20b}$$

$$= \arg\min_{\pi^{-1}\in\Pi^{train}} -\int_{o^{-1},a^{-1}} P_{\mathcal{W}}(o^{-1})P(a^{-1}|o^{-1}, \bar{\pi}^{-1})\log(P(a^{-1}|o^{-1}, \pi^{-1}))do^{-1}da^{-1}$$

$$+ \int_{o^{-1},a^{-1}} P_{\mathcal{W}}(o^{-1})P(a^{-1}|o^{-1}, \bar{\pi}^{-1})\log(P(a^{-1}|o^{-1}, \bar{\pi}^{-1}))do^{-1}da^{-1} \tag{20c}$$

$$= \arg \min_{\pi^{-1} \in \Pi^{\text{train}}} - \int_{o^{-1}, a^{-1}} P_{\mathcal{W}}(o^{-1}) P(a^{-1}|o^{-1}, \pi^{-1}) \frac{\log(P(a^{-1}|o^{-1}, \pi^{-1}))}{\log(P(a^{-1}|o^{-1}, \bar{\pi}^{-1}))} \mathrm{d}o^{-1} \mathrm{d}a^{-1} \tag{20d}$$

$$= \arg \min_{\pi^{-1} \in \Pi^{\text{train}}} \int_{o^{-1}} P_{\mathcal{W}}(o^{-1}) D_{KL}(P(a^{-1}|o^{-1}, \pi^{-1}) || P(a^{-1}|o^{-1}, \bar{\pi}^{-1})) \mathrm{d}o^{-1} \tag{20e}$$

$$= \arg \min_{\pi^{-1} \in \Pi^{\text{train}}} \mathbb{E}_{o^{-1} \sim P_{\mathcal{W}}(\cdot)} \left[ D_{KL}(P(a^{-1}|o^{-1}, \pi^{-1}) || P(a^{-1}|o^{-1}, \bar{\pi}^{-1})) \right] \tag{20f}$$

This corollary holds evidently due to the non-negativity of the KL divergence $D_{KL}(P(a^{-1}|o^{-1}, \pi^{-1}) || P(a^{-1}|o^{-1}, \bar{\pi}^{-1}))$. When $\mathcal{W}$ contains the in-context data of the opponent policy $\bar{\pi}^{-1}$ from $\Pi^{\text{off}}$, $D_{KL}(P(a^{-1}|o^{-1}, \pi_0^{-1}) || P(a^{-1}|o^{-1}, \bar{\pi}^{-1})) = 0$, and PSOM converges to the equivalent class of opponent policies that have the same action distribution as $\bar{\pi}^{-1}$, i.e., $\forall a^{-1}, P(a^{-1}|o^{-1}, \pi_0^{-1}) = P(a^{-1}|o^{-1}, \bar{\pi}^{-1})$. $\square$

By extension, it can be concluded that when $\mathcal{W}$ contains the in-context data of an opponent policy outside of $\Pi^{\text{off}}$, Lemma D.4 ensures the convergence of PSOM algorithm to the opponent policy that minimizes $D_{KL}(P(a^{-1}|o^{-1}, \pi^{-1}) || P(a^{-1}|o^{-1}, \bar{\pi}^{-1}))$ between actions of the converged opponent policy and the true opponent policy.

Using a similar proof methodology as in Lemma D.4, it can be proved straightforward that: Let $\pi_0'^{-1} = \arg \min_{\pi^{-1} \in \Pi^{\text{train}}} f(\pi^{-1}; \mathcal{W}') = \arg \min_{\pi^{-1} \in \Pi^{\text{train}}} - \int_{o^1, a^1, o'^1, r^1} P_{\mathcal{W}'}(o^1, a^1, o'^1, r^1) \log(P(o'^1, r^1|o^1, a^1, \pi^{-1})) \mathrm{d}o^1 \mathrm{d}a^1 \mathrm{d}o'^1 \mathrm{d}r^1$, then $\forall f(\pi^{-1}; \mathcal{W}') \neq f(\pi_0'^{-1}; \mathcal{W}')$, $\frac{P(\pi^{-1}|(o^1, a^1, o'^1, r^1)_{1:n})}{P(\pi_0'^{-1}|(o^1, a^1, o'^1, r^1)_{1:n})} \xrightarrow{P} 0$.

Now, we introduce a novel lemma to elucidate the properties of the opponent policies attained through the original PS algorithm employing $\mathcal{W}'$ while also concluding the relative optimality of the solutions produced by the PSOM algorithm in comparison to it.

**Lemma D.6.** *Given $o^{-1}, o^1, a^1, \pi_0^{-1}, \bar{\pi}^{-1}$, if $\forall a^{-1}, P(a^{-1}|o^{-1}, \pi_0^{-1}) = P(a^{-1}|o^{-1}, \bar{\pi}^{-1})$ holds, it can be deduced that $\forall o'^1, r^1, P(o'^1, r^1|o^1, a^1, \pi_0^{-1}) = P(o'^1, r^1|o^1, a^1, \bar{\pi}^{-1})$, but the reverse is not true.*

*Proof.* For the forward deduction (*i.e.*, $\Rightarrow$), we have:

$$\forall o'^1, r^1, P(o'^1, r^1|o^1, a^1, \pi_0^{-1})$$

$$= \sum_{o^{-1}, a^{-1}} P(o^{-1}|o^1) P(a^{-1}|o^{-1}, \pi_0^{-1}) P(o'^1, r^1|o^1, a^1, o^{-1}, a^{-1}) \tag{21a}$$

$$= \sum_{o^{-1}, a^{-1}} P(o^{-1}|o^1) P(a^{-1}|o^{-1}, \bar{\pi}^{-1}) P(o'^1, r^1|o^1, a^1, o^{-1}, a^{-1}) \tag{21b}$$

$$= P(o'^1, r^1|o^1, a^1, \bar{\pi}^{-1}). \tag{21c}$$

For the backward deduction (*i.e.*, $\Leftarrow$), counterexamples exist. For example, when $P(o'^1, r^1|o^1, a^1, o^{-1}, a^{-1})$ takes equal values for some $a^{-1} \in \bar{\mathcal{A}}^{-1} \subset \mathcal{A}^{-1}$, it is possible that $\forall a^{-1} \in \mathcal{A}^{-1} \backslash \bar{\mathcal{A}}^{-1}, P(a^{-1}|o^{-1}, \pi_0^{-1}) = P(a^{-1}|o^{-1}, \bar{\pi}^{-1})$ and $\sum_{a^{-1} \in \bar{\mathcal{A}}^{-1}} P(a^{-1}|o^{-1}, \pi_0^{-1}) = \sum_{a^{-1} \in \bar{\mathcal{A}}^{-1}} P(a^{-1}|o^{-1}, \bar{\pi}^{-1})$, so that $\forall a^{-1} \in \mathcal{A}^{-1}, P(a^{-1}|o^{-1}, \pi_0^{-1}) = P(a^{-1}|o^{-1}, \bar{\pi}^{-1})$ does not necessarily holds. This implies that using $(o^1, a^1, o'^1, r^1)$ tuples as in-context data will lead to distributions on opponent policies other than the equivalence class of $\bar{\pi}^{-1}$ with non-zero probability, resulting in suboptimality. $\square$

According to Lemma D.6, $\pi_0^{-1} \subset \pi_0'^{-1}$ holds. According to Corollary D.5, we have $\bar{\pi}^{-1} \in \pi_0^{-1}$. Thus, we can conclude that $\bar{\pi}^{-1} \in \pi_0^{-1} \subset \pi_0'^{-1}$. Hence, the PSOM algorithm converges to fewer solutions with non-zero probabilities compared to the PS algorithm, thereby providing more optimal solutions. In other words, if the PS algorithm converges to the optimal solution, then the PSOM algorithm converges to the optimal solution. This concludes the proof. $\square$

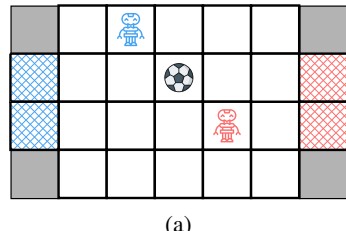

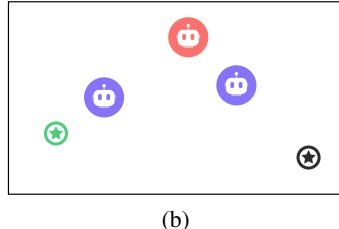

(a)                                                    (b)

Figure 7: The two environmental benchmarks. (a) The `MS` environment. We visualize the entire map at an initial timestep. (b) The `PA` environment. We visualize the clipped map at a certain timestep.

## E  ENVIRONMENTAL SPECIFICS

**Markov Soccer (MS)** (Littman, 1994; Lanctot et al., 2019): A sparse-reward discrete soccer environment that exemplifies a two-player zero-sum game. The game is played on a $4 \times 7$ grid in Fig. 7(a). The game begins with the controlled agent (blue) and the opponent (red) in random squares in the left and right half (except the goals), respectively, and the ball goes to one of them randomly or none of them to a random square in the middle column. The players have five actions: move `N`, `S`, `W`, `E`, or stand still (*i.e.*, `NO-OP`). An action is considered invalid if it leads the player to a shaded square (grey) or outside the border. If a player has the ball, possession will be exchanged and transition will not take place when the two players move into the same square.

Both the control agent's and the opponent's objectives are taking the ball into the *other's goal* (the controlled agent's goal is blue, and the opponent's goal is red), and the game ends when this happens. The reward is sparse, providing $+1$ for success, $-1$ for failure, and $0$ for instances where the maximum timestep limit is exceeded, only at the end of the games.

For specific implementation of `MS`, we adopt the open-source code of `OpenSpiel`, which is available at `https://github.com/deepmind/open_spiel`.

**Particleworld Adversary (PA)** (Lowe et al., 2017): A dense-reward continuous multi-particle environment that exemplifies a three-player non-zero-sum game. `PA` includes two controlled agents (purple), one opponent (red), one normal landmark (black), and one target landmark (green) shown in Fig. 7(b). All agents observe position of landmarks and other agents. There are no borders present on the entire game map.

The controlled agents' objective is to minimize the distance between himself and the target landmark while keeping the opponent away from the target landmark. The opponent's objective is to minimize the distance between himself and the target landmark which is *unknown* to him (*i.e.*, the opponent does not know which one of the landmarks is the target one). This necessitates that the controlled agents develop the skill of strategically distributing themselves to ensure coverage of all landmarks, effectively deceiving the opponent. The two controlled agents and one opponent are given dense rewards induced by their objectives and states using Euclidean distance as the metric.

For specific implementation of `PA`, we adopt the open-source code of `Multi-Agent Particle Environment`, which is available at `https://github.com/openai/multiagent-particle-envs`. Additionally, we introduce a collision mechanism to the original environment to increase the randomness and dynamics of the environment.

## F  NEURAL ARCHITECTURE SETTINGS

In `MS` and `PA`, we adopt the same neural architecture settings as follows:

• **OPE** ($M_{\theta_e}$): The backbone of the OPE architecture is mainly implemented based on the GPT2 (Radford et al., 2019) model of Hugging Face (Wolf et al., 2020). The backbone is a GPT2 encoder composed of 3 self-attention blocks. Each self-attention block consists of a single-head attention layer and a feed-forward layer. Residual connections (He et al., 2016b) and LayerNorm (Ba et al., 2016) are utilized after each layer in the self-attention block. Within each attention layer, dropout (Srivastava et al., 2014) is added to the residual connection and attention weight. Actions

$a^{-1}$, rewards $r^{-1}$, observations $o^{-1}$ are fed into modality-specific linear layers, and a positional episodic timestep encoding is added. We adopt the same timestep encoding as in Chen et al. (2021). Then, we use a *fusion linear layer* to fuse the $a_{t-1}^{-1}, r_{t-1}^{-1}, o_t^{-1}$ embedding tokens at each timestep into *fused embedding tokens*. Finally, the sequences of fused embedding tokens are fed into the backbone, which autoregressively outputs the *per-timestep embedding tokens* corresponding to each $(a_{t-1}^{-1}, r_{t-1}^{-1}, o_t^{-1})$ tuple using a causal self-attention mask.

In offline stage 1, we use Average Pooling (Gholamalinezhad & Khosravi, 2020) to aggregate all the *per-timestep embedding tokens* output by the backbone, *i.e.*, $z_0^{-1}, z_1^{-1}, \ldots$, to obtain the *average trajectory embedding* $\overline{z}^{-1}$ for loss calculation. In offline stage 2 and deployment stage, we input all the per-timestep embedding tokens output by the backbone, *i.e.*, $z^{-1} = \left(z_0^{-1}, z_1^{-1}, \ldots\right)$, as $key$ and $value$ into the cross-attention layers of ICD.

In the backbone, except for the fully connected layer in the feed-forward layer (the feed-forward layer consists of a fully connected layer that increases the number of hidden layer nodes and a projection layer that recovers the number of hidden layer nodes), which consists of 128 nodes with ReLU (Agarap, 2018) activation functions, the other hidden layers are composed of 32 nodes without activation functions. The modality-specific linear layers for actions, rewards, and observations are composed of 32 nodes with ELU (Clevert et al., 2015) activation functions.

- **ICD** ($M_{\theta_d}$): The backbone of the ICD architecture uses the same causal Transformer as in Chen et al. (2021), which is mainly implemented based on the GPT2 model of Hugging Face. The backbone is a GPT2 decoder composed of three self-attention blocks. Each self-attention block consists of a single-head attention layer, a single-head cross-attention layer, and a feed-forward layer. Residual connections and LayerNorm are utilized after each layer in the self-attention block. Within each attention layer, dropout is added to the residual connection and attention weight. RTGs $G^1$, observations $o^1$, actions $a^1$ are fed into modality-specific linear layers, and a positional episodic timestep encoding is added. The sequences consisting of embedding tokens of $G_t^1, o_t^1, a_t^1$ are fed into the backbone, autoregressively predicts actions $a_t^1$ using a causal self-attention mask. Specifically, the obtained hidden states at the positions of $o_t^1$ embedding tokens are fed to a linear layer to output actions.

  In offline stage 2 and deployment stage, we input all embedding tokens output by the backbone as $query$ into the cross-attention layers of all self-attention blocks. For the original input sequence $y_t^1 = (G_0^1, o_0^1, a_0^1, \ldots, G_t^1, o_t^1)$, we truncate the latest $H$ timesteps of it, *i.e.*, $y_t'^1 = (G_{t-H+1}^1, o_{t-H+1}^1, a_{t-H+1}^1, \ldots, G_t^1, o_t^1)$, as input.

  In the backbone, except for the fully connected layer in the feed-forward layer, which consists of 128 nodes with ReLU activation functions, the other hidden layers are composed of 32 nodes without activation functions. The modality-specific linear layers for RTGs, observations, and actions are composed of 32 nodes without activation functions.

- **Ancillary decoder for PEL** ($\pi_{\phi_d}$): The model of $\pi_{\phi_d}$ consists of 2 linear layers, with observation $o^{-1}$, policy embedding $z^{-1}$ as input, and action $a^{-1}$ as output. Among them, $o^{-1}$ and $z^{-1}$ are concatenated and fed to the linear layers, and LayerNorm is added after the first linear layer. All hidden layers comprise 32 nodes; the first hidden layer includes a ReLU activation function.

- **DRON-concat**: For a fair comparison, we use the GPT2 decoder to implement the backbones of the DRON-concat architecture. The DRON-concat architecture contains 2 backbones, and the backbones' architectural design is identical to that of ICD, except that it does not include the cross-attention layer. One of the backbones is used to encode the sequence consisting of RTGs $G^1$, observations $o^1$, and actions $a^1$, while the other backbone is used to encode the opponent's *hand-crafted features* (He et al., 2016a). To be versatile in different environments, we consider using the opponent's previous action and the opponent's previous action frequency as the hand-crafted features. We concatenate the embedding tokens obtained by the two backbone encodings in the dimension of the hidden state and feed them to a *fusion linear layer* to get *concatenated embedding tokens*. Within, the *hand-crafted feature embedding tokens* are aligned with the positions of $a^1$ embedding tokens, and other positions are filled with 0 and aligned with the positions of $G^1, a^1$ embedding tokens. Finally, given the concatenated embedding tokens, the actions $a_t^1$ are autoregressively predicted at the locations of observations $o_t^1$ using the same manner as ICD. Both the fusion linear layer and the linear layer of hand-crafted features comprise 32 nodes without activation functions, while the number of other hidden layer nodes and the activation function settings are identical to those of ICD.

- **DRON-MoE**: For a fair comparison, we use the GPT2 decoder to implement the backbones of the DRON-MoE architecture. The DRON-MoE architecture contains 2 backbones, and the backbones' architectural design is identical to that of ICD, except that it does not include the cross-attention layer. One backbone is used to encode the sequence consisting of RTGs $G^1$, observations $o^1$, and actions $a^1$, while the other backbone is used to encode $G^1, o^1, a^1$ sequence and the opponent's hand-crafted features. For the first backbone: After encoding $G_t^1, o_t^1, a_t^1$, the $o_t^1$ embedding tokens are fed to an *expert linear layer* and output a predefined number of expert of action probability distributions. For the second backbone: Before inputting, the opponent's hand-crafted features are concatenated with the corresponding observations $o^1$ according to the timestep, and they are fed to a *mixing linear layer*; after encoding $G_t^1, o_t^1, a_t^1$ and hand-crafted features, $o^1$ embedding tokens are fed to an *expert linear layer*, and a weight vector with a length of the number of experts is outputted. Finally, the obtained weight vector is used to perform a weighted summation of the obtained action probability distributions, and the actions $a_t^1$ are predicted autoregressively. The mixing linear layer consists of 32 nodes without activation function, and the expert linear layer consists of 32 nodes with Softmax (Bridle, 1989) activation function. The number of other hidden layer nodes and the activation function settings are identical to ICD. The predefined number of experts is set to 5.

- **LIAM**: For a fair comparison, we use the GPT2 model to implement the backbones of the LIAM architecture. The architectural design of LIAM's backbone is the same as that of ICD, except that it does not include a cross-attention layer. In addition to feeding the sequence $G_t^1, o_t^1, a_t^1$ into the backbone and predicting the action $a_t^1$ in an autoregressive manner using a causal self-attention mask, we also employ an *extra decoder* to learn an auxiliary task (Papoudakis et al., 2021). This auxiliary task involves reconstructing the opponent's observations $o_t^{-1}$ and actions $a_t^{-1}$ using the observations $o_t^1$ and actions $a_{t-1}^1$ of the controlled agent. Specifically, we use the $o_t^1$ token embeddings obtained through the backbone as input to the extra decoder to predict $o_t^{-1}$ and $a_t^{-1}$, as the $o_t^1$ token embeddings contain all the information of $o_t^1$ and $a_{t-1}^1$. The extra decoder consists of two linear layers with 32 nodes and no activation functions. The number of other hidden layer nodes and the activation function settings are identical to ICD.

- **MeLIBA**: For a fair comparison, we use the GPT2 model to implement the backbones of the MeLIBA architecture. The architectural design of MeLIBA's backbone is the same as that of ICD, except that it does not include a cross-attention layer. In addition to feeding the sequence $G_t^1, o_t^1, a_t^1$ into the backbone and predicting the action $a_t^1$ in an autoregressive manner using a causal self-attention mask, we also employ *extra encoding layers* and an *extra decoder* to learn an auxiliary task for the Variational AutoEncoders (VAE) (Zintgraf et al., 2021). This auxiliary task aims to maximize Evidence Lower BOund (ELBO) by reconstructing the opponent's future actions $a_t^{-1}, a_{t+1}^{-1}, \cdots$ using the opponent's future observations $o_t^{-1}, o_{t+1}^{-1}, \cdots$ and by minimizing a certain form of KL divergence, conditioned on the past trajectory $o_0^1, a_0^1, r_0^1, \cdots, o_t^1$ of the controlled agent. Specifically, we use additional two-level hierarchical linear layers to encode the controlled agent's past trajectory, where the two levels embed the controlled agent's past trajectory into the Gaussian distribution parameters $\mu$ and $\sigma$ of a *permanent latent variable* (also called *agent character*) and a *temporal latent variable* (also called *mental state*), respectively. On the one hand, $\mu$ and $\sigma$ of the permanent latent variable and the temporal latent variable output the controlled agent's actions to respond to the opponent reasonably. On the other hand, the permanent latent variable and the temporal latent variable, sampled from the Gaussian distribution parameterized by $\mu$ and $\sigma$, are used as the input to the extra decoder, where this decoder conditions also on the opponent's future observations $o_t^{-1}, o_{t+1}^{-1}, \cdots$ to predict the opponent's future actions $a_t^{-1}, a_{t+1}^{-1}, \cdots$. The extra encoding layers consist of two levels of linear layers, where each level has two linear layers with 32 nodes and no activation functions. The extra decoder consists of two linear layers and a recurrent layer, where these layers have 32 nodes and no activation functions. The number of other hidden layer nodes and the activation function settings are identical to ICD.

- **Prompt-DT**: We implement Prompt-DT's architecture directly based on its open-source code, which is available at `https://github.com/mxu34/prompt-dt`. We use *high-quality data* against opponent policies from offline datasets as expert demonstrations (*i.e.*, *prompts*). High-quality data refers to trajectories of the controlled agent that rank near the top in terms of *return* across all episodes in which it competes against a specific opponent policy. Specifically, we select the top 20% for the high-quality data, sample 1 trajectory, and then sample consecutive fragments with a length of 5 from the trajectory as prompts. The linear layers for RTGs, observations, and ac-

tions in the prompts consist of 32 nodes without activation functions. The number of other hidden layer nodes and the activation function settings are identical to ICD.

## G   DETAILS OF BASELINES

**DRON-concat** (He et al., 2016a): An opponent modeling approach that encodes the opponent's hand-crafted features with an auxiliary network. It concatenates the resulting opponent's hidden states and the hidden states obtained by the controlled agent's observation to aid downstream policy optimization of the controlled agent.

**DRON-MoE** (He et al., 2016a): An opponent modeling approach that encodes the opponent's hand-crafted features using a Mixture-of-Expert (MoE) structure (Masoudnia & Ebrahimpour, 2014). It considers the hidden states obtained by the opponent's hand-crafted features and the controlled agent's observations as a gating signal, applying a weighted summation with a predefined number of expert of the hidden states obtained by the controlled agent's observations.

**LIAM** (Papoudakis et al., 2021): An opponent modeling approach that encodes opponent information with autoencoders (Hinton & Zemel, 1993). It utilizes the observations and actions of the controlled agent to reconstruct those of the opponent, thereby embedding the opponent policy's semantics into a latent space. Then, it uses the obtained embeddings to assist downstream policy optimization of the controlled agent.

**MeLIBA** (Zintgraf et al., 2021): An opponent modeling approach that encodes opponent information with Variational AutoEncoders (VAE) (Kingma & Welling, 2013). It uses VAE to reconstruct the opponent's future actions to infer an approximate belief for the opponent. It aims to learn an approximate Bayesian optimal policy based on the embedding generated by the VAE.

**Prompt-DT** (Xu et al., 2022): An offline meta-RL approach based on the Decision Transformer (DT) (Chen et al., 2021) and prompt learning (Liu et al., 2023). Based on the DT, it samples expert task trajectories as prompts and obtains task adaptability through offline dataset pre-training. This approach holds the potential to partially address the challenges posed by the OOM problem.

**TAO without PEL**: A variant of our approach. It performs concurrent optimization to ICD and OPE utilizing only $\mathcal{L}_{\text{res}}$, without pre-training OPE using $\mathcal{L}_{\text{emb}}$. We introduce this baseline to conduct ablation studies for PEL and scrutinize the efficacy of TAO's architectural design.

# H DETAILS OF OPPONENT POLICIES

## H.1 OPPONENT POLICIES IN MARKOV SOCCER

| | Opponent Policy Name | Source | Description |
|---|---|---|---|
| $o1$ | `SnatchAttackPolicy` | script | `Snatch`: Move towards the football along the shortest path when not possessing the ball. `Attack`: Move towards the goal of the controlled agent nearest to the opponent while in possession of the ball, following the shortest path. |
| $o2$ | `SnatchEvadePolicy` | script | `Snatch`: Move towards the football along the shortest path when not possessing the ball. `Evade`: When possessing the ball, move along the path farthest from the controlled agent. |
| $o3$ | `RLPolicy1` | RL algorithm | This policy is obtained through self-play training using the TRCoPO (Prajapat et al., 2021) algorithm for 30000 episodes. The open-source code is available at `https://github.com/manish-pra/trcopo`. We adopt the original hyperparameter settings from the open-source code. |
| $o4$ | `RLPolicy2` | RL algorithm | This policy is obtained through self-play training using the TRGDA (Prajapat et al., 2021) algorithm for 10000 episodes. The open-source code is available at `https://github.com/manish-pra/trcopo`. We adopt the original hyperparameter settings from the open-source code. |
| $o5$ | `RLPolicy3` | RL algorithm | This policy is obtained through self-play training using the PPO (Schulman et al., 2017) algorithm for 50000 episodes. The open-source code is available at `https://github.com/berkeleydeeprlcourse/homework/tree/master/hw4`. We adopt the original hyperparameter settings from the open-source code, and the other opponent policies trained using PPO also use the same settings. |
| $o6$ | `GuardAttackPolicy` | script | `Guard`: When not possessing the ball, defend in front of the opponent's goal nearest to the controlled agent. `Attack`: Move towards the goal of the controlled agent nearest to the opponent while in possession of the ball, following the shortest path. |
| $o7$ | `GuardEvadePolicy` | script | `Guard`: When not possessing the ball, defend in front of the opponent's goal nearest to the controlled agent. `Evade`: When possessing the ball, move along the path farthest from the controlled agent. |
| $o8$ | `RLPolicy4` | RL algorithm | This policy is obtained through self-play training using the PPO algorithm for 100000 episodes. |

## H.2    OPPONENT POLICIES IN PARTICLEWORLD ADVERSARY

|     | Opponent Policy Name | Source | Description |
| --- | --- | --- | --- |
| $o1$ | FixOnePolicy | script | At the beginning of each episode, randomly select one of two landmarks as the `fixed` target. Throughout this episode, always move towards the `fixed` target along the shortest path. |
| $o2$ | ChaseOnePolicy | script | At the beginning of each episode, randomly select one of the two controlled agents as the `chasing` target. Throughout this episode, always move towards the `chasing` target along the shortest path. |
| $o3$ | MiddlePolicy | script | At the beginning of each episode, set the `midpoint` of the two landmarks as the `fixed` target. Throughout this episode, always move towards the `fixed` target along the shortest path. |
| $o4$ | BouncePolicy | script | At the beginning of each episode, randomly select one of two landmarks as the `fixed` target. During this episode, begin by moving towards the `fixed` target along the shortest path. When the distance to the current `fixed` target is less than or equal to 0.096, switch the `fixed` target to the other landmark and then move towards the new `fixed` target along the shortest path. Repeat this process until the episode ends. |
| $o5$ | RLPolicy5 | RL algorithm | This policy is obtained through self-play training using the PPO algorithm for 5000 episodes. |
| $o6$ | FixThreePolicy | script | At the beginning of each episode, randomly choose one of the two landmarks or the midpoint of the two landmarks as the `fixed` target. Throughout this episode, always move towards the `fixed` target along the shortest path. |
| $o7$ | ChaseBouncePolicy | script | At the beginning of each episode, randomly select one of the two controlled agents as the `chasing` target. During this episode, begin by moving towards the `chasing` target along the shortest path. When the distance to the current `chasing` target is less than or equal to 0.3, switch the `chasing` target to the other controlled agent and then move towards the new `chasing` target along the shortest path. Repeat the process until the episode ends. |
| $o8$ | RLPolicy6 | RL algorithm | This policy is obtained through self-play training using the PPO algorithm for 10000 episodes. |

# I CONSTRUCTION OF OFFLINE DATASETS

The offline dataset, denoted as $\mathcal{D}^{\text{off}} = \{\mathcal{D}^k\}_{k=1,2,\ldots,K}$, where $K$ is the predefined number of opponent policies. For each $k \in [K]$, $\mathcal{D}^k$ consists of totally $A$ episodes' interacting trajectories, generated by the interactions between the opponent policy $\pi^{-1,k}$ and the approximate best response policy $\pi^{1,k,*}$ against it.

However, to bring our research in line with real-world scenarios where the offline datasets are always noisy, we introduce certain modifications: We assume that $\mathcal{D}^k, k \in [K]$ comprises a larger proportion, $\eta$, of *expert data* and a smaller proportion, $1-\eta$, of *noisy data*. The *expert data* is derived from the interactions between the opponent policy $\pi^{-1,k}$ and its approximate best response policy $\pi^{1,k,*}$. The *noisy data* is procured by interacting with $\pi^{-1,k}$ through any differing policy, *e.g.*, play against $\pi^{-1,1}$ with $\pi^{1,2,*}$.

In this work, we set the opponent's policy to $\pi^{-1,k}$ and utilize pre-training with an RL algorithm to derive opponent policy $\pi^{-1,k}$'s approximate best response policy $\pi^{1,k,*}$. Additionally, a rough but practical approach to attain $\pi^{1,k,*}$ involves considering the policies generated by top-ranking algorithms in particular competitive contests as approximate best response policies against $\pi^{-1,k}$.

For the specific implementation of the offline dataset $\mathcal{D}^{\text{off}}$, we adopt the following settings:

- In both `MS` and `PA`, we set $K$ to 5, $A$ to 2000, and $\eta$ to 0.9;

- In both `MS` and `PA`, for each $k \in [K]$, we fix the opponent policy as $\pi^{-1,k}$ and use PPO (Schulman et al., 2017) to pre-train its approximately best response policy $\pi^{1,k,*}$;

- In both `MS` and `PA`, for each $k \in [K]$, the *noisy data* is generated by the interactions between $\pi^{-1,k}$ and all possible $\pi^{1,j,*}$, where $j \in [K] \wedge j \neq k$, each for 50 episodes (*i.e.*, accounted for $\frac{1-\eta}{K-1}$ of the $\mathcal{D}^k$).

# J HYPERPARAMETERS

## J.1 HYPERPARAMETERS FOR OFFLINE STAGE 1

| Hyperparameter Name | MS | PA |
|---|---|---|
| Dimensionality of the opponent's observations | 12 | 8 |
| Dimensionality of the opponent's actions | 5 | 5 |
| Whether to normalize the opponent's observations | $True$ | $True$ |
| Maximum horizon length for each episode | 100 | 100 |
| Total number of training steps | 200 | 500 |
| Batch size ($B$) | 128 | 128 |
| Number of updating epochs at each training step | 10 | 10 |
| Weighting coefficient for generative loss $\mathcal{L}_{\text{gen}}$ ($\alpha$) | 1.0 | 1.0 |
| Weighting coefficient for discriminative loss $\mathcal{L}_{\text{dis}}$ ($\lambda$) | 1.0 | 1.0 |
| Warm-up epochs (the learning rate is multiplied by $\frac{\text{num\_epoch}+1}{\text{warm-up epochs}}$ to allow it to increase linearly during the initial warm-up epochs of training) | 10000 | 10000 |
| Learning rate for AdamW (Loshchilov & Hutter, 2018) optimizer | $1 \times 10^{-2}$ | $1 \times 10^{-2}$ |
| Weight decay coefficient for AdamW optimizer | $1 \times 10^{-4}$ | $1 \times 10^{-4}$ |
| Maximum norm of the gradients (clip if exceeded) | 0.5 | 0.5 |
| Temperature factor for discriminative loss $\mathcal{L}_{\text{dis}}$ ($p$) | 0.1 | 0.1 |
| Number of hidden layer nodes for the GPT2 encoder of OPE (see Appendix F for detailed descriptions) | 32 | 32 |
| Dropout factor for the GPT2 encoder of OPE | 0.1 | 0.1 |
| Number of self-attention blocks for the GPT2 encoder of OPE | 3 | 3 |
| Number of attention head for the GPT2 encoder of OPE | 1 | 1 |
| Random seeds | $\{0, 1, 2, 3, 4\}$ | $\{0, 1, 2, 3, 4\}$ |

### J.2 HYPERPARAMETERS FOR OFFLINE STAGE 2

| Hyperparameter Name | MS | PA |
|---|---|---|
| Dimensionality of the opponent's observations | 12 | 8 |
| Dimensionality of the opponent's actions | 5 | 5 |
| Whether to normalize the opponent's observations | $True$ | $True$ |
| Dimensionality of the controlled agent's observations | 12 | 10 |
| Dimensionality of the controlled agent's actions | 5 | 5 |
| Whether to normalize the controlled agent's observations | $True$ | $True$ |
| Maximum horizon length for each episode | 100 | 100 |
| Length of the consecutive fragment for each trajectory sampled from the offline dateset $\mathcal{D}^{\text{off}}$ by OPE ($H$) | 20 | 20 |
| Number of trajectories sampled by OPE ($C$) | 5 | 5 |
| Length of the input sequence for ICD after truncation ($H$) (see Appendix F for detailed descriptions) | 20 | 20 |
| Reward scaling factor (all the rewards are multiplied by $\frac{1}{\text{reward scaling factor}}$ to reduce the variance of training) | 1 | 100 |
| Total number of training steps | 2000 | 2000 |
| Batch size ($B$) | 128 | 128 |
| Number of updating epochs at each training step | 10 | 10 |
| Warm-up epochs | 10000 | 10000 |
| Learning rate for AdamW optimizer | $1 \times 10^{-4}$ | $1 \times 10^{-4}$ |
| Weight decay coefficient for AdamW optimizer | $1 \times 10^{-4}$ | $1 \times 10^{-4}$ |
| Maximum norm of the gradients | 0.5 | 0.5 |
| Number of hidden layer nodes for the GPT2 encoder of OPE | 32 | 32 |
| Dropout factor for the GPT2 encoder of OPE | 0.1 | 0.1 |
| Number of self-attention blocks for the GPT2 encoder of OPE | 3 | 3 |
| Number of attention head for the GPT2 encoder of OPE | 1 | 1 |
| Number of hidden layer nodes for the GPT2 decoder of ICD | 32 | 32 |
| Dropout factor for the GPT2 decoder of ICD | 0.1 | 0.1 |
| Number of self-attention blocks for the GPT2 decoder of ICD | 3 | 3 |
| Number of attention head for the GPT2 decoder of ICD | 1 | 1 |
| Random seeds | $\{0, 1, 2, 3, 4\}$ | $\{0, 1, 2, 3, 4\}$ |

### J.3 HYPERPARAMETERS FOR DEPLOYMENT STAGE

| Hyperparameter Name | MS | PA |
|---|---|---|
| Dimensionality of the opponent's observations | 12 | 8 |
| Dimensionality of the opponent's actions | 5 | 5 |
| Whether to normalize the opponent's observations | $True$ | $True$ |
| Dimensionality of the controlled agent's observations | 12 | 10 |
| Dimensionality of the controlled agent's actions | 5 | 5 |
| Whether to normalize the controlled agent's observations | $True$ | $True$ |
| Maximum horizon length for each episode | 100 | 100 |
| Total number of episodes for testing | 2500 | 2500 |
| Opponent policy switching interval ($E$) (the opponent switches its policy every $E$ episodes) | 50 | 50 |
| Length of the consecutive fragment for each trajectory sampled from the OCW by OPE ($H$) | 20 | 20 |
| Size of the OCW ($C$) (the OCW collects the latest $C$ opponent trajectories) | 5 | 5 |
| Length of the input sequence for ICD after truncation ($H$) | 20 | 20 |
| Reward scaling factor | 1 | 100 |
| Number of hidden layer nodes for the GPT2 encoder of OPE | 32 | 32 |
| Dropout factor for the GPT2 encoder of OPE | 0.1 | 0.1 |
| Number of self-attention blocks for the GPT2 encoder of OPE | 3 | 3 |
| Number of attention head for the GPT2 encoder of OPE | 1 | 1 |
| Number of hidden layer nodes for the GPT2 decoder of ICD | 32 | 32 |
| Dropout factor for the GPT2 decoder of ICD | 0.1 | 0.1 |
| Number of self-attention blocks for the GPT2 decoder of ICD | 3 | 3 |
| Number of attention head for the GPT2 decoder of ICD | 1 | 1 |
| Random seeds | $\{0, 1, 2, 3, 4\}$ | $\{0, 1, 2, 3, 4\}$ |

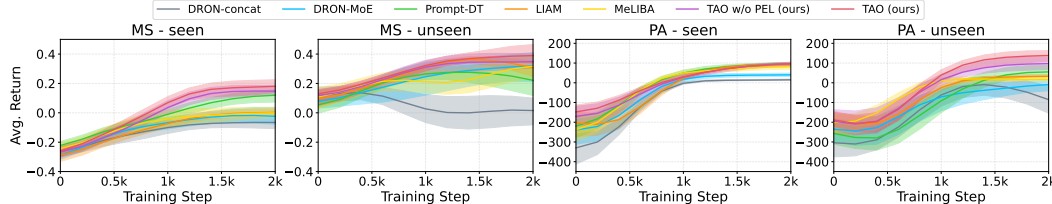

Figure 8: Performance for offline response policy training. All approaches are evaluated against each opponent policy in *seen* and *unseen* for 100 episodes *separately* every 200 training steps. We report the average and standard error of the mean results against all opponent policies over 5 runs.

Table 2: Experimental results on adaptation speed.

|  | MS (Avg. Min. Episode ↓) | | | PA (Avg. Min. Episode ↓) | | |
|---|---|---|---|---|---|---|
|  | *seen* | *unseen* | *mix* | *seen* | *unseen* | *mix* |
| TAO (ours) | **2.74 ± 0.27** | **3.78 ± 0.93** | **2.83 ± 0.32** | **1.9 ± 0.2** | **3.7 ± 0.5** | **2.8 ± 0.2** |
| TAO w/o PEL (ours) | 4.01 ± 0.56 | 5.82 ± 0.71 | 3.84 ± 0.23 | 1.9 ± 0.2 | 8.4 ± 0.9 | 4.1 ± 0.5 |
| DRON-concat | 14.53 ± 1.78 | 29.27 ± 1.44 | 19.60 ± 2.10 | 5.4 ± 1.0 | 7.2 ± 1.6 | 5.2 ± 1.0 |
| DRON-MoE | 13.99 ± 0.97 | 6.62 ± 1.86 | 6.95 ± 1.50 | 3.5 ± 0.3 | 6.7 ± 0.4 | 4.9 ± 0.5 |
| LIAM | 6.90 ± 0.88 | 7.66 ± 0.35 | 5.78 ± 0.68 | 2.6 ± 0.4 | 10.4 ± 2.1 | 6.5 ± 0.9 |
| MeLIBA | 6.12 ± 0.74 | 12.51 ± 1.75 | 6.36 ± 1.08 | 2.9 ± 0.2 | 13.3 ± 1.0 | 6.8 ± 0.8 |
| Prompt-DT | 5.67 ± 0.60 | 17.70 ± 2.29 | 8.19 ± 1.53 | 2.1 ± 0.2 | 16.7 ± 0.9 | 7.5 ± 0.8 |

## K    ADDITIONAL EMPIRICAL ANALYSIS

To further examine TAO's effect and underlying mechanisms, we continue to ask the following questions and design corresponding experiments to answer them:

**Question 5.** *How does TAO perform in the offline training compared with other baselines?*

In Fig. 8, we present the average results of playing against opponent policies in the opponent policy sets *separately* (*i.e.*, without switching opponent policies) during offline response policy training. The evaluation outcomes for TAO exhibit a consistent and incremental improvement under both seen and unseen settings as the training progresses. In contrast, other approaches like DRON-concat display relatively unstable performance during training.

The evaluations during offline training and the subsequent testing results in Fig. 3 demonstrate that the relative performance of all approaches remains constant. However, the performance of most approaches during testing has shown certain improvements compared to the last evaluation of the offline training. This observation can likely be attributed to the increased number of episodes conducted during testing, offering more chances to exploit the *opponent policies with low strength* (*i.e.*, weak and exploitable opponent policies).

**Question 6.** *Does TAO demonstrate faster **adaptation speed** compared with other baselines?*

We measure the **adaptation speed** by the minimum number of episodes to achieve a *specified return*. For all opponent policies in the *seen* test setting, the *specified return* is set to be the average return of all trajectories against this policy in the offline dataset; for all opponent policies in the *unseen* test setting, the *specified return* is set to the average return of all approaches against this opponent policy under the *unseen* test setting.

Table 2 shows the average adaptation speed under the *original opponent policy configuration*. TAO showcases consistently faster adaptation in MS and PA than other approaches, underscoring its capability to acquire knowledge through in-context learning swiftly. Furthermore, it's important to highlight that there isn't a straightforward positive correlation between adaptation speed and performance. For instance, in scenarios involving unseen opponent policies, Prompt-DT performs admirably but often displays a sluggish adaptation speed.

**Question 7.** *How does the opponent context window (OCW) size $C$ affect the efficacy of TAO?*

We vary the OCW size $C$ to investigate how the context length affects TAO, and the relevant experimental results are shown in Fig. 9. In both environments, the average performance exhibits a similar trend pattern as $C$ changes. It is worth noting that, as $C$ increases, the average performance

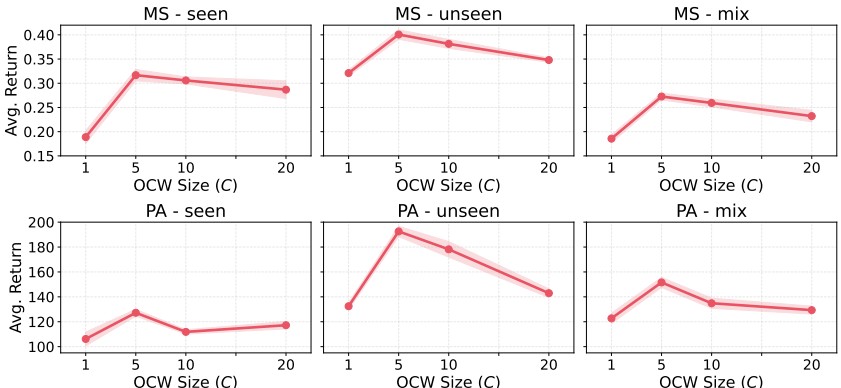

Figure 9: Average performance for TAO on different sizes of the OCW.

does not increase monotonically under the three test settings. Under the *unseen* and *mix* settings, a continual decrease in performance is evident, beginning at $C = 5$. One possible explanation is that when $C$ is relatively large, due to the non-stationary opponent policy switching, TAO is prone to be influenced by the context of the previous opponent, making it challenging to recognize the current opponent accurately. The quest to theoretically and empirically identify the most suitable value for $C$ is reserved for future research endeavors.

## L  LIMITATIONS AND FUTURE WORK

1. We only focus on exploiting opponent with fixed policies. TAO might face difficulties handling opponent who exhibit policy alterations within an episode or evolve (update) their policies as the games unfold. OOM with changing policy opponents represents a formidable but crucial problem;

2. We do not consider recursive reasoning, *i.e.*, scenarios where opponent also perform opponent modeling. A challenging future research direction lies in learning opponent models for recursive reasoning using offline datasets;

3. Our empirical analysis merely models one opponent agent in the environment. In situations involving a larger number of agents to be modeled, it becomes imperative to account for the interrelated behaviors and correlations among these agents to ensure the efficacy of modeling;

4. This paper's scope is limited to examining OOM problems exclusively within competitive settings. Our future research will focus on understanding and mitigating the distinct challenges that arise in cooperative and mixed environments.

