# OpenReview forum: "Towards Offline Opponent Modeling with In-context Learning"
_ICLR.cc/2024/Conference — ICLR 2024 poster_

### Official Review · Reviewer_F2KT · 2023-10-13

**Soundness:** 2 fair
**Presentation:** 2 fair
**Contribution:** 2 fair
**Rating:** 6
**Confidence:** 3

**Summary:**

This work investigates the problem of rapid adaptation to an unknown opponent policy. The proposed approach is to learn a latent opponent model and play the response to the currently predicted opponent. They developed this under the namesake TAO (Transformers Against Opponent), a sequential application of three existing methods (learning policy embeddings, computing responses to known opponents, and few-shot adaptation). TAO is evaluated in the Markov Soccer and Multiagent Particle games and tends to perform at least as well as an existing method.

**Strengths:**

- I view the main effort of this work as addressing the question "Can transformers solve the few-shot opponent adaptation problem?" To which, it appears the answer is yes, at least as well as existing approaches.
- The related work is extensive, especially when it considers different methods of opponent modeling. It is also worth noting that ad-hoc teamwork is a relevant literature area despite focusing on common-interest/team games.
- The implementation details of both their methods and their versions of the baseline methods are extensively detailed in the appendix.

**Weaknesses:**

- The major weakness of this work is the lack of precision in the discussion of their empirical results. In each experiment, the authors skip straight to making a concluding remark after describing their experimental method. A discussion of the results, beyond "see Fig X" is required and why the provided evidence leads to the resulting conclusion. In general, I also found the claims to be too strong for the provided figures.
- This work redefines the notion of "offline opponent modeling" and expands its problem description to also include adaptation. I think this redefinition confuses more than aids in clarifying the problem space. I would strongly recommend against this redefinition. I think this also causes this paper to feel like it's trying to do too much and then only do a cursory treatment of each problem (modeling and adaptation). I think the paper would have been stronger if the authors focused on these problems independently and then together.
- It's not clear to me what is added to our understanding of this problem from using transformers and "in-context learning" --- a vogue rebranding of concepts like correlation devices. I was hoping the authors could comment on their vision of this research, beyond that transformers are a neural network architecture that previous methods cannot readily apply.


**Minor (Not Effecting Score)**
- The language in the paper at times feels very unnatural (uncanny ChatGPT-esque), and I personally found it hard to read.
- In the figures "# Opponent Policy" --> "# Opponent Policies".

**Questions:**

- Sec 3.1, "Specifically Tk1 ... usually with certain noise." The notation is a touch confusing here wrt your datasets. Does this mean that you only consider datasets where each player plays the same policy? If so, why?
- Sec 4.2, how do you control for the ego agent's influence on the trajectory?
  - It's not clear to me what these "fragments" are, why you're taking them, and why it's sensible to stitch together what appears to be sub-trajectories from differing episodes.
  - I would have liked to see baselines/ablations related to their opponent-modeling proposal.
- Sec 4.4, why are the differences in optimality guarantees not directly stated in the paper?
  - What is it about TAO that enables the improvement in results?
  - It's not clear to me that TAO should beget any improvements, so is it proof of a stronger bound?
- Sec 5.1, "We adopt three kinds of test settings for Π test, where seen is equivalent to the offline opponent policy set Π off, unseen contains some previously unseen opponent policies, and mix is a mixture of the former two." It appears _unseen_ and _mix_ are redundant and contain only different partitions of held-in and held-out policies.
  - Why not also test generalizations to strictly held-out policies?
  - How were the ordering of policies selected for the adaptation experiments?
- Figure 4 has a non-standard y scale. Why are all the BR performances so bad against the subset of opponents that necessitated expanding the low-performance bracket? This smells like there is an issue in the quality/diversity of the opponent population because there are only extremely high and low-scoring performances.
- Sec 5.2, Question 1
  - "TAO consistently outperforms other baselines ... this implies that TAO can respond effectively to unseen ..." No evidence supporting this claim, and it is exceptionally strong. At best, the authors can claim that it performs better than baseline methods.
  - Suggest also computing best responses directly to each opponent independently. This allows some measurement of what BR performance actually is, so we can have a better estimate of how much of true BR performance is recovered and how quickly.
  - In Figure~4, while it is true that TAO looks like it's performing well, I don't think there's any clear narrative here. It's losing to other methods sometimes, often draws with many other methods, and the variance is exceptionally large. Indicates the need for additional study as to when each method is effective.
  - How are you measuring "strong adaptability ... swiftly adjusting"?
  - Fig 3 (b) is not explained ever and I'm not sure what it's trying to say.
- Sec 5.2, Question 2
  - It's not clear to me how a tSNE embedding is sufficient to show "ability to recognize". I would expect a clear quantitive result here.
  - One such way to do this would be to look at the similarity of the embeddings of known opponent trajectories compared to circumstances where the agent has varying degrees of information inaccuracy. For example, it's previous episodic context could be from a different opponent. Another option is to measure the clustering of the embeddings. Or one could consider training a classifier on the frozen embeddings to see how well it could classify their originating opponent policy.
  - Moreover, the authors need to supply similarity comparisons of their opponent policies.
    - From investigating their description in the appendix I'm actually surprised they're not overlaping in tSNE space. Some are very similar!
    - An explanation for this could be that the game is just very easy and each opponent policy defines nearly unique state spaces (after the initial few steps). Any performance differences could be attributed to that method/BR being poorly trained in that specific game state.
- Sec 5.2, Question 3
  - "This suggests that PEL .... robust in-context learning abilities" No evidence offered to support that PEL offers exactly "more robust in-context learning abilities".
  - Should mention whether these results match previous work and discuss their (ag/disag)reement. This experiment appears to me to be largely a reproducibility experiment of a known method for policy representation learning.
- I would have preferred if the authors had ablated the various three components of TAO. Did the authors investigate these?
- For Figure 3 and 6 could you account for the performance difference between the subplots and why it appears that in 3(b), 4, and 6(b) the error is often much higher than seen in any bar charts of tables.
- Perhaps the most interesting result, to me, is that of Question 7 in the appendix. Where you see evidence that larger context lengths can further confuse TAO as their is further uncertainty as to what evidence should be considered. I was wondering if the authors could speak more on this and how they think this could be addressed?

---

> ### Author Response · Authors · 2023-11-18
> **Response to Reviewer F2KT (1/5)**
>
> Thank you very much for your valuable feedback. In response to your comments, we would like to make the following clarifications and feedback. We hope our explanations and analyses can eliminate concerns and make you find our work stronger.
>
> > “The major weakness of this work is the lack of precision in the discussion of their empirical results. In each experiment, the authors skip straight to making a concluding remark after describing their experimental method. A discussion of the results, beyond "see Fig X" is required and why the provided evidence leads to the resulting conclusion. In general, I also found the claims to be too strong for the provided figures.”
> >
>
> We apologize for any confusion caused by our writing. The experimental section in our current version lacks clarity in logic, and the expressions may not be very accurate, making it challenging to present the experimental results and insights well. This is caused by the compression and omission of many intuitive analyses and explanations due to space limitations. Therefore, we take your valuable feedback into account and rewrite and revise this section to make the experimental descriptions more accurate, complete, and logically structured. The parts we changed are highlighted with blue color. Thank you very much for pointing this out; it greatly helps improve the experiment section of our manuscript. We kindly request your review of the updated version.
>
> > “This work redefines the notion of "offline opponent modeling" and expands its problem description to also include adaptation.”
> >
>
> We apologize for not providing you with a better understanding of the relevant background in the opponent modeling domain. The field of "opponent modeling" itself not only focuses on how to model certain attributes of opponents, such as behavior, beliefs, goals, etc. but also includes how to use this information to adapt to opponents. Both "modeling" and "adapting" are essential and closely connected aspects. In summary, the ultimate goal of "opponent modeling" is to better adapt to opponents in the environment and enhance decision-making.
>
> > “trying to do too much and then only do a cursory treatment of each problem (modeling and adaptation)”
> >
>
> As mentioned by Reviewer 1, we carefully designed a series of techniques to overcome OOM's challenges. TAO is built upon observations and understanding of the OOM problem: 1) OOM requires the characterization of the semantic aspects of opponent policies, and thus, we use PEL to capture high-level features and differences inherent in different opponent policies. 2) OOM is essentially a task adaptation problem suitable for handling with in-context learning methods. Accordingly, we design Stage 2 and the deployment stage to endow TAO with in-context learning capabilities. 3) Transformers have been empirically and theoretically shown to work in leveraging in-context learning capabilities in task adaptation ([1][2][3]). Thus, we adopt a Transformer-based architecture.
>
> [1] Luckeciano C Melo. Transformers are meta-reinforcement learners. In *International Conference on Machine Learning*, 2022.
>
> [2] Michael Laskin, et al. In-context reinforcement learning with algorithm distillation. In *International Conference on Learning Representations*, 2023.
>
> [3] Jonathan N Lee, et al. Supervised pretraining can learn in-context reinforcement learning. *arXiv preprint arXiv:2306.14892*, 2023.
>
> > “It's not clear to me what is added to our understanding of this problem from using transformers and "in-context learning" --- a vogue rebranding of concepts like correlation devices.”
> >
>
> We apologize for any confusion caused, and that was not our intention. In-context learning can be seen as a type of meta-learning method, and it has been empirically proven to be applicable to task adaptation problems in decision-making domains ([2][3][4]). Essentially, TAO falls under the category of in-context learning methods. The choice of using Transformer as the architecture for TAO is grounded in its widely demonstrated strong in-context learning capabilities. All these design choices are made based on our understanding and intuition regarding the challenges posed by the OOM problem.
>
> [4] Yan Duan, et al. RL2: Fast reinforcement learning via slow reinforcement learning. *arXiv preprint arXiv:1611.02779*, 2016.

---

> ### Author Response · Authors · 2023-11-18
> **Response to Reviewer F2KT (2/5)**
>
> > “I was hoping the authors could comment on their vision of this research”
> >
>
> We appreciate your perspective, especially your statement, "I view the main effort of this work as addressing the question 'Can transformers solve the few-shot opponent adaptation problem?'" However, we not only aim to explore the question you mentioned but also strive to uncover the inefficiencies and impracticalities existing in the current opponent modeling research. By inefficiencies, we argue that online learning of opponent models bears substantial complexity and inefficiency. As with many existing algorithms, they gather copious data and undertake intricate online updating processes. By impracticalities, we argue that online learning struggles to satisfy real-world scenarios, given that interactions with opponent policies are not consistently accessible. Moreover, we aim for the approach we designed to be theoretically guaranteed and empirically effective. Our further goal is to contribute foundational insights to the development of autonomous agents. Given the opportunity, we aspire to investigate further the learning of autonomous agents in more complex and realistic environments, building upon the foundations laid by this work.
>
> > “In the figures "# Opponent Policy" --> "# Opponent Policies".”
> >
>
> Thanks for your correction. We have revised this in the new version.
>
> > “Sec 3.1, … Does this mean that you only consider datasets where each player plays the same policy? If so, why?”
> >
>
> In our offline dataset, for any opponent policy type $k$, each game is played with the opponent policy $\pi^{-1,k}$ and its corresponding approximate best response $\pi^{1,k,\ast}$, resulting in trajectories $\tau^{-1,k}$ and $\tau^{1,k}$, respectively. In each game, some players adopt $\pi^{-1,k}$, while the remaining players adopt $\pi^{1,k,\ast}$. Not all players follow the same policy in a given game.
>
> > “Sec 4.2, how do you control for the ego agent's influence on the trajectory?” “I would have liked to see baselines/ablations related to their opponent-modeling proposal.”
> >
>
> Sorry, we didn't quite understand your meaning. Could you please provide more details or clarification regarding these questions?
>
> > “It's not clear to me what these "fragments" are, why you're taking them, and why it's sensible to stitch together what appears to be sub-trajectories from differing episodes.”
> >
>
> We designed the organization of in-context data with the following motivations: Firstly, sampling continuous segments of opponent trajectories is because the opponent policy's style may manifest differences and characteristics over a continuous time segment. Compared to randomly sampling in-context data at each timestep, this approach may reduce learning difficulty. Secondly, randomly sampling such segments from $C$ trajectories, instead of sampling from a single trajectory, considers various situations and corresponding behaviors that the same opponent policy may encounter in different gameplays. This consideration may enhance the robustness of the learning process.
>
> In the early stages of our experiments, we also explored two different sampling methods for in-context data: 1) Randomly sample one trajectory from $\mathcal{T}^{-1,k}$ and then randomly sample $C\cdot H$ tuples of $(o^{-1},a^{-1},r^{-1})$ (not necessarily consecutive) from this trajectory. 2)Randomly sample one trajectory from $\mathcal{T}^{-1,k}$ and then randomly sample $C$ consecutive segments of $(o^{-1},a^{-1},r^{-1})$ with length $H$. However, empirically, these two methods did not perform as well as the final method we adopted. Nonetheless, there was only a minor difference in performance between the different sampling strategies for in-context data.
>
> > “Sec 4.4, why are the differences in optimality guarantees not directly stated in the paper?”
> >
>
> Due to space constraints, we placed detailed theoretical analyses in Appendix D.

---

> ### Author Response · Authors · 2023-11-18
> **Response to Reviewer F2KT (3/5)**
>
> > “It's not clear to me that TAO should beget any improvements, so is it proof of a stronger bound?”
> >
>
> In Theorem 2, we prove that the PSOM algorithm is guaranteed to converge, and it converges to a solution that is "better" than the PS algorithm. Here, "better" means that if PS converges to the optimal solution, PSOM must also converge to the optimal solution. In intuitive terms, when the current true opponent policy belongs to $\Pi^{\text{off}}$, if PS correctly recognizes the current true opponent policy, PSOM is guaranteed to recognize it correctly. Even if PS fails to recognize the current true opponent policy correctly, PSOM still has a possibility of making a correct recognition. Based on Theorem 1, TAO shares the same properties as the PSOM algorithm mentioned above. For the optimality of the PS algorithm, please refer to the analysis of its regret bounds in [5].
>
> [5] Ian Osband, et al. (more) efficient reinforcement learning via posterior sampling. *Advances in Neural Information Processing Systems*, 2013.
>
> Please note that we have not provided a guarantee that our algorithm has "improvements" compared to the baselines we have compared with.
>
> > “Sec 5.1, … Why not also test generalizations to strictly held-out policies?”
> >
>
> We are sorry for any confusion. In the statement "unseen contains some previously unseen opponent policies," "some" refers to the fact that there is more than one policy in the *unseen* set, not that some proportion of them are previously unseen. In our experimental setup, all opponent policies in the *unseen* set are strictly different from those in the *seen* set.
>
> > “How were the ordering of policies selected for the adaptation experiments?”
> >
>
> In the adaptation experiments, we switch the opponent policy every $E$ episodes. Specifically, "switching" means randomly sampling an opponent policy from the corresponding opponent policy set according to a uniform distribution. Taking "seen" as an example, where there are a total of $5$ opponent policy types, "switching" implies randomly sampling an opponent policy from $o1, o2, ..., o5$.
>
> > “Figure 4 has a non-standard y scale. Why are all the BR performances so bad against the subset of opponents that necessitated expanding the low-performance bracket? This smells like there is an issue in the quality/diversity of the opponent population because there are only extremely high and low-scoring performances.”
> >
>
> In PA, the observed phenomenon is not only related to the strength/diversity of opponent policies but also to the nature of the PA game itself—it is a game with relatively dense rewards. Therefore, for low-strength opponent policies, their exploitation by the controlled agent can easily accumulate over time, resulting in exaggerated return values for the controlled agent in each episode. In practice, it is essential to focus on the performance differences between approaches relative to the same opponent policy.
>
> > “Suggest also computing best responses directly to each opponent independently. This allows some measurement of what BR performance actually is, so we can have a better estimate of how much of true BR performance is recovered and how quickly.”
> >
>
> Thank you very much for your suggestion. We have added the performance of each opponent policy's approximate best response in the two environments to the table below, hoping it will be helpful for you. We will incorporate these important results into the paper in subsequent revisions.
>
> |  | o1 | o2 | o3 | o4 | o5 | o6 | o7 | o8 |
> | --- | --- | --- | --- | --- | --- | --- | --- | --- |
> | MS (avg. score) | 0.79+-0.04 | 0.46+-0.04 | 0.00+-0.01 | -0.01+-0.01 | 0.09+-0.05 | 0.96+-0.01 | 0.99+-0.01 | 0.00+-0.05 |
> | PA (avg. return) | 25.17+-0.95 | 777.88+-1.48 | 31.05+-0.27 | 33.94+-0.55 | 19.76+-0.43 | 24.86+-1.56 | 775.14+-4.57 | 31.27+-0.61 |
>
> Each opponent policy plays against its approximate best response for 200 episodes. The results in the table show the average return of the approximate best response over five random seeds.

---

> ### Author Response · Authors · 2023-11-18
> **Response to Reviewer F2KT (4/5)**
>
> > “Fig 3 (b) is not explained ever and I'm not sure what it's trying to say.”
> >
>
> Figure 3(b) illustrates the performance of all approaches at the granularity of opponent policies (50 episodes). We aim to convey that our approach exhibits relatively competitive performance overall when facing different opponent policies.
>
> > “Sec 5.2, Question 2, It's not clear to me how a tSNE embedding is sufficient to show "ability to recognize". I would expect a clear quantitive result here.” “One such way to do this would be to look at the similarity of the embeddings of known opponent trajectories compared to circumstances where the agent has varying degrees of information inaccuracy. For example, it's previous episodic context could be from a different opponent. Another option is to measure the clustering of the embeddings. Or one could consider training a classifier on the frozen embeddings to see how well it could classify their originating opponent policy.”
> >
>
> Thank you very much for your suggestion. We have added quantitative metrics to measure TAO's ability to recognize opponent policies based on opponent policy embeddings. Specifically, we followed your advice: training a classifier on the frozen embeddings to see how well it could classify their originating opponent policy. The results can be found at [this link](https://postimg.cc/nsSZHvdv), and we will incorporate these important results into the paper in the subsequent revisions.
>
> We use 70% of opponent trajectories from the offline dataset as the training set and 30% as the test set. The results in the figure represent the accuracy, averaged over five random seeds, and the standard deviation.
>
> Under the quantitative evaluation method you suggested, the trained classifier can effectively classify the true opponent policy types to which different frozen embeddings belong.
>
> > “Moreover, the authors need to supply similarity comparisons of their opponent policies.” “From investigating their description in the appendix I'm actually surprised they're not overlaping in tSNE space. Some are very similar!” “An explanation for this could be that the game is just very easy and each opponent policy defines nearly unique state spaces (after the initial few steps). Any performance differences could be attributed to that method/BR being poorly trained in that specific game state.”
> >
>
> I believe your analysis has some validity. However, the similarity between opponent policies is not the primary focus of this work.
>
> In PEL, we employ the discriminative loss, aiming to distinguish the embeddings of "different" opponent policies as much as possible and bring the embeddings of "same" opponent policies as close as possible. "different" but not "dissimilar" and "same" but not "similar.”
>
> **\*** Please note that even if two opponent policies are similar under certain measurements, the best responses to them may not be similar.
>
> Considering (**\***), we aim to distinguish even highly similar opponent policies, helping offline stage 2 better learn responses to different opponent policies. Thus, during deployment, TAO strives to differentiate seen opponent policies and provide correct responses. For an unseen opponent policy, if its trajectory is similar to one generated by a seen opponent policy, the two trajectory embeddings will be close in the embedding space. In this case, TAO can leverage knowledge about responding to the seen opponent policy to respond to the unseen opponent policy. If the trajectory of the unseen opponent policy is not similar to any seen opponent policy, its embedding might be a combination of embeddings from various seen opponent policies. TAO will appropriately combine the knowledge of responding to each seen opponent policy in that combination to generate a suitable response. Such intuitions stem from the discussions and analyses in the proving process of Theorem 2. Please refer to Appendix D.3 for relevant details.
>
> Also, because of (**\***), bringing embeddings of "similar" opponent policies as close as possible may not be effective, as their best responses may differ significantly. Therefore, learning embeddings based on similarity might not only fail to assist in learning downstream response policies but could also confuse the model. Another potentially more appropriate direction is to make *some certain embedding* (different from the embedding obtained by PEL) of "similar" approximate best response policies as close as possible, which might be more helpful for learning response policies against different opponent policies. The ICD component of TAO essentially adopts this idea and has shown promising results empirically.

---

> ### Author Response · Authors · 2023-11-18
> **Response to Reviewer F2KT (5/5)**
>
> > “Should mention whether these results match previous work and discuss their (ag/disag)reement. This experiment appears to me to be largely a reproducibility experiment of a known method for policy representation learning.”
> >
>
> Sorry, we didn't quite understand your meaning. Could you please provide more details or clarification regarding these questions?
>
> > “I would have preferred if the authors had ablated the various three components of TAO. Did the authors investigate these?”
> >
>
> We are sorry. Which additional ablation experiments do you think are needed? We are willing to supplement. We have conducted the following ablation studies in this paper: 1) the quality of policy embedding that TAO learned; 2) the contribution of each component in PEL on TAO; 3) the robustness of TAO to different opponent policy configurations; 4) the influence of different sizes of OCW on TAO.
>
> > “For Figure 3 and 6 could you account for the performance difference between the subplots and why it appears that in 3(b), 4, and 6(b) the error is often much higher than seen in any bar charts of tables.”
> >
>
> The experimental results in Figures 3 and 6 are consistent, but the ranges of the axes are different. The errors in Figures 3(a) (Figure 6(a)), 3(b) (Figure 6(b)), and Figure 4 are all standard deviations of the means over five random seeds, just with different granularities of averaging. The standard deviations in Figure 3(a) are calculated using the mean of returns over $2500$ episodes, Figure 3(b) over $50$ episodes, and Figure 4 over $1$ episode. Clearly, with more episodes used to calculate the mean, the variance of that mean would be lower.
>
> > “Perhaps the most interesting result, to me, is that of Question 7 in the appendix. Where you see evidence that larger context lengths can further confuse TAO as their is further uncertainty as to what evidence should be considered. I was wondering if the authors could speak more on this and how they think this could be addressed?”
> >
>
> We are excited that you are interested in exploring potential ideas for our future work. One possible direction is to maintain a larger OCW size and dynamically remove outdated data. Specifically, a naive implementation could involve explicitly predicting whether new trajectories and old trajectories belong to the same opponent policy. When the confidence is high, we could consider removing all old trajectories to avoid their negative impact on current decision-making.
>
> All your questions and feedback have greatly contributed to improving our manuscript. With the valuable input from you and all other reviewers, our paper has undergone significant changes. We welcome further comments from you and will promptly consider your suggestions for revisions. If you feel that we have addressed your concerns and the quality of the paper has improved, we hope you will reconsider your rating.

---

> > ### Comment · Reviewer_F2KT · 2023-11-20
> >
> > I'd like to first thank the authors for extensively addressing my questions and comments to this point. I have correspondingly increased my rating of the manuscript. Follow up questions and comments follow.
> >
> >
> > > This consideration may enhance the robustness of the learning process.
> >
> > The speculation of robustness from data augmentation appears a bit intuitive. However, it's not clear to me that this generates self-consistent trajectories once the sampled segments are concatenated. I would have preferred this methodological detail be explicitly analyzed as it may confound results.
> >
> >
> > > “This work redefines the notion of "offline opponent modeling" and expands its problem description to also include adaptation.”
> >
> > I do not disagree with the authors that modeling and adaptation are both related problems. However, the problem of building an opponent model can be studied in insolation, and he resulting model can be used beyond adaptation. I am strongly encouraging the authors to modify the language to not conflate the problems.
> >
> >
> > > “It's not clear to me what is added to our understanding of this problem from using transformers and "in-context learning" --- a vogue rebranding of concepts like correlation devices.”
> >
> > My question here was maybe too indirect. What have we learned about the opponent modeling problem or in-context learning from applying transformers to be its solution? "In-context learning" under different names is the standard approach for adapting policies, so what did we learn that's new here?
> >
> > > “Sec 4.2, how do you control for the ego agent's influence on the trajectory?” “I would have liked to see baselines/ablations related to their opponent-modeling proposal.”
> >
> > The distribution of trajectories depends on the joint policy of both the ego agent (say, the one doing the modeling) and the policy of the opponent. This means that the dataset is biased towards data from the ego agent which can limit the quality of the opponent model. I was curious if anything was done to specifically address this multiagent off-policy learning problem.
> >
> >
> > > Therefore, for low-strength opponent policies, their exploitation by the controlled agent can easily accumulate over time
> >
> > To clarify, this means the author agrees that there is an effect of the quality/diversity of the opponent population. I will concede that adapting to bad opponent policies is a reasonable test, but it seems like the more expected experiment would be learning against all performant opponents that play drastically differently.
> >
> > > BR Scores
> >
> > Thank you for including these results, they greatly aid the interpretation of the paper's results.
> >
> > > Figure 3(b) illustrates the performance of all approaches at the granularity of opponent policies (50 episodes). We aim to convey that our approach exhibits relatively competitive performance overall when facing different opponent policies.
> >
> > I am still unsure what the authors are saying here. The x-axis is "# of Opponent Policies" and goes from [0, 50]. However, this comment suggests that this should be episodes instead? I also am not sure what this means "performance of all approaches at the granularity of opponent policies," to me this is the BR scores you provided above.

---

> > > ### Author Response · Authors · 2023-11-21
> > > **Response to Official Comment by Reviewer F2KT (1/2)**
> > >
> > > Thank you very much for carefully reading our responses and increasing the score. We also appreciate your further comments to help improve our paper. We provide the following clarifications and feedback in response to your additional comments and questions. We hope our explanations and analyses address your concerns and further enhance your appreciation of our work.
> > >
> > > > “However, it's not clear to me that this generates self-consistent trajectories once the sampled segments are concatenated. I would have preferred this methodological detail be explicitly analyzed as it may confound results.”
> > > >
> > >
> > > In the design of TAO, in-context data serves as a raw representation of the opponent policy, used to recognize the current true opponent policy. There are no specific requirements for the form of in-context data, and it does not need to be a self-consistent real trajectory. We only need to ensure that the same form of in-context data is used during both the training and testing phases.
> > >
> > > The form of in-context data we finally adopted stems from the intuitions that given an opponent policy, its play style might be more pronounced over continuous timesteps (so we sample consecutive fragments), and it can showcase different playing behaviors across various episodes (so we sample from multiple trajectories). Through such a form of in-context data, we can represent opponent policies more comprehensively.
> > >
> > > In designing the form of in-context data, we also took into account the "confounding issue" you mentioned. Therefore, we added additional encoding to in-context data to address this problem. Specifically, we added *timestep embeddings* (indicating the current timestep of each token) and *trajectory index embeddings* (indicating which trajectory the token belongs to) to the token embeddings for each observation, action, and reward in the in-context data.
> > >
> > > > “However, the problem of building an opponent model can be studied in insolation, and he resulting model can be used beyond adaptation. I am strongly encouraging the authors to modify the language to not conflate the problems.”
> > > >
> > >
> > > Thank you very much for your clarification and explanation; your points are very valid. Indeed, we have somewhat expanded the concept of opponent modeling, as you mentioned. As you rightly pointed out, an opponent model can serve not only for adaptation but also for other downstream tasks. In this paper, our intention is to present a complete approach that not only learns an opponent model but also addresses the aspect of adaptation. We have clearly pointed this out in the revised manuscript to avoid any confusion. Once again, I appreciate your valuable input.
> > >
> > > > “What have we learned about the opponent modeling problem or in-context learning from applying transformers to be its solution? "In-context learning" under different names is the standard approach for adapting policies, so what did we learn that's new here?”
> > > >
> > >
> > > Our work introduces a novel and more general perspective to the field of opponent modeling. We integrate the knowledge of responding to diverse opponent policies into a carefully designed Transformer architecture tailored for opponent modeling. Through the use of in-context learning, we facilitate the retrieval of this knowledge and generalize it to the modeling and adaptation to unknown opponent policies.
> > >
> > > Theoretically, we have proved that our approach shares the same theoretical guarantees as the PS algorithm, which has favorable theoretical guarantees for generalization ([6]). The intuition behind this is further revealed in the analysis of Theorem 2: when facing a seen opponent policy, TAO can accurately recognize it. When facing an unseen opponent policy, TAO recognizes it as a combination of all seen opponent policies and utilizes *the knowledge of tackling all seen opponent policies* (*i.e.*, best responses to all seen opponent policies) to make appropriate responses. We believe that this ability to generalize against opponent policy knowledge is a valuable contribution lacking in the field of opponent modeling.
> > >
> > > Empirically, we have validated the effectiveness of our approach in some classic environments. The Transformer is widely recognized for its scalability. Therefore, we are eager to leverage the foundation laid by our current work to further explore the performance of the Transformer in larger-scale environments and with larger datasets, especially in modeling and adaptation aspects. We believe this could have foundational implications for the development of autonomous agents.
> > >
> > > [6] Ian Osband, et al. (more) efficient reinforcement learning via posterior sampling. *Advances in Neural Information Processing Systems*, 2013.

---

> > > ### Author Response · Authors · 2023-11-21
> > > **Response to Official Comment by Reviewer F2KT (2/2)**
> > >
> > > > “The distribution of trajectories depends on the joint policy of both the ego agent (say, the one doing the modeling) and the policy of the opponent. This means that the dataset is biased towards data from the ego agent which can limit the quality of the opponent model. I was curious if anything was done to specifically address this multiagent off-policy learning problem.”
> > > >
> > >
> > > Thank you for further clarification. From our understanding, we believe you are referring to the following issue: during deployment, the opponent policy may take actions in states that were not encountered during training, resulting in previously never seen in-context data during training. This, in turn, leads to off-policy learning for the controlled agent.
> > >
> > > This issue is indeed crucial, and we have taken some corresponding mitigation measures. In our offline dataset, given an opponent policy $\pi^{-1,k}$, the corresponding set of controlled agent trajectories $\mathcal{T}^{1,k}$ not only includes trajectories from mostly *approximate best responses* $\pi^{1,k,\ast}$ (expert policies against $\pi^{-1,k}$) but also includes some trajectories from *arbitrary other policies* (non-expert policies against $\pi^{-1,k}$). This way, we incorporate action information of the opponent policy $\pi^{-1,k}$ in more states, improving the representation of the opponent policy in the sampled in-context data during training. The resulting model may alleviate the off-policy issue when deployed.
> > >
> > > However, a comprehensive solution to the multiagent off-policy learning problem still requires careful research and discussion. It is a fascinating and worthwhile issue for further exploration, and we appreciate you bringing it up.
> > >
> > > > “To clarify, this means the author agrees that there is an effect of the quality/diversity of the opponent population. I will concede that adapting to bad opponent policies is a reasonable test, but it seems like the more expected experiment would be learning against all performant opponents that play drastically differently.”
> > > >
> > >
> > > In our current version, the opponent policy set we defined includes policies with different strengths and styles. However, as you pointed out, there is room for further improvement in the quality of our opponent policy set. We are employing the method of diversity-driven population training ([7][8][9]) to train a population of opponent policies that excel in both quality and diversity. We will evaluate the performance of all approaches on this new opponent policy set. We will try our best to complete the new experiments and add these results to the final version.
> > >
> > > [7] Parker-Holder, Jack, et al. "Effective diversity in population based reinforcement learning." *Advances in Neural Information Processing Systems* 33 (2020): 18050-18062.
> > >
> > > [8] Wu, Shuang, et al. "Quality-Similar Diversity via Population Based Reinforcement Learning." *The Eleventh International Conference on Learning Representations*. 2022.
> > >
> > > [9] Zhao, Rui, et al. "Maximum entropy population-based training for zero-shot human-ai coordination." *Proceedings of the AAAI Conference on Artificial Intelligence*. Vol. 37. No. 5. 2023.
> > >
> > > > “Thank you for including these results, they greatly aid the interpretation of the paper's results.”
> > > >
> > >
> > > Thank you again for pointing this out. We will incorporate this crucial result into the paper.
> > >
> > > > “I am still unsure what the authors are saying here. The x-axis is "# of Opponent Policies" and goes from [0, 50]. However, this comment suggests that this should be episodes instead? I also am not sure what this means "performance of all approaches at the granularity of opponent policies," to me this is the BR scores you provided above.”
> > > >
> > >
> > > In Figure 3(b), we present the results of an experiment with the following setup: all approaches play with an opponent who switches its policy every $E=50$ episode. The opponent, during each switch, uniformly samples a policy from the *mix opponent policy set* (*i.e.*, $\\{o1, o2, o3, o4, o5, o6, o7, o8\\}$). We conduct a total of $2500$ episodes, with $50$ switches in total. Therefore, you can interpret the x-axis as the "number of switches" in opponent policies. For instance, the 0th switch might sample $o2$, and the 1st switch might sample $o8$. We haven't explicitly labeled the *specific types of opponent policies* ($o1,...,o8$) in the figure. Specifically, each data point in the figure represents the average performance over $E=50$ episodes against the sampled opponent policy at that point.
> > >
> > > Again, Thank you for your extensive and valuable comments, which have greatly improved our paper. We hope we have addressed all your concerns. We look forward to your continued feedback and further support for our work.

---

> ### Author Response · Authors · 2023-11-23
> **Looking forward to feedback from Reviewer F2KT**
>
> Dear Reviewer F2KT, thanks again for your suggestion to strengthen this work. As the rebuttal period is ending today, we wonder if our response answers your ***follow-up*** questions. It would be greatly appreciated if you would re-evaluate our paper based on our responses. If there are any remaining uncertainties, we will make every effort to address them in the remaining limited time to ensure clarity and satisfaction. Thanks again for your very constructive and insightful feedback.

---

### Official Review · Reviewer_Z2nF · 2023-10-19

**Soundness:** 3 good
**Presentation:** 3 good
**Contribution:** 2 fair
**Rating:** 6
**Confidence:** 4

**Summary:**

This work studies the offline opponent modeling problem. To tackle this problem, the authors first use a policy embedding learning process to learn the embedding of opponent policies. This process adopts a generative loss to imitate opponents and a discriminative loss to discern different opponent policies. Next, the authors introduce a decoder that conditions the opponent policy embedding and the controlled agent's trajectory and outputs the agent's action. Through the supervised training based on offline data, the decoder learns to respond to the opponent policy given the policy embedding. After deployment, the proposed method collects the latest trajectories of opponents, generates corresponding embeddings, and samples the actions that adapt to the current opponent policy.

**Strengths:**

1. This paper is organized well and written clearly.

2. The authors perform extensive experiments and use various settings to support their claims.

**Weaknesses:**

1. The technical contribution of this paper is not very impressive. The main working flow of the proposed method is an integration of existing works and does not provide new insights.

2. The authors miss some works that should be compared with.

3. The environments used in the experiments are quite simple.

Please see more discussions in the following Questions section.

**Questions:**

1. In section 4.2, the function GetOffD begins by sampling $C$ trajectories from $\tau^{-1,k}$, following which it samples $H$ consecutive
fragments from each trajectory and ultimately stitches them together. The reason for this design should be explained more clearly. Specifically, during deployment (section 4.3), the method collects the latest $C$ trajectories which are reasonable. However, during the training, the sampled $C$ trajectories are often from different periods which may not be very related to $y_t$ and $a_t$. Why this would help with the training?

2. In experiments, the authors mention "previously unseen opponent policies". How to define "unseen"? Is it possible that an unseen policy is similar to a certain seen policy?

3. Why the win rate can have a negative value?

4. Some compared works are out-dated, e.g., He et al., 2016a. Why not compare with Zintgraf et al., 2021 mentioned in the related work section. In addition, there exist some related works which could be compared with. For example,

-- Rahman, Muhammad A., et al. "Towards open ad hoc teamwork using graph-based policy learning." International Conference on Machine Learning. PMLR, 2021.

-- Zhang, Ziqian, et al. "Fast Teammate Adaptation in the Presence of Sudden Policy Change." UAI 2023.

-- Zhu, Zhengbang, et al. "MADiff: Offline Multi-agent Learning with Diffusion Models." arXiv preprint arXiv:2305.17330 (2023).

All these works model agents' policy and they claim their method can adapt to different agent policies. I understand that their scenarios are cooperative systems. However, their methods are comparatively straightforward to adapt to the setting of this paper.

5. The environments used for evaluation are very simple. Is it possible to conduct experiments in more complex domains?

---

> ### Author Response · Authors · 2023-11-18
> **Response to Reviewer Z2nF (1/3)**
>
> Thank you very much for your recognition of our paper's writing, experimental design, and results, as well as for the valuable feedback you provided. In response to your comments, we would like to make the following clarifications and feedback. We hope our explanations and analyses can eliminate concerns and make you find our work stronger.
>
> > “The main working flow of the proposed method is an integration of existing works and does not provide new insights.”
> >
>
> We apologize for not letting you see our work's main contributions and insights. As Reviewer 1 mentioned, we are dedicated to addressing the challenging OOM problem. To do so, we employ a series of carefully designed and appropriate techniques to overcome the obstacles. Our components may not be very complex, yet complexity does not necessarily imply goodness. Our work forms a general, simple, and efficient approach to solving the OOM problem.
>
> Our approach design relies on the deep understanding of the OOM problem: 1) OOM emphasizes extracting semantic information from opponent policies, which, compared to a naive task, often involves much more complex semantics. Hence, we use PEL to characterize opponent policies. 2) OOM is essentially a task adaptation problem, well-suited for handling with in-context learning methods. Accordingly, we design offline stage 2 and the deployment stage to equip TAO with in-context learning capabilities. 3) Transformers have been empirically and theoretically proven to work in leveraging in-context learning capabilities, demonstrating outstanding performance in task adaptation ([1][2][3]). Therefore, we adopt a Transformer-based architecture.
>
> To help you understand, we summarize the main contributions of this work as follows: 1) Formalize an OOM problem to address inefficiencies and impracticalities in the existing opponent modeling domain. 2) Introduce a general in-context learning-based approach, TAO, and empirically demonstrate its effectiveness in addressing the OOM problem. 3) Theoretically prove the equivalence between TAO and PSOM algorithms, as well as the convergence of TAO in opponent policy recognition.
>
> [1] Luckeciano C Melo. Transformers are meta-reinforcement learners. In *International Conference on Machine Learning*, 2022.
>
> [2] Michael Laskin, et al. In-context reinforcement learning with algorithm distillation. In *International Conference on Learning Representations*, 2023.
>
> [3] Jonathan N Lee, et al. Supervised pretraining can learn in-context reinforcement learning. *arXiv preprint arXiv:2306.14892*, 2023.
>
> > “In section 4.2, the function GetOffD begins by sampling $C$ trajectories from $\mathcal{T}^{-1,k}$, following which it samples $H$ consecutive fragments from each trajectory and ultimately stitches them together. The reason for this design should be explained more clearly. ”
> >
>
> We sincerely apologize for the lack of clarity in this description. We have rewritten and revised this section to help you better understand our approach. The parts we changed are highlighted with blue color. Thank you very much for pointing this out; it greatly helps improve our manuscript. We kindly request your review of the updated version.
>
> We designed in-context data with the following motivations: Firstly, sampling continuous segments of opponent trajectories is because the opponent policy's style may manifest differences and characteristics over a continuous time segment. Compared to randomly sampling in-context data at each timestep, this approach may reduce learning difficulty. Secondly, randomly sampling such segments from $C$ trajectories, instead of sampling from a single trajectory, considers various situations and corresponding behaviors that the same opponent policy may encounter in different gameplays. This consideration may enhance the robustness of the learning process.
>
> In the early stages of our experiments, we also explored two different sampling methods for in-context data: 1) Randomly sample one trajectory from $\mathcal{T}^{-1,k}$ and then randomly sample $C\cdot H$ tuples of $(o^{-1},a^{-1},r^{-1})$ (not necessarily consecutive) from this trajectory. 2)Randomly sample one trajectory from $\mathcal{T}^{-1,k}$ and then randomly sample $C$ consecutive segments of $(o^{-1},a^{-1},r^{-1})$ with length $H$. However, empirically, these two methods did not perform as well as the final method we adopted. Nonetheless, there was only a minor difference in performance between the different sampling strategies for in-context data.

---

> > ### Comment · Reviewer_Z2nF · 2023-11-21
> > **Reply to rebuttal**
> >
> > Thanks for the response. I have carefully read the response and my concerns are addressed. The manuscript has also been improved substantially after the revision. For example, the method description is clearer now. There was a sentence "We conduct joint optimization of OPE and ICD to uncover the intricate relationships between the **current** opponent’s trajectory and the associated response actions". The "**current** opponent’s trajectory" can cause the misunderstanding that the sampled $C$ trajectories are **current** and should be related to $y_t$ and $a_t$. In the revision, this sentence has been revised. Moreover, I do agree that "complexity does not necessarily imply goodness" and I think we should also appreciate the "simple yet effective" method although the motivation and design of this work are akin to the works from related domains, e.g., decision transformer. Therefore, I have increased my score.
> >
> > In addition, I have one follow-up question. In Equation (5), the learning of response policy given the opponent trajectory requires the collection of the interaction data between different opponent policies and corresponding response policies. Will the required amount of data be huge?

---

> > > ### Author Response · Authors · 2023-11-21
> > >
> > > Thank you for carefully reviewing our response and improving the score. We also appreciate your further feedback to help enhance the quality of our paper. We provide the following clarifications and feedback in response to your additional comments. We hope our explanations and analyses further strengthen your recognition of our work.
> > >
> > > > “the learning of response policy given the opponent trajectory requires the collection of the interaction data between different opponent policies and corresponding response policies. Will the required amount of data be huge?”
> > > >
> > >
> > > Thank you for pointing this out; it is an important question. Data efficiency is a focal point in our considerations and a significant motivation for formalizing the OOM problem.
> > >
> > > Please allow us to use a real-life example to further illustrate our motivation: In the context of basketball, many top NBA players use the method of watching replay videos to study an opponent's playing style and weaknesses, devising corresponding strategies to counter them. Drawing inspiration from real-life intuitions like this, we aim to learn an opponent model that characterizes the semantics of opponent policies and generates appropriate responses with only not-too-many high-quality samples.
> > >
> > > In the current version, our offline dataset collected $2000$ episodes of samples for each opponent policy, which is much smaller than the quantities needed by most opponent modeling approaches adopting online learning settings. In earlier experiments, we found that our approach could perform well if the number of collected episodes exceeded certain thresholds. Therefore, we chose an appropriately sized value. We will supplement experiments regarding the impact of *offline dataset size* (how many episodes of samples are collected for each opponent policy) on performance and incorporate the results into the paper.
> > >
> > > Again, Thank you for your valuable feedback and valid assistance in improving our paper. We hope we have addressed your concerns. We welcome your new comments and look forward to your continued support.

---

> ### Author Response · Authors · 2023-11-18
> **Response to Reviewer Z2nF (2/3)**
>
> > “However, during the training, the sampled $C$ trajectories are often from different periods which may not be very related to $y_t$ and $a_t$. Why this would help with the training?”
> >
>
> The $C$ randomly sampled trajectories are all in-context data for a given opponent policy, and these data are primarily used to recognize the current opponent policy. Therefore, they do not necessarily need to have any direct association with $y_t, a_t$. They just need to be generated by the given opponent policy. Sampling such in-context data from multiple trajectories may enhance the robustness of the learning process. Recognition of the opponent policy is crucial for effectively utilizing $y_t, a_t$ to learn how to respond to the given opponent policy.
>
> > “How to define "unseen"? Is it possible that an unseen policy is similar to a certain seen policy?”
> >
>
> "Unseen" refers to opponent policies that have not appeared in the offline opponent policy set $\Pi^{\text{off}}$ (consisting of all opponent policies used to generate opponent trajectories in the offline dataset). The focus of this work is not on the similarity of policies within the opponent policy set. However, we have attempted to make the unseen policies as different as possible from the seen policies. Please refer to Appendix H for a detailed description of all opponent policies.
>
> > “Why the win rate can have a negative value?”
> >
>
> Thank you very much for your correction. We have changed the "win rate" to "score" in our new version to make it more rigorous.
>
> > “Some compared works are out-dated, e.g., He et al., 2016a. Why not compare with Zintgraf et al., 2021 mentioned in the related work section.”
> >
>
> Thank you very much for your suggestion. Zintgraf et al., 2021 is indeed a suitable baseline for comparison. We are happy to conduct additional experiments, and the results can be found at [this link](https://postimg.cc/cgKW6rjP). We will incorporate these important results into the paper in subsequent revisions.
>
> The new experiment involves MeLIBA (Zintgraf et al., 2021) against the *non-stationary unknown opponent* for 2500 games. The results in the figure represent the mean and standard deviation of average returns across five random seeds.
>
> It can be observed that with the addition of MeLIBA, TAO maintains relatively competitive performance among all approaches.
>
> > “In addition, there exist some related works which could be compared with. For example,… However, their methods are comparatively straightforward to adapt to the setting of this paper.”
> >
>
> Due to time constraints, we have added MeLIBA as a new baseline. As for the other three approaches you mentioned, we are currently trying our best to include them in our comparison. However, please understand that these three approaches are not easily adaptable to the OOM problem setting:
>
> Paper [4] involves computing graph-based value estimates with specific dependencies, focusing on scenarios where agents enter and leave at any time. The accurate estimation of values heavily relies on online learning. However, the OOM setting does not primarily focus on value estimation. Our offline dataset makes it challenging to accurately compute certain values, such as the Joint Action Value mentioned in [4].
>
> Papers [5] and [6] adopt a learning approach with Centralized Training Decentralized Execution (CTDE), which is a paradigm limited to cooperative environments. This training paradigm is relatively dependent on centralized value estimates, and in the OOM setting, the offline dataset does not include information to compute these values. Additionally, our work focuses on OOM in competitive environments, making training paradigms like CTDE less suitable as the controlled agent and opponents have different reward functions and objectives. Furthermore, there are significant differences between the setting in [6] and OOM; they focus on using an offline dataset to learn how to simultaneously control multiple agents to collaborate effectively, whereas our focus is on modeling and adapting to other agents using an offline dataset.
>
> [4] Rahman, Muhammad A., et al. "Towards open ad hoc teamwork using graph-based policy learning." International Conference on Machine Learning. PMLR, 2021.
>
> [5] Zhang, Ziqian, et al. "Fast Teammate Adaptation in the Presence of Sudden Policy Change." UAI 2023.
>
> [6] Zhu, Zhengbang, et al. "MADiff: Offline Multi-agent Learning with Diffusion Models." arXiv preprint arXiv:2305.17330 (2023).

---

> ### Author Response · Authors · 2023-11-18
> **Response to Reviewer Z2nF (3/3)**
>
> > “The environments used for evaluation are very simple. Is it possible to conduct experiments in more complex domains?”
> >
>
> Our experimental environment is based on classic benchmarks commonly used in the opponent modeling domain. [7], [8], [9], [10], [11], [12], [13] also utilize environments similar or identical to MS adopted in our study.
>
> [7] Michael L Littman. Markov games as a framework for multi-agent reinforcement learning. In
>
> *Machine Learning Proceedings*, 1994.
>
> [8] He He, et al. Opponent modeling in deep reinforcement learning. In *International Conference on Machine Learning*, 2016a.
>
> [9] Zhang-Wei Hong, et al. A deep policy inference q-network for multi-agent systems. In *International Conference on Autonomous Agents and MultiAgent Systems*, 2018.
>
> [10] Yan Zheng, et al. A deep bayesian policy reuse approach against non-stationary agents. *Advances in Neural Information Processing Systems*, 2018.
>
> [11] Tianpei Yang, et al. Towards efficient detection and optimal response against sophisticated opponents. In *International Joint Conference on Artificial Intelligence*, 2019.
>
> [12] Manish Prajapat, et al. Competitive policy optimization. In *Uncertainty in Artificial Intelligence*, 2021.
>
> [13] Zhe Wu, et al. L2e: Learning to exploit your opponent. In *International Joint Conference on Neural Networks*, 2022.
>
> Papers [14], [15], [16], [17], [18], including the ones you mentioned [5], [6], also utilize environments similar or identical to PA adopted in our study, such as the Predator-Prey scenario in the Multi-Agent Particle Environment.
>
> [14] Ying Wen, et al. Probabilistic recursive reasoning for multi-agent reinforcement learning. In *International Conference on Learning Representations*, 2019.
>
> [15] Georgios Papoudakis, et al. Variational autoencoders for opponent modeling in multi-agent systems. *arXiv preprint arXiv:2001.10829*, 2020.
>
> [16] Georgios Papoudakis, et al. Agent modelling under partial observability for deep reinforcement learning. *Advances in Neural Information Processing Systems*, 2021.
>
> [17] Haobo Fu, et al. Greedy when sure and conservative when uncertain about the
>
> opponents. In *International Conference on Machine Learning*, 2022.
>
> [18] Xiaopeng Yu, et al. Model-based opponent modeling. *Advances in Neural Information Processing Systems*, 2022.
>
> If you are willing to recommend some suitable environments, we would be happy to evaluate on them.
>
> All your questions and feedback have contributed significantly to improving our paper. We hope we have addressed all your concerns. Feel free to continue providing comments; we will promptly consider your suggestions for further revisions.

---

> ### Author Response · Authors · 2023-11-23
> **Looking forward to feedback from Reviewer Z2nF**
>
> Dear Reviewer Z2nF, thanks again for your suggestion to strengthen this work. As the rebuttal period is ending today, we wonder if our response answers your ***follow-up*** questions. It would be greatly appreciated if you would re-evaluate our paper based on our responses. If there are any remaining uncertainties, we will make every effort to address them in the remaining limited time to ensure clarity and satisfaction. Thanks again for your very constructive and insightful feedback.

---

> > ### Comment · Reviewer_Z2nF · 2023-11-23
> >
> > Yes. My follow-up questions are addressed. Thanks for the response.

---

> > > ### Author Response · Authors · 2023-11-23
> > >
> > > Thank you very much for your response. We are pleased to have addressed all your concerns, which improved our paper significantly. BTW, Happy Thanksgiving Day.

---

### Official Review · Reviewer_QAav · 2023-10-31

**Soundness:** 3 good
**Presentation:** 2 fair
**Contribution:** 3 good
**Rating:** 6
**Confidence:** 4

**Summary:**

This paper proposes a new setting for the opponent modeling problem (OOM) where environment and opponent policies are unavailable. Under this setting, the authors introduce a novel model, named Transformer Against Opponent (TAO). TAO adopts Transformer’s capabilities on sequence modeling and in-context learning to learn discriminative policy embedding and predict response actions. The authors provide a theoretical analysis that establishes TAO's equivalence to Bayesian posterior sampling in opponent modeling and guarantees TAO's convergence in opponent policy recognition. The paper also includes empirical analysis to demonstrate TAO's effectiveness in achieving competitive results against other approaches for the OOM problem.

**Strengths:**

1. The paper is overall well-structured and easy to follow. The authors provide a comprehensive analysis of the problem they are addressing and the existing work in the field.

2. The new setting of learning opponents offline seems reasonable to me. And the idea of modeling opponent policies as an in-context learning problem is interesting.

3. Experiment results demonstrate the superior performance of TAO on the OOM problem.

**Weaknesses:**

My concerns mainly lie in the experiment section.

-	The authors should give a more detailed description of the constructed datasets which are the key points of this paper.

-	For the mix type, what is the proportion of the number of the opponent policies between seen and unseen? How do the models, including TAO and baselines, perform on different proportions?

-	What’s the meaning of “# of opponent policy” in Fig 3b and Fig 6b? Does it stand for the number of policy items or the number of policy types, for seen or unseen or total? From the results in Fig 3b and Fig 6b, in the PA dataset, the model's performance exhibits a noticeable trend of initially improving and then declining under different number of opponent policies. However, in the MS dataset, while the model's performance fluctuates, there is no clear trend. Why is there such a difference and what causes this performance variation?

-	The authors should also provide a visualization of unseen policies in Fig 5a to make the demonstration more convincing.

Minor issue:
Para 2 in introduction, numerous real-life …**: For instance,** …

**Questions:**

-	Does GPT2 stay frozen or get fine-tuned in policy embedding learning?

-	Under the setting of OOM, no opponent policy should be available for learning except for trajectory sampling. Does the discriminative loss cause policy label exposure to the model and violate the setting?

---

> ### Author Response · Authors · 2023-11-18
> **Response to Reviewer QAav (1/2)**
>
> Thank you very much for your recognition of our paper's writing, problem setting, experimental results, and the valuable feedback you provided. In response to your comments, we would like to make the following clarifications and feedback. We hope our explanations and analyses can eliminate concerns and make you find our work stronger.
>
> > “a more detailed description of the constructed datasets”
> >
>
> Considering the space limitations, we detailed the dataset's construction in Appendix I. We would be more than willing to revise the paper in subsequent iterations to provide a more detailed description of this crucial parts in the main text.
>
> > “For the mix type, what is the proportion of the number of the opponent policies between seen and unseen?”
> >
>
> Our opponent policy configurations are as follows (excluding those in Question 4): $o1, o2, o3, o4, o5$ are opponent policies in the *seen* set, while $o6, o7, o8$ are opponent policies in the *unseen* set. Therefore, in the *mix* set, the ratio of *seen* to *unseen* is $5:3$.
>
> > “How do the models, including TAO and baselines, perform on different proportions?”
> >
>
> In response to this question, we are pleased to provide additional experiments. You can find the results at [this link](https://postimg.cc/Vr38tzPv). We will incorporate these important results into the paper in subsequent revisions.
>
> We randomly select several opponent policies from the *mix* opponent policy set according to various **[Seen:Unseen=a:b]** ratios to create a *non-stationary unknown* opponent who switches policy periodically. All approaches engage in 2500 episodes of plays against this opponent. The results in the above figure represent the mean and standard deviation of average returns across five random seeds.
>
> It can be observed that TAO demonstrates competitive performance across different ratios of the *mix* set.
>
> > “What’s the meaning of “# of opponent policy” in Fig 3b and Fig 6b? Does it stand for the number of policy items or the number of policy types, for seen or unseen or total?”
> >
>
> We apologize for any confusion. It stands for the number of policy items, just like *timestep/episode*, but with a coarser granularity. In our experimental setup, we switch opponent policy every $E$ episodes. By "switch," we mean randomly sampling an opponent policy from the corresponding opponent policy set according to a uniform distribution. Taking the *mix* opponent policy set as an example, there are a total of $8$ opponent policy types, and "switching" entails randomly sampling an opponent policy from $o1, o2, ..., o8$.
>
> > “Why is there such a difference and what causes this performance variation?”
> >
>
> The performance gap is primarily caused by the varying strengths of different opponent policies. We designed opponent policies with differentiated styles and strengths to make them diverse. For example, in the "PA-mix" of Figure 3b, the initially sampled opponents become weaker over time, leading to an overall upward trend in performance for nearly all approaches. However, the later sampled opponents are relatively stronger, resulting in lower absolute performance values for all approaches. In practice, it is more meaningful to focus on the relative performance between different approaches rather than absolute values. This is because some opponent policies are inherently strong, and the absolute performance values of the approximate best response against them are lower.
>
> > “The authors should also provide a visualization of unseen policies in Fig 5a to make the demonstration more convincing.”
> >
>
> The purpose of plotting Figure 5a is to verify whether OPE can effectively distinguish between different opponent policies in the *seen* set, not to focus on the position of unseen opponent policies in the embedding space. This is because we expect to learn the best response corresponding to each opponent policy based on the embedding learned by OPE. An embedding with better discriminability can intuitively facilitate the learning of response policies.
>
> When facing unseen opponent policies, we are interested in how TAO generalizes knowledge about responding to seen opponent policies to play against them. For an unseen opponent policy, if its trajectory is similar to one generated by a seen opponent policy, the two trajectory embeddings will be close in the embedding space. In this case, TAO can leverage knowledge about responding to the seen opponent policy to respond to the unseen opponent policy. If the trajectory of the unseen opponent policy is not similar to any seen opponent policy, its embedding might be a combination of embeddings from various seen opponent policies. TAO will appropriately combine the knowledge of responding to each seen opponent policy in that combination to generate a suitable response. Such intuitions stem from the discussions and analyses in the proving process of Theorem 2. Please refer to Appendix D.3 for relevant details.

---

> ### Author Response · Authors · 2023-11-18
> **Response to Reviewer QAav (2/2)**
>
> > “Minor issue: Para 2 in introduction, numerous real-life …**: For instance,** …”
> >
>
> Thanks for your correction. We have revised this in the new version.
>
> > “Does GPT2 stay frozen or get fine-tuned in policy embedding learning?”
> >
>
> During the offline stage 1 (PEL), we train a model from scratch, starting with an initial model. Our model adopts the architecture design of GPT-2 rather than fine-tuning from any pre-trained weights of GPT-2.
>
> > “Under the setting of OOM, no opponent policy should be available for learning except for trajectory sampling. Does the discriminative loss cause policy label exposure to the model and violate the setting?”
> >
>
> In the OOM setting, we can only access trajectories generated by different opponent policies and cannot directly call the opponent policy itself. However, in the offline dataset, each opponent trajectory can have a label indicating which opponent policy type it belongs to. We use these trajectories and labels to train OPE, aiding the controlled agent in recognizing different opponent policies. This does not violate the problem setting of OOM.
>
> All your questions and feedback have led to significant improvements in our paper. We hope we have addressed all your concerns. Feel free to continue providing comments; we will promptly consider your opinions and make revisions accordingly.

---

> ### Author Response · Authors · 2023-11-21
>
> Thanks again for your time and effort in reviewing our paper! As the discussion period is coming to a close, we would like to know if we have resolved your concerns expressed in the original reviews. We remain open to any further feedback and are committed to making additional improvements if needed. If you find that these concerns have been resolved, we would be grateful if you would consider reflecting this in your rating of our paper.

---

> ### Author Response · Authors · 2023-11-23
> **Looking forward to feedback from Reviewer QAav**
>
> Dear Reviewer QAav, thanks again for your suggestion to strengthen this work. As the rebuttal period is ending today, we wonder if our response answers your questions. It would be greatly appreciated if you would re-evaluate our paper based on our responses. If there are any remaining uncertainties, we will make every effort to address them in the remaining limited time to ensure clarity and satisfaction. Thanks again for your very constructive and insightful feedback.

---

### Official Review · Reviewer_fLaD · 2023-11-01

**Soundness:** 1 poor
**Presentation:** 1 poor
**Contribution:** 2 fair
**Rating:** 5
**Confidence:** 3

**Summary:**

This paper proposes an offline RL method specifically for domains with other agents.
The method consists of a transformer model that is used to encode the opponent trajectory (called opponent policy embedding, OPE) in some latent space and a transformer that, with this encoding as input, is trained to predict the best action through behavioral cloning (called in-context control decoder, ICD).

OPE is trained with two (additive) losses of which one is based on the likelihood of the predictions over the opponent policy and the other on a discriminative loss over the trajectory index.
ICD is trained on, as far as I can tell, the typical behavioral cloning loss (likelihood of predicting the action take in the offline data set).

**Strengths:**

This paper tackles a difficult problem, that of adapting quickly to (sometimes unseen) opponents.
It does so with an arguably scalable approach (e.g. with transformers) that can be deployed online is a real-time manner by pre-computing the heavy work that many algorithms do online instead (to infer their opponents policy).

The solution, additionally, proposes a significant amount of clever and proper uses of existing techniques to tackle the many obstacles that come with a complex problem like theirs.

**Weaknesses:**

My main concern is on the lack of clarity and theoretical rigor in this paper.
Additionally, while there was motivation for the problem setup and high-level design choices, most of the proposed method lacked description or reasoning.

For example, a common occurrence in the paper is the mathematical description of a loss used with barely any description of what it achieves (I *believe* equation 5 is typical behavioral cloning; supervised learning on the action in the data set, but I had to infer this), often including cryptic terms as "GetOnD" which "samples fragments from each trajectory of W and then stitches them together".
It was sometimes unclear what even the intended task was of a particular component in the model and necessary to reverse engineer it from the proposed solution.

Lack of rigor is especially visible in the theoretical analysis where the first theorem reads as "assume that (...) M is (...), then <statement>" but (I assume) the notation is such that <statement> does not even mention "M", making it unclear why that assumptions affects the statement at all to begin with.
The second theorem states their method has "better optimally guarantees" but does not mention what "better" means in this context.

As a result this paper reads as a recipe for training a particular model to solve their a problem setting of which the generality is difficult to judge.
This is to the detriment of the paper because, despite theirr focus on opponent modeling, I would not be surprised if (with some notation/generality) adjustments this could just as easily be applied to any (non-opponent) meta-learning tasks.

**Questions:**

In what way, during execution, do we assume that the agent has access to the opponent trajectory?

---

> ### Author Response · Authors · 2023-11-18
> **Response to Reviewer fLaD (1/2)**
>
> Thank you very much for your valuable feedback. In response to your comments, we would like to make the following clarifications and feedback. We hope our explanations and analyses can eliminate concerns and make you find our work stronger.
>
> > “My main concern is on the lack of clarity and theoretical rigor in this paper.” “most of the proposed method lacked description or reasoning” “For example, a common occurrence in the paper is the mathematical description of a loss used with barely any description of what it achieves” “It was sometimes unclear what even the intended task was of a particular component in the model and necessary to reverse engineer it from the proposed solution.”
> >
>
> We apologize for any confusion caused by the current version, acknowledging that the logic may not be sufficiently sound and the expressions not entirely clear. This is caused by the compression and omission of many intuitive analyses and explanations due to space limitations. Indeed, we haven't effectively conveyed the motivations, insights, and results of each component design in our approach. Therefore, we take your valuable suggestions into account and proceed to rewrite and revise the section "Transformer Against Opponent." The parts we changed are highlighted with blue color. We hope this could help give you a clearer and more profound understanding of our approach. Thank you very much for pointing this out; it greatly helps improve our manuscript. We kindly request your review of the updated version.
>
> > “I *believe* equation 5 is typical behavioral cloning”
> >
>
> Equation 5 and Behavior Cloning (predicting actions conditioned on observations) appear similar. However, strictly speaking, there are some distinctions between them. Equation 5 represents a form of autoregressive supervised learning to predict the controlled agent's (near-optimal) actions conditioned on the opponent's in-context data and the controlled agent's history. It serves as a crucial foundation for TAO to perform in-context learning during deployment.
>
> > “making it unclear why that assumptions affects the statement at all to begin with. The second theorem states their method has "better optimally guarantees" but does not mention what "better" means in this context.”
> >
>
> We apologize for the omission of many notations, leading to unclear expressions. This is because of space constraints in the main text, and we placed most symbolic definitions and explanations in the appendix. We have incorporated your constructive feedback and revised the theoretical analysis section. We have rephrased the parts lacking rigor, providing a clearer presentation of our theoretical results and the insights they offer. The parts we changed are highlighted with blue color. Here, we supplement some key explanations to address your concerns and hope to enhance your understanding of our theoretical results:
>
> - In Theorem 1, we omitted the following formal reasoning: $P_{\text{TAO}}(\xi^1_H|\mathcal{W},\bar{\pi}^{-1})\doteq P(\xi^1_H|\mathcal{W},\bar{\pi}^{-1},M_{\theta})$. Under the assumption D.1, it can be derived that $P(\xi^1_H|\mathcal{W},\bar{\pi}^{-1},M_{\theta})=P_{\text{pre}}(\xi^1_H|\mathcal{W},\bar{\pi}^{-1})$, where $P_{\text{pre}}(\xi^1_H|\mathcal{W},\bar{\pi}^{-1})\doteq P(\xi^1_H|\mathcal{W},\bar{\pi}^{-1},P_\text{pre})$. Subsequently, leveraging Lemma D.2 and Lemma D.3, we employ mathematical induction to prove that $P_{\text{pre}}(\xi^1_H|\mathcal{W},\bar{\pi}^{-1})=P_{\text{PSOM}}(\xi^1_H|\mathcal{W},\bar{\pi}^{-1})$ holds (please refer to Appendix D.2 for a detailed process). Intuitively, Theorem 1 elucidates that given the true opponent policy and the opponent's in-context data, TAO and PSOM algorithms yield equal probabilities for generating any controlled agent's trajectory. In other words, they would elicit the same responses to a given true opponent policy.
> - In Theorem 2, we prove that the PSOM algorithm is guaranteed to converge, and it converges to a solution that is "better" than the PS algorithm. Here, "better" means that if PS converges to the optimal solution, PSOM must also converge to the optimal solution. In intuitive terms, when the current true opponent policy belongs to $\Pi^{\text{off}}$, if PS correctly recognizes the current true opponent policy, PSOM is guaranteed to recognize it correctly. Even if PS fails to recognize the current true opponent policy correctly, PSOM still has a possibility of making a correct recognition. Based on Theorem 1, TAO shares the same properties as the PSOM algorithm mentioned above. For the optimality of the PS algorithm, please refer to the analysis of its regret bounds in [1].
>
> [1] Ian Osband, et al. (more) efficient reinforcement learning via posterior sampling. *Advances in Neural Information Processing Systems*, 2013.

---

> ### Author Response · Authors · 2023-11-18
> **Response to Reviewer fLaD (2/2)**
>
> > “As a result this paper reads as a recipe for training a particular model to solve their a problem setting of which the generality is difficult to judge.”
> >
>
> We focus on developing an offline opponent modeling approach to address this community's previously neglected inefficiencies and impracticalities. By inefficiencies, we argue that online learning of opponent models bears substantial complexity and inefficiency. As with many existing algorithms, they gather copious data and undertake intricate online updating processes. By impracticalities, we argue that online learning struggles to satisfy real-world scenarios, given that interactions with opponent policies are not consistently accessible.
>
> Our approach design relies on the deep understanding of the OOM problem: 1) OOM emphasizes extracting semantic information from opponent policies, which, compared to a naive task, often involves much more complex semantics. Hence, we use PEL to characterize opponent policies. 2) OOM is essentially a task adaptation problem, well-suited for handling with in-context learning methods. Accordingly, we design offline stage 2 and the deployment stage to equip TAO with in-context learning capabilities. 3) Transformers have been empirically and theoretically proven to work in leveraging in-context learning capabilities, demonstrating outstanding performance in task adaptation ([2][3][4]). Therefore, we adopt a Transformer-based architecture.
>
> [2] Luckeciano C Melo. Transformers are meta-reinforcement learners. In *International Conference on Machine Learning*, 2022.
>
> [3] Michael Laskin, et al. In-context reinforcement learning with algorithm distillation. In *International Conference on Learning Representations*, 2023.
>
> [4] Jonathan N Lee, et al. Supervised pretraining can learn in-context reinforcement learning. *arXiv preprint arXiv:2306.14892*, 2023.
>
> > “if (with some notation/generality) adjustments this could just as easily be applied to any (non-opponent) meta-learning tasks.”
> >
>
> Please note that our main focus in this paper is opponent modeling under an offline setting, i.e., OOM. We aim to develop a general, efficient, and practical approach, TAO, for offline opponent modeling to address the inefficiencies and impracticalities in current opponent modeling research. Furthermore, we demonstrate that the TAO can theoretically guarantee the convergence of opponent policy recognition and achieve good results empirically.
>
> For other meta-learning tasks, our approach can be adapted in a formal and theoretical manner as well. This is because our approach possesses a certain universality and generality. However, it needs to be tailored based on the characteristics of the specific problem settings, and rigorous theoretical analysis and experimental validation are still required.
>
> > “In what way, during execution, do we assume that the agent has access to the opponent trajectory?”
> >
>
> Considering the space constraints, we explained this in the paragraph "**Accessibility of opponent information**" in Appendix B: during execution, we assume that the controlled agent (controlled by all baselines) can collect the complete trajectory of the current opponent in an episode only after this episode ends. The opponent trajectory consists of many consecutive $(o^{-1},a^{-1},r^{-1})$ sequences. This is a common and reasonable assumption in the field of opponent modeling, as seen in [5][6][7]. This assumption allows for limited opponent information gathering, enabling different approaches to adapt to unknown opponent policies based on their own mechanisms during testing.
>
> [5] Yan Zheng, et al. A deep bayesian policy reuse approach against non-stationary agents. *Advances in Neural Information Processing Systems*, 2018.
>
> [6] Luisa Zintgraf, et al. Deep interactive bayesian reinforcement learning via meta-learning. In *International Conference on Autonomous Agents and MultiAgent Systems*, 2021.
>
> [7] Haobo Fu, et al. Greedy when sure and conservative when uncertain about the opponents. In *International Conference on Machine Learning*, 2022.
>
> All your comments have greatly contributed to the improvement of our paper. With the valuable feedback from you and all the other reviewers, the quality of our manuscript has been significantly enhanced. If you have any new comments, please feel free to provide them, and we will promptly address and incorporate them. If you find that we have addressed your concerns and the changes made to the paper are reasonable, we hope you will reconsider your rating.

---

> > ### Comment · Reviewer_fLaD · 2023-11-20
> >
> > Thank you for the detailed response. I also appreciate the changes made to the paper and will increment my score accordingly.
> > I believe more substantial changes to the paper are necessary for a confident accept, but am willing to change my mind if other reviewers push for it.

---

> > > ### Author Response · Authors · 2023-11-20
> > > **Thank you**
> > >
> > > Thank you very much for carefully reviewing our responses and raising the score. In summary, our new version incorporates the following modifications:
> > >
> > > - We have revised the "Transformer Against Opponent" section to enhance logical coherence and clarity of expression of methodology parts and theoretical analysis. For the methodology parts, we provided additional motivation, insights, and results behind each component design. For the theoretical analysis, we rewrote the original statements that lack rigor, making them more precise and comprehensive. Additionally, we provided insights derived from our theoretical results.
> > > - We have revised the "Experiments" section, addressing exaggerated language and imprecise descriptions to make it more objective and rigorous. Additionally, we have provided more detailed descriptions and analysis of the experimental results to better showcase the findings and insights derived from the experiments.
> > > - We have added additional experimental results (these new results are currently included in the responses to the reviewers due to time constraints, and we will update the complete new results in our paper). These additions encompass:1) Experiment results for all methods under the *mix* setting, facing different ratios of [seen:unseen] opponent policies. 2) Introduction of a new baseline, MeLIBA (Zintgraf et al., 2021, *i.e.*, [6]), in experiments against the *non-stationary unknown* opponent during testing. 3) Performance results for the approximate best response of each opponent policy in all environments. 4) Quantitative analysis of the policy embeddings TAO learned in terms of their ability to recognize different opponent policies. Following the valuable suggestion from Reviewer F2KT, we trained a classifier on the frozen embeddings to evaluate its performance in classifying their originating opponent policies. These additional experiments aim to provide a more comprehensive evaluation of our approach.
> > >
> > > We believe that our revised version has addressed the concerns raised by the reviewers and has undergone a significant quality improvement based on the reviewers' feedback.

---

> ### Author Response · Authors · 2023-11-21
>
> Thanks again for your time and effort in reviewing our paper! As the discussion period is approaching, we would like to know if you have any further concerns. We are committed to making additional improvements if needed. We would also like to cordially invite you to review the discussions between us and the other reviewers, which we believe will deepen your understanding and appreciation of our work. We look forward to your further support for our paper.

---

### Meta-Review · Area_Chair_6WYs · 2023-12-07

**Metareview:**

This paper proposes an architecture for doing offline opponent modelling. The key idea is to use in-context learning instantiated via a transformer architecture.

Specifically, the authors suggest to first learn an embedding space for the opponent behaviour and then to learn a generalised approximate best response function to each of the policies using supervised learning.

The strengths of the paper are:
1) It is overall easy to follow at a high level
2) The problem setting of offline opponent modelling and the overall approach are interesting.
3) The experimental analysis is thorough and follows best practice from the opponent modelling domain

There are some weaknesses which the reviewers picked up on:
1) The baselines seem a little outdated and should be updated to include more current work.
2) It is unclear whether the "in-context learning" aspect and the allusion to "pretrained transformers" is mostly branding, since the authors train the transformers from scratch.

Given the balance of strength and weaknesses I am leaning accept for this paper, assuming that there is sufficient capacity. I also recommend that for the final version the authors ablate the architecture e.g. by replacing the transformer with an LSTM, and include the additional baselines requested by the reviewers.

**Justification For Why Not Higher Score:**

Some relevant experimental evaluations are missing.

**Justification For Why Not Lower Score:**

I believe that overall the paper makes a good contribution.

---

### Decision · Program_Chairs · 2024-01-16

Accept (poster)